# MolecularIQ: Characterizing Chemical Reasoning Capabilities Through Symbolic Verification on Molecular Graphs

**Christoph Bartmann**[1,*]  **Johannes Schimunek**[1,*]  **Mykyta Ielanskyi**[1]  **Philipp Seidl**[1]
**Günter Klambauer**[1,2]  **Sohvi Luukkonen**[1]

[1]ELLIS Unit Linz and LIT AI Lab, Institute for Machine Learning,
Johannes Kepler University Linz, Austria
[2]Clinical Research Institute for Medical Artificial Intelligence,
Johannes Kepler University Linz, Austria
*Equal contribution.

## Abstract

A molecule's properties are fundamentally determined by its composition and structure encoded in its molecular graph. Thus, reasoning about molecular properties requires the ability to parse and understand the molecular graph. Large Language Models (LLMs) are increasingly applied to chemistry, tackling tasks such as molecular name conversion, captioning, text-guided generation, and property or reaction prediction. Most existing benchmarks emphasize general chemical knowledge, rely on literature or surrogate labels that risk leakage or bias, or reduce evaluation to multiple-choice questions. We introduce MolecularIQ, a molecular structure reasoning benchmark focused exclusively on symbolically verifiable tasks. MolecularIQ enables fine-grained evaluation of reasoning over molecular graphs and reveals capability patterns that localize model failures to specific tasks and molecular structures. This provides actionable insights into the strengths and limitations of current chemistry LLMs and guides the development of models that reason faithfully over molecular structure.

## 1 Introduction

**From task-specific chemoinformatics models to unified LLM-based chemistry assistants.** Recent advances in autonomous robotic laboratories and agentic large language model (LLM)-based systems bear the promise of end-to-end automation of the scientific workflow, from hypothesis generation to experiment execution (MacLeod et al., 2020; Tom et al., 2024; Lu et al., 2024a; Yamada et al., 2025). In these emerging pipelines, LLMs are not merely conversational interfaces: they act as decision-making engines that propose experiments, design molecules, and interpret results. In parallel, general-purpose and chemistry-specialized LLMs are being adopted as accessible tools for molecular reasoning and property prediction. A key appeal of the language model approach to chemical problems is that classical chemoinformatics models are typically task-specific (e.g., property prediction, reaction prediction, or de novo design), whereas a single LLM could, in principle, address all of these tasks within one unified interface. Furthermore, these models have the potential to lower barriers for non-experts, including wet-lab chemists, to perform tasks that traditionally required dedicated chemoinformatics expertise. Consequently, the interest in both evaluating the chemistry capabilities of general LLMs (Guo et al., 2024; Mirza et al., 2025; Huang et al., 2024; Lu et al., 2024b; Runcie et al., 2025) and developing domain-specialized chemistry LLMs (Liu et al., 2023b;a; Zhang et al., 2024; Yu et al., 2024; Liu et al., 2024; Fang et al., 2024; Li et al., 2024b; Zeng et al., 2024; Narayanan et al., 2025; Zhao et al., 2025a;c; Ye et al., 2025; Wang et al., 2025a; Tang et al., 2025) has surged.

Leaderboard: https://huggingface.co/spaces/ml-jku/molecularIQ_leaderboard
Code: https://github.com/ml-jku/moleculariq

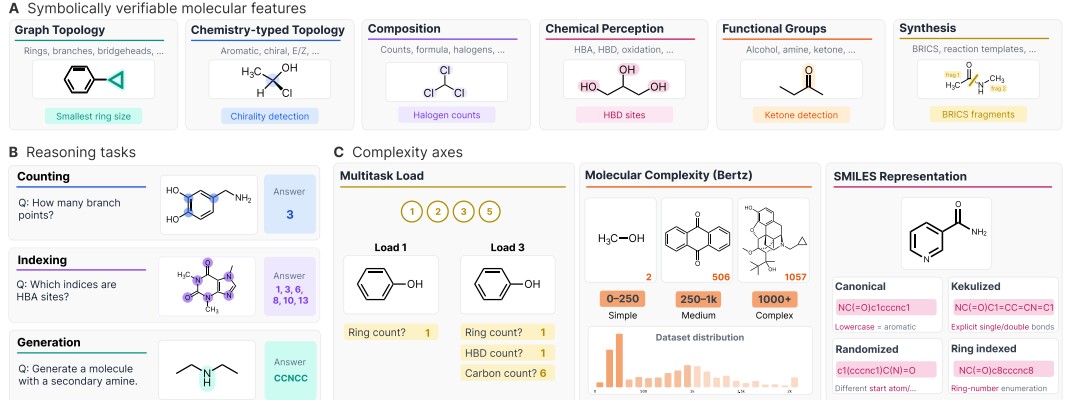

Figure 1: Overview of MolecularIQ benchmark for measuring chemical structure reasoning capabilities of LLMs.

**Structural understanding is the prerequisite for molecular reasoning, not just one capability among many.** A fundamental principle in chemistry and chemoinformatics is the structure-property relationship, which means that the molecular structure of an organic molecule determines its properties. Thus, a model that cannot reliably identify functional groups, ring systems, or atomic connectivity cannot be expected to reason about how these features influence reactivity, toxicity, or biological activity. Structural understanding is, consequently, a key prerequisite upon which all meaningful molecular modification depends.

**Current benchmark design obscures whether LLMs truly reason over molecular structure.** However, current chemistry benchmarks are ill-suited to testing this capability in LLMs. Benchmarks emphasizing general chemistry knowledge mainly measure factual recall, while property and reaction benchmarks rely on public datasets, which likely appear in LLM training corpora, making it impossible to disentangle structural reasoning from pattern recognition and memorization of molecule–property pairs (Carlini et al., 2021). Furthermore, when exact ground truths are unavailable, some benchmarks rely on surrogate verifiers (predictive models or heuristics), which can introduce bias and be exploited, so higher scores do not necessarily reflect genuine reasoning Renz et al. (2019).

**Symbolically verifiable tasks serve as a diagnostic for molecular reasoning competence.** Symbolically verifiable graph-based tasks provide a testbed for evaluating structural reasoning. Since LLMs process molecules as 1D token sequences (e.g., SMILES strings), these tasks implicitly assess the model's ability to infer 2D graph topology from linear representations. Because every answer can be computed exactly from the molecular graph, symbolically generated ground truths guarantee label correctness and eliminate surrogate verifier bias; moreover, while input molecules may appear in pretraining corpora, the specific task formulations and multi-property queries are algorithmically generated, reducing the likelihood of direct answer recall. Although the tasks we consider can be solved trivially by cheminformatics software, their value as evaluation tools lies precisely in this property: they establish a floor below which no model should fall if it has genuinely internalized molecular structure. In turn, a model that appears strong on property prediction benchmarks yet fails at basic substructure identification is likely exploiting dataset-specific correlations rather than reasoning over structure.

**MOLECULARIQ: a fully symbolically verifiable benchmark for assessing reasoning over molecular structures.** Building on this perspective, we introduce MOLECULARIQ, a chemical structure reasoning benchmark composed exclusively of symbolically verifiable tasks defined on molecular graphs. MOLECULARIQ covers three task types – feature counting and indexing, and constrained generation – and spans three complexity dimensions – multitask load, molecule complexity, and molecule representation. This multi-dimensional design enables controlled, fine-grained diagnosis of where and why models fail, complementing existing chemistry benchmarks with a targeted evaluation of internalized structural competence. Figure 1 gives an overview of the MOLECULARIQ benchmark.

**Contributions:**

- We introduce MOLECULARIQ, the first fully symbolically verifiable benchmark for molecular understanding, where every answer can be programmatically validated against the molecular graph. It spans three task types and three complexity axes, enabling granular diagnosis of structural reasoning capabilities.
- We conduct a comprehensive evaluation of 38 generalist, chemistry-specialized, and reasoning-optimized LLMs, revealing that structural understanding remains a critical bottleneck: models that cannot reliably parse molecular graphs lack the foundation required for trustworthy molecular reasoning.
- We release open infrastructure for advancing chemical AI, including a public leaderboard, task generation process, symbolic verification tools, training datasets to both evaluate models but also support reinforcement learning with verifiable rewards.

## 2 RELATED WORK

Research on Large Language Models (LLMs) for chemistry falls into two streams: (1) benchmarking efforts assessing generalist and specialist capabilities, and (2) the development of specialized *chemistry LLMs* via fine-tuning, reinforcement learning (RL), or multimodal architectures.

**Generalist benchmarks and their limits.** Early evaluations prioritized broad coverage using exam-style, multiple-choice questions spanning many fields in chemistry(Sun et al., 2024; Wang et al., 2024a; Huang et al., 2024; Zhang et al., 2024; Wang et al., 2024b; Mirza et al., 2025). While effective for assessing factual recall, such formats risk rewarding pattern matching or memorization over genuine reasoning, and do not test the explicit structural comprehension that underpins chemical properties. To address this, *FrontierScience* (Wang et al., 2025b) recently introduced expert-level, open-ended research tasks. However, its reliance on model-based grading (using advanced LLMs with rubrics) introduces significant evaluation noise. Furthermore, manually curated expert questions are challenging to scale for the systematic and granular diagnosis of model failure modes.

**Benchmarking structural reasoning.** Recent work has shifted toward directly evaluating molecular comprehension. Guo et al. (2024) introduced MolPuzzle to probe structure elucidation from spectra; however, its reliance on textbook-style molecules risks data contamination, while its multimodal nature separates it from pure graph reasoning. Similarly, ChemIQ (Runcie et al., 2025) offers symbolic mostly tasks, but many serve primarily as SMILES parsing checks and have been saturated by base models. Other work has examined how molecular string representation format affects LLM performance on chemical tasks (Baker et al., 2025). A newer wave of benchmarks targets specific subdomains but suffers from data leakage or a narrow scope. FGBench (Liu et al., 2025) evaluates functional-group reasoning but inherits labels directly from MoleculeNet (Wu et al., 2018), a common pretraining corpus. MolErr2Fix (Wu et al., 2025) probes error detection and correction in molecular descriptions but is limited to a single task modality. ChemCoTBench (Li et al., 2025) assesses editing and reaction prediction using USPTO data (Lowe, 2017), again risking contamination, and relies on non-deterministic LLM judges.

Finally, generative benchmarks like TOMG-Bench (Li et al., 2024a) and MEGA (Fernandez et al., 2025) focus on constraint satisfaction and property optimization. While valuable for generation, they do not measure whether a model genuinely understands the underlying structure-property relationships. Crucially, none of these frameworks systematically vary task type, molecular complexity, or representation format to localize precisely *where* reasoning breaks down.

**The model-evaluation gap.** The landscape of chemistry models is diversifying rapidly, spanning instruction-tuned models (Zhang et al., 2024; Yu et al., 2024) and RL-enhanced reasoners (Zhao et al., 2025b; Narayanan et al., 2025). Despite this progress, evaluation remains fragmented. Models are typically assessed on disparate, self-curated datasets for captioning (ChEBI-20), reaction prediction (USPTO), or property prediction (MoleculeNet) (Edwards et al., 2021; Lowe, 2017; Wu et al., 2018). These ad hoc protocols are often tailored to specific models and utilize approximate oracles, undermining meaningful cross-model comparison. Without a unified benchmark that offers symbolic ground truths and controlled complexity axes, disentangling genuine structural reasoning from pattern matching remains an open challenge.

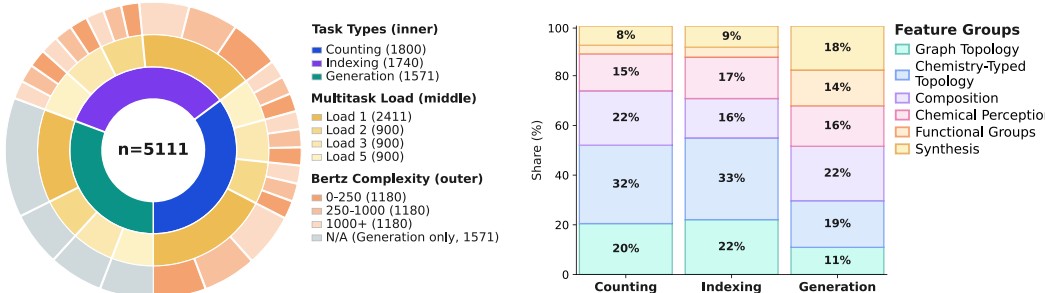

Figure 2: **MOLECULARIQ composition.** (left) Number of questions by task type, multitask load, and Bertz complexity. (right) The percentage of feature types present for counting, indexing, and generation.

## 3 MOLECULARIQ

MOLECULARIQ is a fully symbolically verifiable benchmark for assessing reasoning over molecular structures. It covers three task types – feature counting, index-based attribution, constrained generation – (Section 3.1) covering various symbolically verifiable molecular features across six categories (Section 3.2). Additionally, MOLECULARIQ incorporates three orthogonal complexity axes (Section 3.3): SMILES representation format, molecular complexity, and multitask load. This creates a multi-dimensional capability profile that allows localization of failure modes along bespoke axes and the question types. It is released in two forms (Section 3.4): a static variant, denoted as MOLECULARIQ, containing a total of 5111 questions (Figure 2), and a dynamic variant MOLECULARIQD, designed to address future concerns such as overfitting and saturation.

### 3.1 REASONING TASKS

We define three types of tasks:

- **Feature counting** establishes baseline comprehension of molecular graphs. High performance may arise from shortcuts that bypass true graph reasoning.
- **Index-based attribution** requires models to ground answers in specific atoms/bonds/substructures, closing off spurious solution paths.
- **Constrained generation** frames molecular design as the generation of molecules that satisfy given specifications, a capability essential for practical applications.

For each feature-counting question, except for hydrogen counting and molecular formula, where indexing is not possible, MOLECULARIQ also provides an equivalent index-based question on the same molecule. This allows us to assess whether a model's correct count stems from pattern recognition or genuine reasoning over the molecular graph.

### 3.2 SYMBOLICALLY VERIFIABLE MOLECULAR FEATURES

Our chemical-reasoning tasks target molecular features across six categories: graph topology, chemistry-typed graph topology, composition, chemical perception, functional groups, and synthesis/fragmentation. Below we briefly describe each category; detailed definitions appear in Section A.1. For each feature, we have an RDKit-based symbolic solver (Landrum et al., 2006), enabling both counting and constrained-generation tasks. For all features, except hydrogen counting and molecular formula, we can also determine the indices of the atoms participating in the feature.

**Graph topology** captures purely structural properties of the molecular graph independent of chemical typing. Tasks probe recognition of rings, fused rings, bridgehead atoms, ring size extremes, as well as linear termini and branch points. These test whether a model can parse and reason about the bare connectivity of the molecular skeleton.

**Chemistry-typed graph topology** extends graph topology with chemical annotations. Features to probe include identifying aromatic vs. aliphatic rings, heterocycles, saturation, sp³ carbon hybridization, longest carbon chains, and stereochemical descriptors (R/S centers, E/Z double bonds,

specified vs. unspecified stereochemistry). This family checks whether models integrate chemical typing and stereochemistry into their structural reasoning.

**Composition** focuses on counting and enumerating atomic constituents of molecules. Tasks involve identifying numbers of carbon, hetero, halogen, heavy, or hydrogen atoms, and deriving the molecular formula. These probe basic but essential symbolic operations directly tied to chemical composition.

**Chemical perception** covers substructural and electronic properties commonly used in cheminformatics. Tasks include identifying hydrogen bond donors (HBD) and acceptors (HBA), rotatable bonds, and assigning oxidation states. This family assesses whether models can recognize functional features that underpin molecular properties.

**Functional groups** targets detection of chemically meaningful substructures such as alcohols, amines, carboxylic acids, and other standard functional groups. This probes chemical reasoning at the level of reactivity patterns and transformations.

**Synthesis and fragmentation** encompasses tasks relevant to retrosynthesis and scaffold analysis. It includes BRICS fragmentation, template-based reaction prediction, and Murcko scaffold extraction. These tasks evaluate whether models can reason about molecules in terms of synthetic building blocks and core scaffolds, bridging structural analysis with applications in drug design.

## 3.3 COMPLEXITY AXES

**SMILES representations.** We use SMILES (Weininger, 1988), the most widely adopted molecular string representation, to represent the molecular graphs. Their non-uniqueness and the availability of aromatic vs. kekulized (dearomatized) forms provide an additional, controllable complexity axis. If an LLM truly reasons over structure, canonicalization should be irrelevant; conversely, a drop from canonical to non-canonical SMILES would indicate reliance on memorized canonical patterns. Yu et al. (2024) reported that training the chemistry LLM LlaSMol with canonical SMILES improved performance, and Runcie et al. (2025) observed significant performance drops on several ChemIQ tasks when using randomized (non-canonical) SMILES. Likewise, perfect graph understanding would render kekulization immaterial—though kekulized inputs may be harder because inferring aromaticity adds a reasoning step. To probe both factors, we apply two independent random decisions per molecule, each with 50% probability. Additionally, for a subset of models, we also tested the effect of ring index relabeling on counting and indexing questions involving cyclic molecules.

**Molecular complexity.** Understanding how models perform across different regimes of molecular complexity is essential for identifying the types of molecules on which they succeed and for diagnosing failure modes. As molecules grow larger and more branched, with more rings and functional groups, models must keep track of longer-range structural dependencies and a growing number of relevant features, which can challenge both representation and reasoning. In our benchmark, we adopt a coarse-grained categorization based on the Bertz complexity index (Bertz, 1981), defining three bins: 0–250, 250–1000, and 1000+. Figure A1 illustrates example molecules per complexity bin, and Figure A2 shows that MOLECULARIQ spans a broader complexity range than ChemIQ and ChemCOTBench. The dynamic dataset version allows adjusting these bins for customized evaluation, for example to probe how model performance changes as molecular complexity increases.

**Multitask load dimension.** Chemical reasoning often involves solving several sub-tasks simultaneously, motivating an explicit evaluation along the multitask axis. Many realistic tasks, such as reaction prediction, mechanism recognition, or constraint-driven molecule design, decompose into elementary operations like identifying functional groups or determining implicit hydrogens. Prior work has reported low performance on format conversion tasks, such as SMILES→IUPAC (Runcie et al., 2025) or IUPAC→SMILES (Narayanan et al., 2025), which require accurate coordination of multiple atomic and structural counts. Poor results may stem either from inherent task difficulty or from a broader inability of models to manage multiple chemical sub-tasks. To disentangle these factors, MOLECULARIQ systematically varies the multitask load. We evaluate four regimes: a single-task setting (one count, index, or constraint) and multitask settings with 2, 3, or 5 simultaneous requests.

## 3.4 BENCHMARK CREATION

This section outlines the procedure for constructing MOLECULARIQ. The benchmark consists of sampled molecules with exact, precomputed ground-truth values for the task each molecule was assigned to. We anticipate that future models will be fine-tuned on this task battery. To prevent data leakage, we also provide a pool of training molecules.

**Molecule pool.** We retrieved all single fragment molecules that contained at least one carbon atom from PubChem (Kim et al., 2023). To avoid contamination of the training pool and redundant evaluations, we removed all molecules in *ether0*, *ChemIQ*, *LlaSMol*, and *ChemDFM* evaluation sets. **Training and test pools.** To promote best practices for model development, we partition the initial molecule pool into three sets: a training pool, an *easy* test pool, and a *hard* test pool. The easy pool contains molecules from clusters present in training; the hard pool is sampled from held-out clusters. We cluster molecules using MinHashLSH (Broder, 1997; Indyk & Motwani, 1998; Leskovec et al., 2020) over ECFP fingerprints (2, 512) (Rogers & Hahn, 2010) with a 0.7 Tanimoto similarity threshold. The resulting splits comprise 1.3M training molecules and 1.0M molecules each in the easy and hard test sets.

MOLECULARIQ is our primary benchmark. It comprises 849 unique molecules drawn from the hard test pool and a total of 5,111 questions distributed across the complexity axes. For all molecules in the hard test pool, we compute the ground-truth values for the molecular features and Bertz complexities using RDKit. Then, each question is formed by sampling a (task, question template, feature(s), molecule) tuple. The sampled molecule serves as part of the input for counting and indexing tasks, and as a feasibility reference for constrained generation to ensure the constraints are satisfiable. Benchmark samples are produced via a weighted scheme that balances reasoning tier, multitask load, molecular complexity, and ground-truth value distributions (see Section A.2).

MOLECULARIQD. We provide a framework to dynamically adjust and scale the dataset to community needs. It supports sampling new molecules from the defined pools and efficiently computing ground truths for any RDKit-readable SMILES. Potential uses include: (i) creating validation sets from the easy test pool for model training; (ii) refreshing or expanding MOLECULARIQ when models saturate or overfit; and (iii) constructing focused evaluation sets for specific subfields (e.g., natural products).

## 4 EVALUATION AND LEADERBOARD

**Language Model Evaluation Harness integration.** We integrate MOLECULARIQ into the `lm-evaluation-harness` framework (Gao et al., 2024), enabling standardized evaluation across both local and API-served models. Each of the evaluated models (Section B.1) uses tailored configurations with model-specific sampling parameters, preprocessing functions, and extraction methods to ensure fair comparison while optimizing for each model's strengths (Section B.2).

**Symbolic extraction.** Fair benchmark comparison requires reliable answer extraction that focuses on semantic content rather than surface formatting. Recent work has highlighted how weak extraction procedures can artificially inflate or deflate model performance, with models potentially learning format compliance rather than genuine capabilities (Chandak et al., 2025; Shao et al., 2025; Gudibande et al., 2024). We address this through: (a) hierarchical extraction procedures that gracefully handle diverse output formats, and (b) key-specific matching with canonical normalization that decouples format compliance from chemical accuracy while preserving property-level granularity for fine-grained performance analysis.

**Scoring.** Each task is scored with a binary symbolic verifier; per-instance accuracy is the mean over three independent rollouts to account for stochastic generation (Wang et al., 2022; Brown et al., 2020). Reported scores therefore lie in $[0, 1]$ (higher is better). For a subset of models, we also performed eight roll-outs, and for those models, the differences were negligible compared to three rollouts (Table C11).

**Metrics.** We report (i) overall *accuracy* averaged across all benchmark instances and (ii) granular breakdowns by reasoning task, multitask load, molecular complexity, molecular-feature category, and SMILES format, enabling systematic assessment of model strengths and weaknesses (Srivastava et al., 2023). For some analyses, we use a binary *success* metric: an instance counts as successful

Table 1: Results on the MOLECULARIQ benchmark for top-performing models. We compare the top 3 chemistry-specialized LLMs and top 10 generalist LLMs, reporting overall accuracy and accuracy across task families (in %). Models are sorted by overall accuracy. Error bars indicate standard errors across instances. Model size refers to parameter count; R and C indicate reasoning and chemistry models.

| | Size | R | C | Overall | Task family | | |
| | | | | | Counting | Indexing | Generation |
|---|---|---|---|---|---|---|---|
| *Chemistry LLMs* | | | | | | | |
| TxGemma-27B | 27B | | ✓ | $5.0 \pm 0.2$ | $7.0 \pm 0.5$ | $1.8 \pm 0.3$ | $6.2 \pm 0.5$ |
| Ether0 | 24B | ✓ | ✓ | $6.5 \pm 0.3$ | $3.2 \pm 0.3$ | $0.1 \pm 0.0$ | $17.5 \pm 0.8$ |
| ChemDFM-R-14B | 14B | ✓ | ✓ | $8.7 \pm 0.4$ | $12.9 \pm 0.8$ | $2.8 \pm 0.4$ | $10.5 \pm 0.8$ |
| *Generalist LLMs* | | | | | | | |
| GLM-4.6 | 355B (A32B) | ✓ | | $16.2 \pm 0.4$ | $15.9 \pm 0.7$ | $11.3 \pm 0.6$ | $22.0 \pm 0.9$ |
| Qwen-3 30B | 30B (A3B) | ✓ | | $22.7 \pm 0.5$ | $23.0 \pm 0.9$ | $16.5 \pm 0.8$ | $29.4 \pm 1.0$ |
| SEED-OSS | 36B | ✓ | | $24.3 \pm 0.5$ | $26.8 \pm 0.9$ | $25.6 \pm 0.9$ | $20.0 \pm 0.9$ |
| GPT-OSS 120B (Med) | 120B (A5B) | ✓ | | $26.5 \pm 0.5$ | $29.1 \pm 0.9$ | $22.3 \pm 0.8$ | $28.0 \pm 1.0$ |
| GPT-OSS 20B (Med) | 20B (A4B) | ✓ | | $27.2 \pm 0.5$ | $27.9 \pm 0.9$ | $22.2 \pm 0.8$ | $32.1 \pm 1.0$ |
| Qwen-3 Next 80B | 80B (A3B) | ✓ | | $32.3 \pm 0.6$ | $30.7 \pm 1.0$ | $26.9 \pm 0.9$ | $40.0 \pm 1.1$ |
| GPT-OSS 20B (High) | 20B (A4B) | ✓ | | $35.8 \pm 0.6$ | $39.1 \pm 1.0$ | $32.8 \pm 1.0$ | $35.2 \pm 1.0$ |
| DeepSeek-R1-0528 | 671B (A37B) | ✓ | | $37.1 \pm 0.6$ | $36.4 \pm 1.0$ | $30.2 \pm 0.9$ | $45.5 \pm 1.0$ |
| Qwen-3 235B | 235B (A22B) | ✓ | | $39.2 \pm 0.6$ | $37.1 \pm 1.0$ | $34.5 \pm 1.0$ | $46.7 \pm 1.1$ |
| GPT-OSS 120B (High) | 120B (A5B) | ✓ | | $\mathbf{47.5 \pm 0.6}$ | $\mathbf{46.8 \pm 1.1}$ | $\mathbf{42.5 \pm 1.1}$ | $\mathbf{53.7 \pm 1.1}$ |

if at least two of the three rollouts are correct, and the *success rate* is the percentage of successful instances. Additionally, to evaluate whether failures are due to semantic errors or malformatted outputs, we measure *type-validity rate*, the percentage of outputs satisfying expected syntactic constraints, for each task category (integers for counting, integer lists for indexing, valid SMILES for generation).

**Living benchmark.** The full leaderboard will be hosted online: https://huggingface.co/spaces/ml-jku/molecularIQ_leaderboard. Following the model of the Open LLM Leaderboard (Fourrier et al., 2024), we will accept submissions of new chemistry LLMs (open weights or free API access). Authors provide (i) an lm-evaluation-harness configuration file, (ii) optional model-specific preprocessing and extraction functions, and (iii) a brief description of training methodology and data sources.

## 5 RESULTS AND DISCUSSION

We evaluated 38 open-weight LLMs on MOLECULARIQ using the protocol in Section 4: 27 general-purpose and 11 chemistry-specialized models. See Table B1 for model details. In this section, we focus on the analysis of the top-10 models and key findings, focusing on the overall results and counting tasks. Table 1 reports the overall score and per–reasoning-task scores for the top models, and Figure 3 illustrates the performance profile of these models on the counting tasks. In Section C, we provide results for all evaluated models for all task types and further break down performance by multitask load, molecular complexity, molecular–feature category, and SMILES format, and extended failure mode analysis.

**Large state-of-the-art LLMs with high reasoning budgets lead molecular-structure reasoning.** GPT-OSS 120B (High) is the best-performing model overall and across most subcategories in our multi-dimensional complexity profiling. Nine out of ten top models belong to the same family of modern mixture-of-experts (MoE) reasoning models, with SEED-OSS being the non-MoE model. Overall, this aligns with trends in general language modeling and reasoning benchmarks, where these models are also state-of-the-art (e.g., LiveBench (White et al., 2025)). However, on LiveBench, Qwen-3, DeepSeek-R1, and GLM-4.6 models significantly outperform GPT-OSS, whereas on our molecular-structure tasks, we observe the inverse, with GLM-4.6 in particular performing notice-

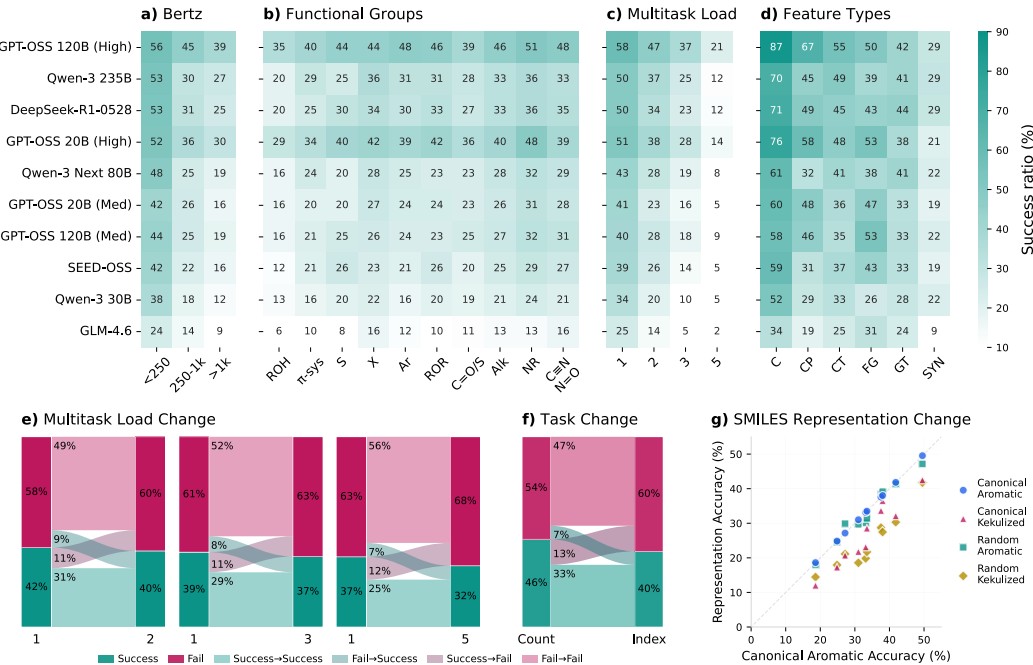

Figure 3: **Performance profile of top 10 models for counting tasks. (a–d)** Heatmaps of average success ratio per model (%) by **(a)** Bertz complexity of the molecule, **(b)** functional-group families (ROH: hydroxyls, π-sys: π-systems, S: sulfurs, Hal: halides, Ar: aromatics, ROR: ethers, C=O/S: (thio)carbonyls, Alk: alkyls, NR: amines, C≡N/N=O: nitriles/nitros), **(c)** multitask load, and **(d)** molecular feature groups (C: composition, CP: chemical perception, CT: chemistry-typed topology, FG: functional groups, SYN: synthesis). **(e–f)** Success/failure transitions averaged across top models for matched questions **(e)** as additional sub-tasks are added to the same prompt, and **(f)** when moving from counting to indexing variants of the same underlying property, probing whether correct counts are grounded in explicit substructure identification. **(g)** Sensitivity to input representation, comparing accuracy between different SMILES representations.

ably worse than the other models. We also find that a higher reasoning budget generally improves performance (Figures C7 and C8); for GPT-OSS, the budget-induced performance gaps within a given model are larger than the gaps between different model sizes.

**Constrained generation exposes model limitations under compositional demands.** Most top-10 models score highest on constrained generation (Table 1), yet this strength does not generalize. Accuracy drops sharply as constraint-set prevalence (i.e., how frequently the requested features co-occur in PubChem molecules) decreases (Figure C1) and once three or more constraints are requested (Figure C12). These patterns indicate that models lack genuine compositional reasoning, as success on common constraints does not transfer to rarer combinations.

**Multitask load makes questions harder overall, but can aid solving individual structural sub-tasks.** Figures 3(a) and (c) show that counting task performance declines with both Bertz complexity and multitask load, but the effect of multitask load is substantially larger. Similar patterns persist across all models and indexing (Tables C2–C3). However, in the multitask case with $n$ subtasks, an answer counts as correct only if all $n$ subtasks are answered correctly. Let $p_{single}$ denote the single task success rate. If multitask load had no additional effect, the expected success rate at load $n$ would be $p_{single}^n$. For all top models, the observed $n$-task success rates exceed this baseline, indicating that multitask prompting actually helps models solve the underlying structural reasoning subtasks, even though the full questions become harder. This is consistent with Figure 3(e), which shows that the solvability of most properties is largely stable across task loads, and that, for slightly more properties, adding subtasks increases their solvability rather than decreasing it.

**Top models transfer from counting to indexing with only modest performance loss.** For the top models, the performance gap between feature *counting* and *indexing* is relatively small, with a

decrease of ∼5–30% depending on the model (Table 1), suggesting that most correct counts arise from correctly locating the underlying substructures, i.e., genuine graph-based reasoning, instead of pattern recognition or memorized feature statistics. This is consistent with Figures 3(e) and C11, which show that the solvability of most molecular features stays the same for the counting and indexing questions (∼80%), and that, for slightly more properties, changing the task from counting to indexing increases their solvability rather than decreasing it.

**SMILES perturbations reveal reliance on canonicalization and token patterns.** We test whether models rely on canonical SMILES conventions by evaluating counting and indexing under randomized, kekulized, and ring-enumerated representations. Performance drops consistently across all perturbations and all top-10 models (Figures 3(g), C5, C6). These sensitivity patterns suggest a strong reliance on canonical token patterns, ring-closure conventions, and aromatic notations.

**Feature- and functional-group–level analyses expose uneven ability profiles.** Figure 3(d) reveals uneven competence to answer questions about different feature types. Composition-related features are easiest (70–90% for top models), while synthesis/fragmentation remains difficult (Tables C4–C7). Figure 3(b) shows disparities in handling molecules with different functional groups: aromatic and alkyl patterns are handled well, but organosulfur and oxidized nitrogen motifs (C≡N/N=O) yield low success rates (Figure C10 (b)).

**Failures reflect missing reasoning, not extraction artifacts or invalid outputs.** One concern for symbolically evaluated benchmarks is that scores might be dominated by extraction errors rather than genuine reasoning failures. Two analyses argue against this. First, Table C13 and Figure C13 show that type validity is high, often 80–90% for top models, while accuracy is substantially lower, indicating that most failures are *semantically* wrong answers, not malformed outputs. Second, Figure C8 reveals that incorrect answers are on average significantly longer than correct ones across all top-10 models, suggesting verbose chain-of-thought correlates with confusion. To understand *why* errors occur, we analyze 300 question – trace pairs on which no model succeeds, rating chains-of-thought along chemistry and reasoning dimensions (Section C.2.6, Figure C14 & C15). The chemistry failure-mode heatmap shows that failing traces handle basic SMILES parsing and ring topology but break down on functional-group recognition, property attribution, and stereochemistry. The reasoning heatmap indicates adequate task fidelity and method selection but weak constraint tracking, quantitative precision, and error awareness. We provide example failures from different models along with their judged scores in Appendix C.2.7. These illustrate common mistakes such as relying on SMILES lowercase conventions for aromaticity or using canonical @/@@ characters instead of applying CIP rules for stereochemistry assignment. Together with more than 1000 universally failed questions (Figure C9), these findings confirm that MOLECULARIQ probes genuine weaknesses in chemical reasoning, not output formatting.

**Naïve chemistry fine-tuning systematically hurts.** Recent open-weight general-purpose LLMs surpass chemistry-specialized models on MOLECULARIQ, reflecting the rapid gains in generic reasoning. More strikingly, chemistry fine-tuning on task sets that differ from ours, apart from limited overlap (e.g., SMILES→formula for several models; constrained generation for ether0), does not translate into better structure reasoning and systematically degrades performance relative to the base models (Table C12). Only ChemDFM-R shows a minor improvement over its base model. This suggests that narrow, task-specific instruction tuning can miscalibrate reasoning or overfit to format artifacts, yielding weaker transfer to our symbolically evaluated tasks. On average, chemistry-tuned models have an 18 percentage point lower type validity rate than their base models (median 11 points), indicating that the chemistry tuning has degraded their general language modeling capabilities and their ability to follow formatting instructions by overfitting them to narrow, task-specific output patterns (Table C13).

## 6 LIMITATIONS AND OUTLOOK

**Narrow feature suite.** By design, MOLECULARIQ focuses exclusively on symbolically verifiable tasks. This ensures exact ground truth but excludes important classes of problems, such as property prediction (e.g., solubility, bioactivity) or reaction outcomes, where exact symbolic solvers are unavailable. As a result, our benchmark cannot yet cover the full spectrum of tasks relevant to molecular design and drug discovery. Future work may combine symbolic tasks with trusted numerical approximations (e.g., QM-based solvers) to broaden task coverage while maintaining verifiability.

**Restriction to 2D structure.** Our benchmark currently evaluates models exclusively based on 2D molecular representations (SMILES). This choice reflects the present state of chemistry-capable LLMs: SMILES, as compact string encodings of molecular graphs, remain the primary and most intuitive input format for text-based models, whereas 3D representations would require multimodal architectures that are only beginning to emerge. SMILES, therefore, remains the most practical and widely supported interface for assessing molecular reasoning capabilities in today's models. Nevertheless, we acknowledge that many chemically relevant phenomena depend on 3D information, including stereochemistry, conformational preferences, and spatial constraints.

**Focus on single molecules represented with a single modality.** All tasks in MOLECULARIQ operate on individual molecules. However, many critical applications in chemistry involve reasoning across sets of molecules, such as reaction prediction, retrosynthesis planning, or relative ranking of candidate designs. Extending the benchmark to multi-molecule tasks would enable evaluation of these higher-order reasoning capabilities. Our benchmark reflects current practice, where chemistry models rely mainly on SMILES and natural language. Future models will likely integrate multiple modalities to capture a broader range of chemical information.

**Outlook.** Iterative evaluation of state-of-the-art chemical reasoning capabilities is a cornerstone for sustainable model development. We envision MOLECULARIQ playing a central role in future model evaluation, evolving into a living benchmark. Moreover, we anticipate that MOLECULARIQD will allow the community to adapt evaluation sets to emerging needs, while its symbolic solver can serve as an efficient reward model for reinforcement learning.

## ACKNOWLEDGEMENTS

The ELLIS Unit Linz, the LIT AI Lab, and the Institute for Machine Learning are supported by the Federal State of Upper Austria. We thank the projects FWF AIRI FG 9-N (10.55776/FG9), AI4GreenHeatingGrids (FFG- 899943), Stars4Waters (HORIZON-CL6-2021-CLIMATE-01-01), and FWF Bilateral Artificial Intelligence 10.55776/COE12. We thank NXAI GmbH, Audi AG, Silicon Austria Labs (SAL), Merck Healthcare KGaA, GLS (Univ. Waterloo), TÜV Holding GmbH, Software Competence Center Hagenberg GmbH, dSPACE GmbH, TRUMPF SE + Co. KG. We thank Niklas Schmidinger and Thomas Schmied for encouraging us to share our chemical reasoning benchmark and for highlighting its relevance to the LLM community.

## ETHICS STATEMENT

This work introduces a benchmark for evaluating chemical reasoning in large language models. The benchmark is based solely on publicly available molecular structures from PubChem and symbolically verifiable tasks, avoiding the use of sensitive or proprietary data. Our tasks are designed to measure reasoning ability rather than to suggest new molecules with biological activity, and thus do not constitute drug discovery or generate compounds of concern. We caution that improvements on our benchmark should not be interpreted as evidence of safety or efficacy in real-world chemical applications. The benchmark is intended to advance transparency, reproducibility, and rigorous evaluation in AI for chemistry while mitigating risks of data leakage and misuse.

## REPRODUCIBILITY STATEMENT

The exact benchmark version used in this paper is publicly available on the Hugging Face Hub as https://huggingface.co/datasets/ml-jku/moleculariq-v0.0, and the training pool is provided separately as https://huggingface.co/datasets/ml-jku/moleculariq-trainPool. The public leaderboard is hosted at https://huggingface.co/spaces/ml-jku/molecularIQ_leaderboard and is evaluated on a non-public successor version of MolecularIQ to maintain benchmark integrity. A central overview repository linking all MolecularIQ components, including symbolic RDKit-based solvers, task generators, dataset construction, and evaluation code, is available at https://github.com/ml-jku/moleculariq.

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

# MolecularIQ - Appendix

# A  MOLECULARIQ DATASET DETAILS

This section provides extended information on molecular features included (Section A.1), the dataset creation procedure (Section A.2), and presents examples of the questions included (Section A.3).

## A.1  MOLECULAR FEATURES

MOLECULARIQ covers 30 different molecular features across the six groupings introduced in Section 3.2. Table A1 summarizes the categorization, and in the following paragraphs, we give a short description of each feature.

Table A1: Overview of tasks in MOLECULARIQ

| Feature category | Name |
| --- | --- |
| Graph topology | Rings |
| Graph topology | Fused rings |
| Graph topology | Bridgehead atoms |
| Graph topology | Smallest or largest ring size |
| Graph topology | Chain termini |
| Graph topology | Branch points |
| Chemistry-typed graph topology | Aromatic rings |
| Chemistry-typed graph topology | Aliphatic rings |
| Chemistry-typed graph topology | Heterocycles |
| Chemistry-typed graph topology | Saturated rings |
| Chemistry-typed graph topology | $sp^3$ carbons |
| Chemistry-typed graph topology | Longest carbon chain |
| Chemistry-typed graph topology | R or S stereocenter |
| Chemistry-typed graph topology | Unspecified stereocenter |
| Chemistry-typed graph topology | E or Z stereochemistry double bond |
| Chemistry-typed graph topology | Stereochemistry unspecified double bond |
| Chemistry-typed graph topology | Stereocenters |
| Composition | Carbon atoms |
| Composition | Hetero atoms |
| Composition | Halogen atoms |
| Composition | Heavy atoms |
| Composition | Hydrogen atoms |
| Composition | Molecular formula |
| Chemical perception | HBA |
| Chemical perception | HBD |
| Chemical perception | Rotatable bonds |
| Chemical perception | Oxidation state |
| Functional groups | Functional groups |
| Synthesis and fragmentation | BRICS decomposition |
| Synthesis and fragmentation | Template based reaction prediction |
| Synthesis and fragmentation | Murcko scaffold |

### GRAPH TOPOLOGY

**Rings**. Ring detection is a fundamental aspect of molecular structure analysis, as cyclic motifs are present in the majority of biologically active compounds. This task requires identifying whether a molecule contains ring structures and determining their number. While trivial for cheminformatics toolkits, it challenges LLMs to correctly parse connectivity patterns beyond simple linear chains. Performance on this task provides a baseline indicator of whether models can capture cyclicity, which underpins aromaticity, rigidity, and scaffold formation in medicinal chemistry.

**Fused rings**. Fused or condensed rings occur when two or more rings share one or more atoms, producing scaffolds such as naphthalene or steroid backbones. Detecting fused rings requires models not only to identify individual cycles but also to recognize shared boundaries, which adds a layer of structural complexity. Correct reasoning here is crucial for capturing polycyclic scaffolds that dominate many drug classes. Errors indicate that models may treat rings independently without appreciating their overlap and fusion.

**Bridgehead atoms**. Bridgehead atoms are special positions where two or more rings intersect, often in bicyclic or polycyclic systems. These atoms influence molecular strain, conformational rigidity, and reactivity. Identifying them requires models to trace cycles through shared atoms and to distinguish simple substitution from ring–ring junctions. Success on this task indicates structural reasoning beyond mere ring counting, while failure highlights limitations in global graph understanding.

**Smallest or largest ring size**. Ring size strongly influences chemical reactivity, strain, and biological activity. This task challenges models to determine the size of either the smallest or the largest ring in a molecule, requiring explicit counting of atoms within cycles. Such reasoning tests whether models can not only detect rings but also evaluate their dimensions, which are essential for understanding macrocycles or strained small rings. Performance degradation here would suggest that models rely on heuristics instead of robust ring-perception logic.

**Chain termini**. Chain termini are the end points of a linear carbon or heteroatom chain, typically corresponding to substituents or free valences. Detecting termini requires parsing connectivity to identify atoms of degree one. While chemically straightforward, this task probes whether models can recognize basic graph-theoretic features such as leaf nodes, which are important for subsequent reasoning about branching, chain length, and attachment sites. Failures indicate difficulty in translating molecular strings into graph structures.

**Branch points**. Branch points occur where a linear chain splits into two or more substituents, creating tree-like structures. These positions affect molecular shape, flexibility, and lipophilicity, which in turn influence pharmacokinetics. Detecting branch points requires identifying atoms of degree three or more in the molecular graph. Correct reasoning here demonstrates that a model can capture local topology rather than treating molecules as simple chains.

Chemistry-typed graph topology

**Aromatic rings**. Aromatic systems are rings with delocalized $\pi$-electrons, following Hückel's rule (Hückel, 1931). They play a central role in stability, reactivity, and biological activity. Identifying aromatic rings is more complex than detecting generic cycles, since resonance and hybridization must be considered. This task, therefore, probes whether a model understands chemical typing beyond pure graph topology.

**Aliphatic rings**. Aliphatic rings are saturated, non-aromatic cycles such as cyclohexane or cyclopentane. While structurally simple, they contribute unique conformational preferences and steric constraints. Distinguishing aliphatic from aromatic rings requires reasoning about bond order and hybridization, not just connectivity. Success on this task indicates a model's ability to classify rings by chemical type rather than by topological presence alone.

**Heterocycles**. Heterocycles contain at least one non-carbon atom in the ring, such as nitrogen in pyridine or oxygen in furan. These motifs dominate pharmaceuticals, dyes, and agrochemicals due to their tunable electronic properties. Detecting heterocycles requires models to distinguish atom types inside rings and recognize their chemical implications. This task thus evaluates both structural perception and atom-level classification.

**Saturated rings**. Saturated rings are cycles with only single bonds, often flexible and conformationally rich. They differ strongly from aromatic or unsaturated rings in chemical behavior and reactivity. Identifying them requires counting bond types within cycles and ensuring that no unsaturation is present. Accurate reasoning here shows that a model can integrate bond order information with ring detection.

**sp$^3$ carbons**. Carbons in sp$^3$ hybridization are bonded to four atoms via single bonds, leading to a tetrahedral geometry. The proportion of sp$^3$ carbons is a well-known descriptor of drug-likeness and scaffold diversity. Counting them requires distinguishing hybridization states based on connectivity and bond order. This task highlights whether a model can map molecular graphs to stereoelectronic properties.

**Longest carbon chain**. The length of the longest continuous carbon chain is fundamental in nomenclature and structural description. It often defines the molecule's main scaffold in IUPAC naming. dentifying it requires global graph traversal to find maximum-length paths, testing whether models can perform non-trivial structural reasoning beyond local atom counting.

**R or S stereocenter**. Chirality is a key determinant of biological activity, with enantiomers often having drastically different pharmacological profiles. Assigning R/S configuration requires applying the Cahn–Ingold–Prelog rules (Cahn et al., 1966) to rank substituents and determine stereochemical orientation. This task is demanding because it requires integrating atom identity, connectivity, and three-dimensional orientation. Strong performance indicates sophisticated reasoning about stereochemistry rather than two-dimensional connectivity alone.

**Unspecified stereocenter**. In many SMILES strings or molecular depictions, stereochemistry is left undefined even when a chiral center exists. Detecting such unspecified stereocenters is important for identifying ambiguity in data curation and preventing misinterpretation of molecular activity. Models must therefore distinguish between the presence of chirality and the explicit encoding of stereochemistry. This task tests awareness of missing information, not just positive structural features.

**E or Z stereochemistry double bond**. Double bonds with restricted rotation can give rise to geometric isomers, classified as E (opposite) or Z (together). Correct assignment requires recognizing substituent priorities on both carbons and applying geometric rules. This property has significant effects on molecular conformation and bioactivity. The task evaluates whether a model can interpret bond stereochemistry beyond connectivity.

**Stereochemistry unspecified double bond**. In some structures, double bonds are present without explicit E/Z specification. Identifying such cases is crucial for flagging ambiguous molecular data. This requires detecting when stereochemical information is missing, rather than misclassifying it as a defined configuration. The task complements stereochemistry classification tasks by emphasizing uncertainty detection.

**Stereocenters**. Stereocenters are atoms (often carbons) that give rise to stereoisomerism by connecting to distinct substituents. Counting them provides insight into molecular complexity and the potential number of stereoisomers. Models must integrate atom identity, connectivity, and substituent uniqueness to succeed. This task probes holistic reasoning about stereogenicity across the entire molecule.

COMPOSITION

**Carbon atoms**. Simply counting carbons may appear trivial, but it provides a baseline check of whether a model parses molecular graphs correctly. Carbon count influences molecular weight, scaffold size, and lipophilicity. Models must map SMILES tokens to atom identity and correctly handle ring closures and branching. Errors here indicate failure in basic atom-level parsing.

**Hetero atoms**. Heteroatoms such as nitrogen, oxygen, and sulfur govern hydrogen bonding, polarity, and reactivity. Counting them is critical for estimating drug-likeness and physical properties. The task requires distinguishing carbons and hydrogens from all other elements in the structure. This tests whether models encode chemical types accurately and can generalize across diverse element sets.

**Halogen atoms**. Halogens (F, Cl, Br, I, At) are common substituents in medicinal chemistry, modulating lipophilicity, metabolic stability, and binding affinity. Detecting and counting them requires element-specific recognition. While simple for cheminformatics libraries, this task challenges LLMs to generalize symbol recognition across less frequent elements. Strong performance indicates robust chemical token handling.

**Heavy atoms**. Heavy atoms are defined as all non-hydrogen atoms in the molecule. This count correlates with molecular complexity and is often used as a simple descriptor for dataset stratification. The task is straightforward but requires models to systematically distinguish hydrogens from all other atom types. Failures reveal gaps in even the most basic structural parsing.

**Hydrogen atoms**. Hydrogen count is essential for deriving molecular formulae and understanding saturation. While hydrogens are often implicit in SMILES, inferring their correct number requires applying valence rules. This makes the task more challenging than simply reading atom tokens. Success demonstrates that models can handle implicit chemical information, not just explicit tokens.

**Molecular formula**. The molecular formula condenses the atom counts of all elements into a canonical string, such as $C_6H_6$. Deriving it requires accurate atom counting across the molecule, includ-

ing implicit hydrogens. This is a global summarization task that integrates several simpler counting tasks. It tests whether models can consolidate structural information into a chemically valid representation.

### CHEMICAL PERCEPTION

**Hydrogen bond acceptors (HBA)**. HBAs are atoms with lone pairs (commonly oxygen or nitrogen) capable of accepting hydrogen bonds. They determine solubility, binding interactions, and pharmacokinetics. Identifying HBAs requires applying chemical perception rules, not just atom types, since context (e.g., resonance, hybridization) matters. This task probes whether models encode functional group semantics.

**Hydrogen bond donors (HBD)**. HBDs are groups (e.g., OH, NH) that provide a hydrogen atom for hydrogen bonding. They are central in protein–ligand recognition and solubility. Counting HBDs requires models to parse both atom type and bonding pattern. Accurate predictions reflect chemical reasoning about donor functionality rather than raw atom identity.

**Rotatable bonds**. Rotatable bonds provide conformational flexibility, which influences molecular dynamics and bioavailability. Identifying them requires excluding bonds in rings and terminal bonds while recognizing non-rigid single bonds. This task goes beyond simple bond counting, demanding perception of conformational constraints. It evaluates whether models can integrate structural context into bond classification.

**Oxidation state**. The oxidation state of an atom reflects the distribution of electrons in bonds and is critical for predicting redox behavior. Assigning it requires applying electron-counting rules consistently across a molecule. This task is symbolically verifiable but conceptually demanding, since it combines atom types, bonding patterns, and electronegativity rules. Performance here indicates whether models can implement algorithmic chemical reasoning.

### FUNCTIONAL GROUPS

**Functional groups**. Functional groups define classes of reactivity, such as alcohols, amines, and carbonyls. Identifying them requires pattern recognition across atom types, bond orders, and connectivity motifs. This task encapsulates the chemist's notion of "chemical perception," linking structure to reactivity. It tests whether models can map from subgraph motifs to abstract functional categories.

### SYNTHESIS AND FRAGMENTATION

**BRICS decomposition**. BRICS (Breaking of Retrosynthetically Interesting Chemical Substructures (Degen et al., 2008)) is a rule-based fragmentation method that splits molecules into chemically meaningful building blocks. Performing this decomposition requires applying substitution rules and graph-partitioning logic. The task challenges models to reproduce a widely used cheminformatics procedure. Success indicates capability for rule-driven reasoning over chemical graphs.

**Template-based reaction prediction**. Template-based methods encode chemical transformations as reaction rules (encoded as SMIRKS), which can be applied to reactants to generate products. Each template specifies a substructure match and an atom-mapped rewrite; the model must (i) localize the reactive site(s) in the input molecule and (ii) apply the transformation to yield the product. This setup mirrors how rule libraries are used in medicinal chemistry and CASP, emphasizing structural reasoning and atom mapping rather than surface-level classification. To avoid condition-dependent or stoichiometric ambiguities, we restrict the benchmark to common, unimolecular, single-step reactions with deterministic $1 \rightarrow 1$ mappings (e.g., oxidations, reductions, substitutions, additions, eliminations, aromatic substitutions, rearrangements, protection/deprotection, ring openings/cleavages). These constraints provide unambiguous ground-truth products and isolate a model's ability to understand and manipulate molecular structure.

**Murcko scaffold**. The Murcko scaffold (Bemis & Murcko, 1996) is defined as the union of all ring systems and linkers, with side chains removed. It represents the core structure underlying a molecule's chemical series. Extracting it requires identifying which atoms belong to the scaffold versus substituents. This task tests whether models can abstract molecular graphs into canonical cores, a key step in medicinal chemistry analysis.

## A.2 SAMPLING AND TASK CONSTRUCTION

**Corpus preparation.** Starting from a pool of molecules, we filter them to have between 5-50 heavy atoms, SMILES strings shorter than 100 characters, and valid (parseable) SMILES representations, ensuring the benchmark focuses on small molecules of reasonable complexity while avoiding trivial small molecules or overly complex macromolecules that could skew property distributions. Further, we calculate target properties of these compounds using our implemented solver functions. To ensure representative coverage across molecular complexity, we stratify our dataset $\mathcal{D}$ of molecules into three Bertz complexity bins (Bertz, 1981): $[0, 250)$, $[250, 1000)$, and $[1000, \infty)$. Figure A1 illustrates a few example molecules from each complexity bin. From the property table indexed by SMILES, we construct three task families—*count*, *index*, and *generation*—each in single- and multi-feature variants. Sampling employs inverse-frequency weighting to promote diversity and prevent over-representation of common cases.

**Single-feature count and index task generation.** For each complexity bin and feature, we generate 10 examples per bin using an inverse-frequency sampling scheme designed to maximize diversity. For each molecule, we calculate property frequency, invert it, and down-weight zero cases by 0.5 to obtain meaningful examples that avoid trivial "absent" cases.

Each selected molecule-feature pair generates a *count* question. If the property has a corresponding index mapping, we additionally create a paired *index* question for the same molecule. This allows for full traceability of model capabilities across different task types, while maintaining the same molecule.

**Multi-feature count and index task construction.** To test multi-step complex reasoning, we create tasks requiring simultaneous analysis of $n \in \{2, 3, 5\}$ properties on the same molecule. Starting from molecules already used in single-property tasks, we:

1. Begin with properties already associated with that molecule using doubly inverse-frequency sampling with zero bias (as in single-property tasks)
2. Add additional properties using inverse-frequency logic to maintain diversity
3. Apply plausibility filters: reject samples with more than one zero-valued property to avoid trivial cases
4. Target 100 items per complexity bin and per property count $n$

We maintain the same pairing logic between count and index tasks as before. This approach ensures multi-property tasks build naturally from single-property foundations while maintaining chemical plausibility. It also allows us to trace whether a model can handle multitask load meaningfully, ensuring we don't lose skills already demonstrated in simpler tasks.

**Single-constraint generation.** We derive exact equality constraints (property, value) from single-count tasks, supplemented by additional constraint tasks not present in the count case. We aim to generate the same number of single-constraint tasks as count tasks. However, finding enough examples for certain count properties is infeasible, so we supplement with the unique property set. If that is smaller than the target value, we take all; otherwise, we apply inverse-frequency sampling.

**Multi-constraint generation sampling** Our multi-constraint generation tasks require molecules to simultaneously satisfy $k \in \{2, 3, 5\}$ property constraints, creating increasingly complex reasoning challenges. The sampling procedure addresses two key challenges: ensuring chemical satisfiability while maintaining appropriate task difficulty, and preserving lineage relationships to parent tasks.

**Constraint candidate construction.** For each molecule, we construct a prioritized pool of candidate constraints following a hierarchical scheme: (1) *seed constraints* from single-constraint parent tasks (mandatory for lineage), (2) properties from multi-property parent tasks, (3) additional molecular count properties, and (4) special properties including molecular formulas, Murcko scaffolds, and functional groups. This hierarchy ensures that multi-constraint tasks build upon simpler prerequisites while introducing novel property combinations.

**Information-theoretic scoring.** We score $k$-constraint combinations based on their collective information gain and rarity. For a combination $\mathcal{C} = \{c_1, ..., c_k\}$, we compute the cumulative reduction

Complexity bin: 0-250

172     180     199     206     208

219     235     237     245     246

Complexity bin: 250-1000

335     434     443     500     508

577     686     760     772     954

Complexity bin: 1000-inf

1031     1097     1110     1214     1215

1414     1454     1611     1644     2170

Figure A1: Example molecules from each Bertz complexity bin, labeled with their Bertz complexity.

ratio:

$$R(\mathcal{C}) = \prod_{i=2}^{k} \frac{|\mathcal{M}_{c_1} \cap ... \cap \mathcal{M}_{c_{i-1}}|}{|\mathcal{M}_{c_1} \cap ... \cap \mathcal{M}_{c_i}|} \tag{1}$$

where $\mathcal{M}_{c_i}$ denotes the set of molecules satisfying constraint $c_i$. Constraints are randomly ordered before computing per-step reduction ratios. This measures how much each additional constraint reduces the satisfying molecule set. We require each non-zero constraint to achieve a reduction ratio $\geq 1.1$ (reducing the satisfying set by at least 10%), while zero-valued constraints must achieve $\geq 2.0$ (50% reduction) to justify their inclusion. Combinations must satisfy between 10 and 1,200 molecules for $k = 2$, with relaxed bounds of 5-1,500 molecules for $k = 5$, ensuring non-trivial but achievable tasks.

**Diversity-aware selection.** The final selection employs Monte Carlo sampling with rejection criteria ensuring: (1) at least one seed constraint for task lineage, (2) no redundant property pairs (e.g., molecular formula with elemental counts), (3) at most one zero-valued constraint meeting the stricter reduction threshold, and (4) adherence to $k$-specific satisfiability bounds. Tasks achieving molecule counts within ideal ranges (60-220 for $k = 2$, 20-120 for $k = 5$) receive scoring bonuses, while combinations with prelevance rates below 5% are prioritized for their rarity.

**Solver reliability** Our symbolic solver leverages RDKit (Landrum et al., 2006), the industry-standard cheminformatics toolkit. While we inherit RDKit's computational assumptions, this ensures compatibility with existing workflows and reproducibility across research groups. Rather than resolving fundamental chemical ambiguities (e.g., tautomerism, aromaticity models), we prioritize *self-consistency*: tautomers maintain distinct identities, aromatic and Kekulé representations yield identical outputs despite different internal states, and undefined stereochemistry is explicitly tracked. Validation on 10,000 randomly sampled molecules demonstrates 100% internal consistency-stereochemistry counts always sum correctly, property calculations remain deterministic, and all 50+ solver methods produce valid outputs without crashes. This approach acknowledges that different chemical representations can be equally valid while ensuring that our benchmark produces reproducible results regardless of the underlying representational choices.

**Quality controls and reproducibility** Every generated task undergoes self-consistency validation: property values computed during generation match those at evaluation, constraint satisfiability is verified against the source molecular dataset, and parent-child task relationships preserve property values across difficulty levels. To ensure reliable and reproducible benchmarks, we implement several quality measures:

- **Deterministic sampling:** All random processes use fixed seeds
- **Usage tracking:** Counters ensure balanced coverage across categories and properties
- **Deduplication:** Tasks are deduplicated and assigned unique identifiers
- **Validation:** All constraints are verified against source molecules using symbolic checkers

The resulting dataset is exported as a HuggingFace with complete test splits and task-type organization. Figure 2 provides a visual overview of the dataset's structure—including task-type distribution, multitask load, and Bertz complexity—where the sunburst on the left depicts the three hierarchical levels and the stacked bars on the right show the feature-group composition across counting, indexing, and generation tasks. Figure A2 shows the distribution of molecular Bertz complexity for MOLECULARIQ compared with ChemCOTBench and ChemIQ.

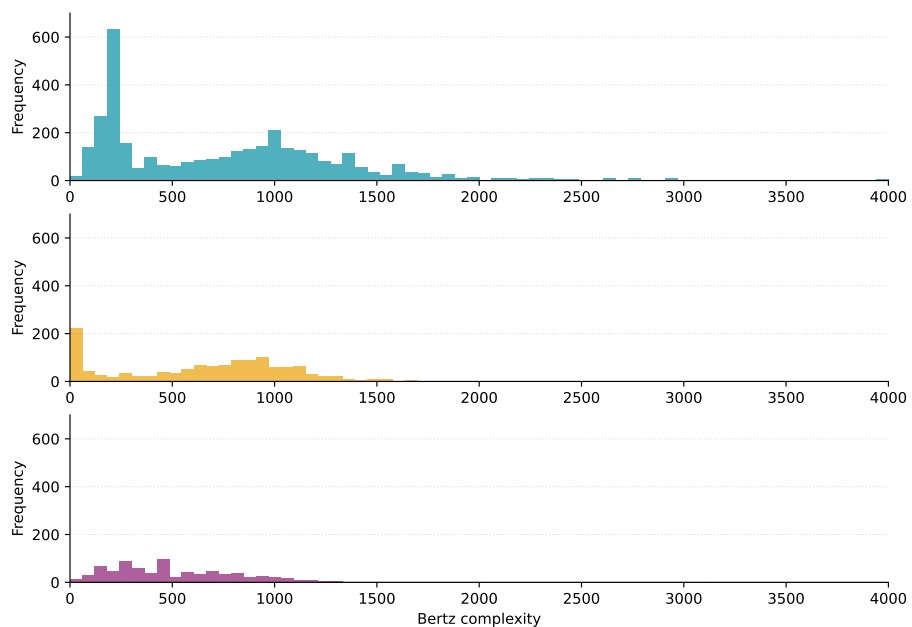

Figure A2: Bertz complexity distribution of molecules in MOLECULARIQ (top), ChemCOTBench (middle), and ChemIQ (bottom).

## A.3 QUESTION TEMPLATES AND EXAMPLES

Questions are generated from predefined templates (see Figure A3). Separate template sets are used for single- and multi-feature counting, single- and multi-feature indexing, and constrained generation. Individual questions are instantiated by replacing the template keywords *constraint*, *count_type(s)*, and *index_type(s)* with the corresponding property descriptors. These descriptors are preprocessed by a natural language formatter to ensure grammatical correctness and consistency. For the final test set, we further refined the questions using API calls to both GPT-5-mini and Claude 4.5 Sonnet; employing multiple models mitigates potential bias toward any single model's phrasing or style. Question quality was verified through both automatic and manual checks, such as manually inserting SMILES strings and verifying the correctness of JSON keys.

```
task_templates = {
    "single_count": {
        "Count the number of {count_type} in the molecule {smiles}",
        "Analyze {smiles} and report the number of {count_type}.",
        "What is the {count_type} count in {smiles}?",
        ...
    },
    "multi_count": {
        "For the molecule {smiles}, count the following features: {
            count_types}.",
        "List the counts for each of these features in {smiles}: {
            count_types}.",
        "Summarize the counts for {count_types} detected in {smiles}.",
        ...
    },
    "single_index": {
        "Identify the indices of {index_type} in the molecule {smiles}.",
        "Output the index list of {index_type} in compound {smiles}.",
        "Which atom indices represent {index_type} in {smiles}?",
        ...
    },
    "multi_index": {
        "For {smiles}, please locate the indices of: {index_types}.",
        "For compound {smiles}, locate: {index_types}.",
        "Where are the {index_types} located in {smiles}?",
        ...
    },
    "constraint_generation": {
        "Generate a molecule with {constraint}."
        "Propose a molecular structure that respects {constraint}."
        "Can you generate a molecule that satisfies {constraint}?"
    }
}
```

Figure A3: Task templates to generate single- and multitask questions for counting tasks, indexing tasks, and constraint generation.

Here, we show some example questions:

---

**Question example for single count tasks**

**Question**: How many carbon atoms are in C1(CC)N(N=C1C(CC)CCC)C? Return the numerical answer as a JSON object with the field carbon_atom_count.

---

**Question example for single index-based attribution tasks**

**Question**: Which atoms in c1(CN2c3cc(F)ccc3C(N)C2=O)ccccn1 are part of saturated rings? Please provide your answer using saturated_ring_index for the list of atom indices.

---

**Question example for single constraint generation tasks**

**Question**: Recommend a molecule that fulfills exactly 13 hydrogen bond acceptors. Return the result as a JSON object using the key 'smiles'.

**Question example for multi count tasks**

**Question**: How many unspecified E/Z (stereochemistry) double bonds are in C(C(c1ccc(cc1Cl)F)O)c1ccsc1, what is its molecular formula, and how many atoms make up the longest continuous carbon chain? Return the answers as a JSON object using the keys stereochemistry_unspecified_double_bond_count, molecular_formula_count, and longest_carbon_chain_count.

**Question example for multi index-based attribution tasks**

**Question**: Which atom indices in N#CC1CNCCN1CC1CCOCO1 are:
- hydrogen bond acceptors,
- heteroatoms,
- atoms in aliphatic rings,
- branch points,
- atoms in Z-double bonds?

Give the actual atom index positions for each feature and provide the answer as a JSON object mapping hba_index, hetero_atom_index, aliphatic_ring_index, branch_point_index, and e_z_stereochemistry_double_bond_z_index to their respective index lists.

**Question example for multi constraint generation tasks**

**Question**: Build a molecule containing exactly 5 S-stereocenters and exactly 5 saturated rings. Respond with a JSON mapping whose key is 'smiles'.

# B  DETAILS ON THE BENCHMARK EXPERIMENT

## B.1  MODELS

This section provides details on the models included in the benchmark experiment, such as the full model name and its associated name on HuggingFace.

Table B1: Model nomenclature mapping our abbreviations to official names and HuggingFace identifiers.

| Abbreviation Used | Model Name | HuggingFace Identifier |
|---|---|---|
| *Chemistry LLMs* | | |
| ChemLLM | ChemLLM (Zhao et al., 2025c) | AI4Chem/ChemLLM-7B-Chat |
| LlaSMol | LlaSMol (Yu et al., 2024) | osunlp/LlaSMol-Mistral-7B |
| MolReasoner-Cap | MolReasoner (Zhao et al., 2025a) | guojianz/MolReasoner (captioning file) |
| MolReasoner-Gen | MolReasoner (Zhao et al., 2025a) | guojianz/MolReasoner (generation file) |
| Llama-3 MolInst | Mol-Instructions (Fang et al., 2024) | zjunlp/llama3-instruct-molinst-molecule-8b |
| ChemDFM-8B | ChemDFM (Zhao et al., 2025c) | OpenDFM/ChemDFM-v1.0-8B |
| ChemDFM-13B | ChemDFM (Zhao et al., 2025c) | OpenDFM/ChemDFM-v1.0-13B |
| ChemDFM-R-14B | ChemDFM-R (Zhao et al., 2025b) | OpenDFM/ChemDFM-R-14B |
| TxGemma-9B | TxGemma (Wang et al., 2025a) | google/txgemma-9b-chat |
| TxGemma-27B | TxGemma (Wang et al., 2025a) | google/txgemma-27b-chat |
| Ether0 | Ether0 (Narayanan et al., 2025) | futurehouse/ether0 |
| *Generalist LLMs* | | |
| Gemma-2 9B | Gemma-2-9b-it (Gemma Team, 2024) | google/gemma-2-9b-it |
| Gemma-2 27B | Gemma-2-27b-it (Gemma Team, 2024) | google/gemma-2-27b-it |
| Gemma-3 12B | Gemma-3-12b-it (Gemma Team, 2025) | google/gemma-3-12b-it |
| Gemma-3 27B | Gemma-3-27b-it (Gemma Team, 2025) | google/gemma-3-27b-it |
| LLaMA-2 13B | LLaMA-2-13b-chat (Touvron et al., 2023) | meta-llama/Llama-2-13b-chat-hf |
| LLaMA-3 8B | Llama 3 8B (AI@Meta, 2024) | meta-llama/Meta-Llama-3-8B-Instruct |
| LLaMA-3.3 70B | NVIDIA Llama 3.3 70B Instruct FP8 (NVIDIA, 2025b) | nvidia/Llama-3.3-70B-Instruct-FP8 |
| Mistral-7B v0.1 | Mistral 7B (Jiang et al., 2023) | mistralai/Mistral-7B-v0.1 |
| Mistral-Small | Mistral-Small-24B-Instruct-2501 (Mistral AI, 2025) | mistralai/Mistral-Small-24B-Instruct-2501 |
| Nemotron-Nano 9B v2 | NVIDIA-Nemotron-Nano-9B-v2 (NVIDIA, 2025a) | nvidia/NVIDIA-Nemotron-Nano-9B-v2 |
| SEED-OSS | Seed-OSS-36B-Instruct (ByteDance Seed Team, 2025) | ByteDance-Seed/Seed-OSS-36B-Instruct |
| Qwen-2.5 7B | Qwen2.5-7B-Instruct (Qwen Team, 2024) | Qwen/Qwen2.5-7B-Instruct |
| Qwen-3 8B | Qwen3-8B (Qwen Team, 2025) | Qwen/Qwen3-8B |
| Qwen-3 14B | Qwen3-14B (Qwen Team, 2025) | Qwen/Qwen3-14B |
| Qwen-3 32B | Qwen3-32B (Qwen Team, 2025) | Qwen/Qwen3-32B |
| Qwen-3 30B | Qwen3-30B-A3B (Qwen Team, 2025) | Qwen/Qwen3-30B-A3B-Thinking-2507 |
| Qwen-3 Next 80B | Qwen3-Next-80B-A3B (Qwen Team, 2025) | Qwen/Qwen3-Next-80B-A3B-Thinking |
| Qwen-3 235B | Qwen3-235B-A22B (Qwen Team, 2025) | Qwen/Qwen3-235B-A22B-Thinking-2507-FP8 |
| GPT-OSS 20B (Low*) | GPT-oss-20b (OpenAI, 2025) | openai/gpt-oss-20b |
| GPT-OSS 20B (Med*) | GPT-oss-20b (OpenAI, 2025) | openai/gpt-oss-20b |
| GPT-OSS 20B (High*) | GPT-oss-20b (OpenAI, 2025) | openai/gpt-oss-20b |
| GPT-OSS 120B (Low*) | GPT-oss-120b (OpenAI, 2025) | openai/gpt-oss-120b |
| GPT-OSS 120B (Med*) | GPT-oss-120b (OpenAI, 2025) | openai/gpt-oss-120b |
| GPT-OSS 120B (High*) | GPT-oss-120b (OpenAI, 2025) | openai/gpt-oss-120b |
| GLM-4.6 | GLM-4.6 (Zeng et al., 2025; Z.ai, 2025) | zai-org/GLM-4.6-FP8 |
| DeepSeek-R1-0528 | DeepSeek-R1-0528 (Guo et al., 2025) | nvidia/DeepSeek-R1-0528-FP4-v2 |

\* Low/Med/High refer to reasoning effort settings, not separate models

## B.2  DETAILS ON EVALUATION

We evaluate models using the `lm-evaluation-harness` framework (Gao et al., 2024), which provides a uniform interface across local and API-served models, standardized task configuration, and comprehensive logging of prompts, seeds, and versions. We integrate MOLECULARIQ as a custom task to ensure reproducible evaluation while allowing model-specific optimization of prompting strategies and extraction methods to minimize formatting bias.

### B.2.1  INTEGRATION INTO `LM-EVALUATION-HARNESS`

**Framework integration.** We add MOLECULARIQ as a YAML-configured task under `lm_eval/tasks/MOLECULARIQ/` within the `lm-evaluation-harness` framework (Gao et al., 2024). The dataset is hosted on the Hugging Face Hub and, upon publication, will be accessible through the standard dataset loading interface.

**MOLECULARIQ task configuration.**

```
1   task: moleculariq_pass_at_k
2   dataset_path: moleculariq
3   dataset_name: default
4   output_type: generate_until
5   test_split: test
6   repeats: 3
7
8   process_docs: !function task_processor.process_docs
9   doc_to_text: !function task_processor.doc_to_text
10  doc_to_target: target
11  process_results: !function task_processor.process_results_pass_at_k
12
13  metric_list:
14    - metric: avg_accuracy
15      aggregation: mean
16      higher_is_better: true
17    - metric: extracted_answers
18      aggregation: bypass
19      higher_is_better: true
```

**Generation parameters.** To reflect realistic practitioner usage and model provider suggestions, we follow each model's native generation settings. For models that include a `generation_config.json` file, we adopt their specified parameters (temperature, top-p, top-k, etc.). Models without explicit generation configs fall back to backend defaults.

**Flexible prompt and extraction strategies.** Our architecture enables easy experimentation with different prompting and extraction approaches. Each model can have multiple configuration variants to test different system prompts, extraction functions, or preprocessing strategies. We evaluate models using our standardized chemistry expert prompt by default (B.2.2), but additionally, if practitioners provide model-specific prompts or alternative extraction methods, we test multiple configurations and report the best performance.

**Model-specific processing pipeline.** The evaluation system is model-agnostic, with processing functions defined per model configuration rather than at the task level. If none is provided, we fallback to default implementations. An example skeleton config is presented below. For each model, we specify:

- `extraction_function`: Controls how answers are extracted from model outputs. In Section B.2.3 we describe the standard extraction function we used for almost all models in this study.
- `preprocessing_function`: Optional model-specific input preprocessing (e.g., adding specific tags to certain inputs like smiles)
- `system_prompt`/`inline_prompt`: Model-appropriate instruction formats

**API and local model support.** The framework supports both local model inference and API-based evaluation through various backends. For API models, authentication and endpoint configuration are handled through the model configuration file, enabling seamless evaluation of commercial models alongside open-source alternatives.

**Model configuration template.**

```
# Model configuration structure (YAML)
model_name: <model-identifier>
model: <framework>       # e.g., vllm, openai, anthropic
chem_model_config: configs/<config-file>.yaml
model_args:
  pretrained: <model-path>
  tensor_parallel_size: <int>
gen_kwargs:
  temperature: <float> # 0.0-1.0
  top_p: <float>       # 0.0-1.0
  top_k: <int>         # top-k sampling
  min_p: <float>       # minimum probability
  max_gen_toks: <int>  # max generation tokens
  until: []            # stop sequences
  do_sample: <bool>    # sampling enabled
system_prompt: "<prompt-text>"
preprocessing_function:  <module>:<function>
extraction_function: <module>:<function>
```

**Multi-rollout evaluation.** Stochastic variability is captured through the `repeats:3` parameter, which automatically runs three independent generations per instance with different random seeds.

**Metrics and aggregation.** We report `avg_accuracy` as the primary metric, computed as the mean score across the three rollouts for each instance. While the framework supports pass@k evaluation (pass@1, pass@3, etc.), we focus on average accuracy for direct model comparison. Individual extracted answers are logged via the `extracted_answers` metric for debugging and analysis.

**Execution.** Models are evaluated using our wrapper that reads the model-specific configuration and constructs appropriate `lm-eval` arguments:

**MOLECULARIQ execution.**

```
# Evaluate a local model
python evaluate_model.py \
  --model_config configs/nemotron-nano-9b-v2.yaml \
  --task molecular iq_pass_at_k \
  --output_dir results/

# Evaluate an API model (ensure API keys are set)
export OPENAI_API_KEY="your-api-key"
python evaluate_model.py \
  --model_config configs/gpt-4o-mini.yaml \
  --task molecular iq_pass_at_k \
  --output_dir results/

# The wrapper automatically applies:
# - Model-specific extraction and preprocessing functions
# - Appropriate system/inline prompts
# - Generation parameters and rollout settings
# - API authentication and endpoint configuration
```

### B.2.2 SYSTEM PROMPT

---

**System prompt**

You are an expert chemist. Answer molecular property, understanding, structural analysis and molecular generation questions precisely and accurately.

CRITICAL: Only content within<answer></answer> tags will be extracted. ALWAYS return JSON format.

KEY REQUIREMENT: Use EXACT key names from the question. Never modify or invent keys.

INDEXING: Atoms are indexed from 0 to the end of the SMILES string from left to right. Only heavy atoms (skip [H], include [2H]/[3H]).
Examples:
- "CCO": C(0), C(1), O(2)
- "CC(C)O": C(0), C(1), C(2), O(3)
- "CC(=O)N": C(0), C(1), O(2), N(3)

ABSENT FEATURES: Use 0 for counts, [] for indices. Never null or omit.

ALWAYS USE JSON with EXACT keys from the question:

Single count (key from question: "alcohol_count"):
<answer>"alcohol_count": 2</answer>
<answer>"alcohol_count": 0</answer> (if absent)

Single index (key from question: "ketone_indices"):
<answer>"ketone_indices": [5]</answer>
<answer>"ketone_indices": []</answer> (if absent)

Multiple properties (keys from question: "ring_count", "halogen_indices"):
<answer>"ring_count": 2, "halogen_indices": [3, 7]</answer>
<answer>"ring_count": 0, "halogen_indices": []</answer> (if all absent)

Constraint generation:
<answer>"smiles": "CC(O)C"</answer>

Include ALL requested properties. Never null or omit.

---

### B.2.3 VERIFIERS AND ANSWER EXTRACTION

**Design motivation.** Requiring models to emit exact JSON schemas risks conflating chemical reasoning capabilities with instruction-following proficiency. To isolate assessment of chemical understanding, our verifiers accept flexible output formats while maintaining semantic precision through key-specific matching. Specifically, we support JSON dictionaries, comma-separated values, varied bracket notations, and natural language property names (e.g., accepting both `{"ring_count": 3}` and `"number of rings:  3"`), all normalized to canonical forms. This design is motivated by empirical evidence showing substantial variation in instruction-following across models (Zhou et al., 2023), and our observation that models may correctly identify chemical properties but express them in unexpected formats. By decoupling format compliance from chemical accuracy, we ensure that evaluation genuinely measures understanding of molecular properties rather than syntactic adherence, while our key-specific approach preserves the granular trackability necessary for detailed performance analysis.

**Output parsing hierarchy.** We extract answers using a four-tier cascade, applied in order until successful:

1. **Explicit answer tags and JSON:** If `<answer>...</answer>` tags are found and a JSON-like object is present within them, we parse it as the answer.

2. **Answer tags only:** When only `<answer>...</answer>` tags exist (no JSON), apply lightweight parsing: split on common delimiters (commas, semicolons, line breaks), then coerce strings to appropriate types (numbers, booleans).

3. **JSON without answer tags:** Locate the final well-formed JSON object in the output. Apply repair heuristics: strip markdown code fences, fix trailing commas, normalize quote styles, then parse into structured data.

4. **Raw text fallback:** If everything else fails, we return the last continuous string as the answer.

**Content normalization.** All extracted values undergo standardization before comparison:

- **Text processing:** Unicode normalization, case folding, whitespace compression, punctuation removal.

- **Type coercion:** Convert numbers to appropriate type, standardize bracket notation to `[]`, map null/empty values to empty lists.

- **Value extraction and list normalization:** Extract all values from JSON objects, discarding key names. Scalar values are wrapped in single-element lists for uniform comparison (e.g., `{a:1, b:[2,3]}` becomes `[[1], [2,3]]`).

**Key-specific matching for performance trackability.** To enable fine-grained analysis of model capabilities across varying task complexities, we employ key-specific matching that preserves property-level granularity. Given target dictionary $T = \{(k_i^t, v_i^t)\}_{i=1}^n$ and predicted dictionary $P = \{(k_j^p, v_j^p)\}_{j=1}^m$, we define correctness as:

$$\text{correct}(T, P) = \bigwedge_{i=1}^n \exists_{j \in [1,m]} \left[ \text{canon}(k_i^t) = \text{canon}(k_j^p) \wedge \text{norm}(v_i^t) = \text{norm}(v_j^p) \right]$$

where $\text{canon}(\cdot)$ canonicalizes property names and $\text{norm}(\cdot)$ normalizes values. This formulation permits $|P| \geq |T|$, though extraneous keys are ignored for scoring but kept for manual inspection; duplicate keys are resolved by retaining the final occurrence. An instance receives a score of 1 if all properties match, 0 otherwise. This design crucially enables tracking of individual property performance within all the dimensions that MOLECULARIQ provides (complexity, task type, multitask load, etc.).

**Task-specific verification.** All tasks employ the key-specific matching framework with type-appropriate value comparisons:

- **Counting tasks:** Values must be integers or convertible to integers. For each canonicalized property key, we verify $\text{norm}(v^p) = \text{norm}(v^t)$ where both values are coerced to integers. Multi-count tasks require all properties in the target to have matching predictions.

- **Indexing tasks:** Values are lists of atom indices. For each property, we compare sets of indices: $\text{set}(v^p) = \text{set}(v^t)$, where order within lists is ignored but property associations are preserved (e.g., target `{"aromatic": [0,2], "aliphatic": [1]}` requires both properties to match).

- **Molecular generation:** No predetermined target exists. We extract the generated SMILES string from the predicted dictionary (searching for common key variations like `smiles`, `molecule`), then verify it satisfies all specified constraints through deterministic property validation (functional groups, molecular formula, ring counts, etc.). The score is binary: 1 if all constraints are satisfied, 0 otherwise. Unlike prior work (Narayanan et al., 2025), we exclude purchasability requirements to ensure reproducibility and eliminate API dependencies.

**Cross-complexity analysis benefits.** The key-specific approach enables three critical analyses: (1) **Subtask scaling**—tracking how individual property performance degrades as the total number of properties increases, (2) **Cross-task consistency**—comparing performance on the same property across different task types (counting vs. indexing), and (3) **Error correlation**—identifying which property combinations prove particularly challenging when evaluated together. This granular trackability transforms evaluation from binary pass/fail to actionable insights about which specific capabilities need improvement at each complexity level.

**Reproducibility measures.** We ensure reproducibility by pinning software versions (`lm-eval` commit hashes, dataset versions), model configurations, and random seeds. Hardware-specific non-determinism may persist despite these controls.

## C EXTENDED RESULTS

Here we provide the complete set of evaluation results for all models assessed on MOLECULARIQ, complementing the aggregate metrics reported in the main text. We present detailed breakdowns across the three benchmark dimensions-task category, multitask load, and molecular complexity-as well as analyses of feature families, SMILES formats, and representation variants. These results enable fine-grained inspection of the strengths and limitations of both generalist and chemistry-specialized LLMs.

### C.1 EXTENDED PERFORMANCE BREAKDOWN

This section reports complete performance profiles for all 38 evaluated models across the full MOLECULARIQ task suite. Whereas Section 5 focuses on high-level trends and top-performing models, the tables and figures below provide model-by-model accuracy decomposed along task type (counting, indexing, constrained generation), multitask load, and molecular complexity. We additionally include per-feature-family results covering the six molecular-feature categories defined in Section 3.2.

#### C.1.1 OVERALL PERFORMANCE AND ACROSS BENCHMARK DIMENSIONS

Tables C1–C3 present the core performance breakdown across benchmark dimensions. Table C1 reports overall accuracy across the three task types for all evaluated models. Table C2 shows accuracy as a function of multitask load, ranging from single-task queries to compositions of up to five simultaneous subtasks. Table C3 compares performance across Bertz complexity bins. Together, these results reveal substantial variation in model capabilities: performance degrades with both compositional complexity and molecular complexity for nearly all models, though the rate of degradation varies considerably.

For generation tasks, Bertz complexity cannot be computed directly since there is no input molecule to measure. Instead, we assess task difficulty by the prevalence of molecules satisfying each constraint set within our test pool. Figure C1 plots the percentage of molecules fulfilling a given constraint set against achieved model performance. A clear trend emerges: rarer constraints-those satisfied by fewer molecules in the pool-are substantially harder for models to fulfill, indicating that generation difficulty scales inversely with constraint prevalence.

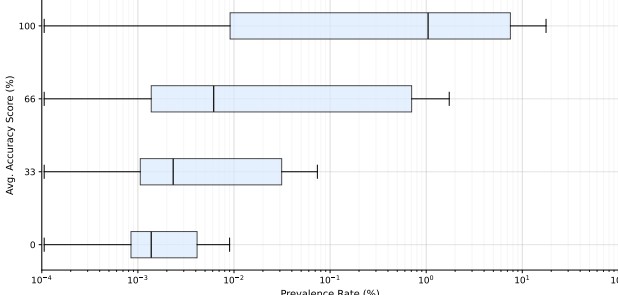

Figure C1: Model performance on generation tasks as a function of constraint prevalence. The x-axis shows the percentage of molecules in the test pool that satisfy a given constraint set; the y-axis shows model accuracy. Rarer constraints (lower prevalence) are consistently harder for models to fulfill.

Table C1: Results on the MOLECULARIQ benchmark. We report overall accuracy as well as accuracy across task families (in %). Error bars indicate standard errors across instances. Model size refers to parameter count; R is a flag for reasoning models.

| | Size | R | Overall | Task family | | |
| --- | --- | --- | --- | --- | --- | --- |
| | | | | Counting | Indexing | Generation |
| *Chemistry LLMs* | | | | | | |
| ChemLLM | 7B | | $2.2 \pm 0.2$ | $2.0 \pm 0.2$ | $2.7 \pm 0.3$ | $1.9 \pm 0.3$ |
| LlaSMol | 7B | | $1.5 \pm 0.1$ | $1.6 \pm 0.2$ | $1.8 \pm 0.3$ | $1.1 \pm 0.2$ |
| MolReasoner-Cap | 7B | ✓ | $3.5 \pm 0.2$ | $5.6 \pm 0.5$ | $0.8 \pm 0.1$ | $4.0 \pm 0.4$ |
| MolReasoner-Gen | 7B | ✓ | $3.2 \pm 0.2$ | $4.9 \pm 0.4$ | $1.5 \pm 0.2$ | $3.2 \pm 0.3$ |
| Llama-3 MolInst | 8B | | $2.0 \pm 0.2$ | $4.6 \pm 0.4$ | $0.5 \pm 0.1$ | $0.7 \pm 0.2$ |
| ChemDFM-8B | 8B | | $3.7 \pm 0.2$ | $4.9 \pm 0.4$ | $0.5 \pm 0.1$ | $6.0 \pm 0.5$ |
| ChemDFM-13B | 13B | | $2.7 \pm 0.2$ | $4.1 \pm 0.3$ | $1.1 \pm 0.2$ | $2.7 \pm 0.3$ |
| ChemDFM-R-14B | 14B | ✓ | $8.7 \pm 0.4$ | $12.9 \pm 0.8$ | $2.8 \pm 0.4$ | $10.5 \pm 0.8$ |
| TxGemma-9B | 9B | | $2.6 \pm 0.2$ | $4.1 \pm 0.4$ | $0.5 \pm 0.1$ | $3.1 \pm 0.3$ |
| TxGemma-27B | 27B | | $5.0 \pm 0.2$ | $7.0 \pm 0.5$ | $1.8 \pm 0.3$ | $6.2 \pm 0.5$ |
| Ether0 | 24B | ✓ | $6.5 \pm 0.3$ | $3.2 \pm 0.3$ | $0.1 \pm 0.0$ | $17.5 \pm 0.8$ |
| *Generalist LLMs* | | | | | | |
| Gemma-2 9B | 9B | | $4.0 \pm 0.2$ | $3.9 \pm 0.3$ | $1.9 \pm 0.3$ | $6.3 \pm 0.5$ |
| Gemma-2 27B | 27B | | $6.9 \pm 0.3$ | $6.6 \pm 0.5$ | $0.9 \pm 0.2$ | $13.9 \pm 0.8$ |
| Gemma-3 12B | 12B | | $5.8 \pm 0.3$ | $6.4 \pm 0.5$ | $2.8 \pm 0.4$ | $8.4 \pm 0.6$ |
| Gemma-3 27B | 27B | | $7.9 \pm 0.4$ | $9.9 \pm 0.7$ | $1.4 \pm 0.3$ | $12.7 \pm 0.8$ |
| LLaMA-2 13B | 13B | | $3.1 \pm 0.2$ | $3.9 \pm 0.4$ | $0.2 \pm 0.1$ | $5.3 \pm 0.5$ |
| LLaMA-3 8B | 8B | | $4.8 \pm 0.3$ | $6.0 \pm 0.5$ | $0.5 \pm 0.1$ | $8.1 \pm 0.6$ |
| LLaMA-3.3 70B | 70B | | $9.1 \pm 0.4$ | $11.1 \pm 0.7$ | $1.9 \pm 0.3$ | $14.7 \pm 0.8$ |
| Mistral-7B v0.1 | 7B | | $1.4 \pm 0.1$ | $2.0 \pm 0.2$ | $1.1 \pm 0.2$ | $1.2 \pm 0.2$ |
| Mistral-Small | 24B | | $8.3 \pm 0.3$ | $9.7 \pm 0.6$ | $1.6 \pm 0.3$ | $14.1 \pm 0.8$ |
| Nemotron-Nano 9B v2 | 9B | ✓ | $11.1 \pm 0.4$ | $12.2 \pm 0.7$ | $6.6 \pm 0.5$ | $14.8 \pm 0.8$ |
| SEED-OSS | 36B | ✓ | $24.3 \pm 0.5$ | $26.8 \pm 0.9$ | $25.6 \pm 0.9$ | $20.0 \pm 0.9$ |
| Qwen-2.5 7B | 7B | | $5.1 \pm 0.3$ | $6.0 \pm 0.5$ | $1.9 \pm 0.3$ | $7.5 \pm 0.6$ |
| Qwen-2.5 14B | 14B | | $6.6 \pm 0.3$ | $7.6 \pm 0.6$ | $1.2 \pm 0.3$ | $11.5 \pm 0.8$ |
| Qwen-3 8B | 8B | ✓ | $11.3 \pm 0.4$ | $12.5 \pm 0.7$ | $5.9 \pm 0.5$ | $15.8 \pm 0.8$ |
| Qwen-3 14B | 14B | ✓ | $13.1 \pm 0.4$ | $14.4 \pm 0.7$ | $7.5 \pm 0.5$ | $17.7 \pm 0.8$ |
| Qwen-3 32B | 32B | ✓ | $15.6 \pm 0.4$ | $17.0 \pm 0.7$ | $9.1 \pm 0.6$ | $21.0 \pm 0.9$ |
| Qwen-3 30B | 30B (A3B) | ✓ | $22.7 \pm 0.5$ | $23.0 \pm 0.9$ | $16.5 \pm 0.8$ | $29.4 \pm 1.0$ |
| Qwen-3 Next 80B | 80B (A3B) | ✓ | $32.3 \pm 0.6$ | $30.7 \pm 1.0$ | $26.9 \pm 0.9$ | $40.0 \pm 1.1$ |
| Qwen-3 235B | 235B (A22B) | ✓ | $39.2 \pm 0.6$ | $37.1 \pm 1.0$ | $34.5 \pm 1.0$ | $46.7 \pm 1.1$ |
| GPT-OSS 20B (Low) | 20B (A4B) | ✓ | $10.8 \pm 0.4$ | $13.0 \pm 0.7$ | $5.7 \pm 0.5$ | $14.0 \pm 0.8$ |
| GPT-OSS 20B (Med) | 20B (A4B) | ✓ | $27.2 \pm 0.5$ | $27.9 \pm 0.9$ | $22.2 \pm 0.8$ | $32.1 \pm 1.0$ |
| GPT-OSS 20B (High) | 20B (A4B) | ✓ | $35.8 \pm 0.6$ | $39.1 \pm 1.0$ | $32.8 \pm 1.0$ | $35.2 \pm 1.0$ |
| GPT-OSS 120B (Low) | 120B (A5B) | ✓ | $15.9 \pm 0.5$ | $17.1 \pm 0.8$ | $10.7 \pm 0.6$ | $20.3 \pm 0.9$ |
| GPT-OSS 120B (Med) | 120B (A5B) | ✓ | $26.5 \pm 0.5$ | $29.1 \pm 0.9$ | $22.3 \pm 0.8$ | $28.0 \pm 1.0$ |
| GPT-OSS 120B (High) | 120B (A5B) | ✓ | $\mathbf{47.5 \pm 0.6}$ | $\mathbf{46.8 \pm 1.1}$ | $\mathbf{42.5 \pm 1.1}$ | $\mathbf{53.7 \pm 1.1}$ |
| GLM-4.6 | 355B (A32B) | ✓ | $16.2 \pm 0.4$ | $15.9 \pm 0.7$ | $11.3 \pm 0.6$ | $22.0 \pm 0.9$ |
| DeepSeek-R1-0528 | 671B (A37B) | ✓ | $37.1 \pm 0.6$ | $36.4 \pm 1.0$ | $30.2 \pm 0.9$ | $45.5 \pm 1.1$ |

Table C2: Accuracy (%) on MOLECULARIQ by multitask load, the number of items requested per prompt (1,2,3,5). Error bars indicate standard errors across instances. Model size refers to parameter count; R is a flag for reasoning models.

| | Size | R | 1 | 2 | 3 | 5 |
|---|---|---|---|---|---|---|
| | | | **Multitask load** | | | |
| *Chemistry LLMs* | | | | | | |
| ChemLLM | 7B | | $2.8 \pm 0.2$ | $0.3 \pm 0.2$ | $0.1 \pm 0.1$ | $0.0 \pm 0.0$ |
| LlaSMol | 7B | | $2.8 \pm 0.2$ | $0.2 \pm 0.1$ | $0.0 \pm 0.0$ | $0.0 \pm 0.0$ |
| MolReasoner-Cap | 7B | ✓ | $6.3 \pm 0.4$ | $1.5 \pm 0.3$ | $0.3 \pm 0.1$ | $0.1 \pm 0.1$ |
| MolReasoner-Gen | 7B | ✓ | $5.5 \pm 0.4$ | $1.5 \pm 0.3$ | $0.2 \pm 0.1$ | $0.0 \pm 0.0$ |
| Llama-3 MolInst | 8B | | $3.6 \pm 0.3$ | $0.6 \pm 0.2$ | $0.3 \pm 0.1$ | $0.0 \pm 0.0$ |
| ChemDFM-8B | 8B | | $7.1 \pm 0.4$ | $1.0 \pm 0.2$ | $0.1 \pm 0.1$ | $0.1 \pm 0.1$ |
| ChemDFM-13B | 13B | | $4.7 \pm 0.3$ | $0.5 \pm 0.2$ | $0.3 \pm 0.1$ | $0.0 \pm 0.0$ |
| ChemDFM-R-14B | 14B | ✓ | $16.4 \pm 0.8$ | $3.6 \pm 0.6$ | $1.2 \pm 0.4$ | $0.3 \pm 0.2$ |
| TxGemma-9B | 9B | | $5.1 \pm 0.4$ | $0.4 \pm 0.2$ | $0.0 \pm 0.0$ | $0.0 \pm 0.0$ |
| TxGemma-27B | 27B | | $9.6 \pm 0.5$ | $1.8 \pm 0.4$ | $0.4 \pm 0.1$ | $0.1 \pm 0.1$ |
| Ether0 | 24B | ✓ | $11.7 \pm 0.6$ | $4.2 \pm 0.6$ | $1.1 \pm 0.3$ | $0.3 \pm 0.1$ |
| *Generalist LLMs* | | | | | | |
| Gemma-2 9B | 9B | | $7.0 \pm 0.4$ | $0.7 \pm 0.2$ | $0.4 \pm 0.1$ | $0.0 \pm 0.0$ |
| Gemma-2 27B | 27B | | $13.2 \pm 0.6$ | $2.6 \pm 0.4$ | $1.1 \pm 0.3$ | $0.1 \pm 0.1$ |
| Gemma-3 12B | 12B | | $10.8 \pm 0.6$ | $0.2 \pm 0.1$ | $0.0 \pm 0.0$ | $0.0 \pm 0.0$ |
| Gemma-3 27B | 27B | | $14.9 \pm 0.7$ | $3.3 \pm 0.5$ | $1.1 \pm 0.3$ | $0.1 \pm 0.1$ |
| LLaMA-2 13B | 13B | | $6.2 \pm 0.4$ | $0.4 \pm 0.2$ | $0.2 \pm 0.1$ | $0.0 \pm 0.0$ |
| LLaMA-3 8B | 8B | | $9.4 \pm 0.5$ | $0.9 \pm 0.3$ | $0.3 \pm 0.1$ | $0.0 \pm 0.0$ |
| LLaMA-3.3 70B | 70B | | $17.1 \pm 0.7$ | $3.6 \pm 0.6$ | $0.9 \pm 0.3$ | $0.6 \pm 0.3$ |
| Mistral-7B v0.1 | 7B | | $2.4 \pm 0.2$ | $0.2 \pm 0.1$ | $0.0 \pm 0.0$ | $0.0 \pm 0.0$ |
| Mistral-Small | 24B | | $15.9 \pm 0.7$ | $3.3 \pm 0.5$ | $1.0 \pm 0.2$ | $0.2 \pm 0.1$ |
| Nemotron-Nano 9B v2 | 9B | ✓ | $20.8 \pm 0.7$ | $5.0 \pm 0.6$ | $1.9 \pm 0.3$ | $0.5 \pm 0.2$ |
| SEED-OSS | 36B | ✓ | $38.8 \pm 0.9$ | $18.9 \pm 1.1$ | $10.8 \pm 0.9$ | $4.2 \pm 0.5$ |
| Qwen-2.5 7B | 7B | | $8.9 \pm 0.5$ | $1.6 \pm 0.4$ | $0.3 \pm 0.2$ | $0.2 \pm 0.1$ |
| Qwen-2.5 14B | 14B | | $12.5 \pm 0.7$ | $2.2 \pm 0.5$ | $0.8 \pm 0.3$ | $0.4 \pm 0.2$ |
| Qwen-3 8B | 8B | ✓ | $20.4 \pm 0.7$ | $5.7 \pm 0.6$ | $2.8 \pm 0.4$ | $0.7 \pm 0.2$ |
| Qwen-3 14B | 14B | ✓ | $23.6 \pm 0.8$ | $6.9 \pm 0.7$ | $3.0 \pm 0.4$ | $1.0 \pm 0.3$ |
| Qwen-3 32B | 32B | ✓ | $27.2 \pm 0.8$ | $9.7 \pm 0.8$ | $4.4 \pm 0.5$ | $1.6 \pm 0.3$ |
| Qwen-3 30B | 30B (A3B) | ✓ | $35.9 \pm 0.8$ | $18.4 \pm 1.0$ | $10.0 \pm 0.8$ | $4.5 \pm 0.5$ |
| Qwen-3 Next 80B | 80B (A3B) | ✓ | $46.6 \pm 0.9$ | $29.0 \pm 1.2$ | $19.6 \pm 1.1$ | $9.6 \pm 0.8$ |
| Qwen-3 235B | 235B (A22B) | ✓ | $55.3 \pm 0.9$ | $36.8 \pm 1.3$ | $24.4 \pm 1.2$ | $13.1 \pm 0.9$ |
| GPT-OSS 20B (Low) | 20B (A4B) | ✓ | $19.8 \pm 0.7$ | $5.5 \pm 0.6$ | $1.9 \pm 0.4$ | $0.8 \pm 0.2$ |
| GPT-OSS 20B (Med) | 20B (A4B) | ✓ | $42.4 \pm 0.8$ | $21.7 \pm 1.1$ | $14.1 \pm 0.9$ | $5.3 \pm 0.6$ |
| GPT-OSS 20B (High) | 20B (A4B) | ✓ | $51.4 \pm 0.9$ | $32.6 \pm 1.3$ | $22.4 \pm 1.1$ | $10.4 \pm 0.8$ |
| GPT-OSS 120B (Low) | 120B (A5B) | ✓ | $27.8 \pm 0.8$ | $9.6 \pm 0.8$ | $4.3 \pm 0.6$ | $1.9 \pm 0.4$ |
| GPT-OSS 120B (Med) | 120B (A5B) | ✓ | $41.1 \pm 0.8$ | $21.4 \pm 1.1$ | $12.9 \pm 0.9$ | $6.0 \pm 0.6$ |
| GPT-OSS 120B (High) | 120B (A5B) | ✓ | $\mathbf{61.5 \pm 0.9}$ | $\mathbf{46.7 \pm 1.4}$ | $\mathbf{36.8 \pm 1.4}$ | $\mathbf{21.2 \pm 1.2}$ |
| GLM-4.6 | 355B (A32B) | ✓ | $27.6 \pm 0.8$ | $10.8 \pm 0.8$ | $4.4 \pm 0.5$ | $2.3 \pm 0.4$ |
| DeepSeek-R1-0528 | 671B (A37B) | ✓ | $52.7 \pm 0.9$ | $34.7 \pm 1.3$ | $22.7 \pm 1.1$ | $12.0 \pm 0.8$ |

Table C3: Accuracy (in %) on MOLECULARIQ by Bertz complexity bin. Error bars indicate standard errors across instances. Model size refers to parameter count; R is a flag for reasoning models.

| | Size | R | Bertz Complexity | | |
| --- | --- | --- | --- | --- | --- |
| | | | 0–250 | 250–1000 | 1000+ |
| *Chemistry LLMs* | | | | | |
| ChemLLM | 7B | | $2.0 \pm 0.3$ | $0.9 \pm 0.2$ | $0.6 \pm 0.2$ |
| LlaSMol | 7B | | $2.1 \pm 0.3$ | $1.2 \pm 0.2$ | $1.2 \pm 0.2$ |
| MolReasoner-Cap | 7B | ✓ | $5.3 \pm 0.5$ | $2.4 \pm 0.4$ | $1.3 \pm 0.3$ |
| MolReasoner-Gen | 7B | ✓ | $5.4 \pm 0.5$ | $1.8 \pm 0.3$ | $1.0 \pm 0.2$ |
| Llama-3 MolInst | 8B | | $3.5 \pm 0.5$ | $2.0 \pm 0.3$ | $1.6 \pm 0.3$ |
| ChemDFM-8B | 8B | | $4.0 \pm 0.5$ | $2.1 \pm 0.3$ | $1.5 \pm 0.3$ |
| ChemDFM-13B | 13B | | $3.2 \pm 0.4$ | $2.1 \pm 0.3$ | $1.3 \pm 0.2$ |
| ChemDFM-R-14B | 14B | ✓ | $11.9 \pm 0.9$ | $7.2 \pm 0.8$ | $4.3 \pm 0.6$ |
| TxGemma-9B | 9B | | $3.8 \pm 0.5$ | $1.7 \pm 0.3$ | $1.3 \pm 0.3$ |
| TxGemma-27B | 27B | | $7.2 \pm 0.6$ | $3.2 \pm 0.4$ | $2.7 \pm 0.3$ |
| Ether0 | 24B | ✓ | $2.3 \pm 0.3$ | $1.6 \pm 0.3$ | $1.1 \pm 0.2$ |
| *Generalist LLMs* | | | | | |
| Gemma-2 9B | 9B | | $3.0 \pm 0.4$ | $2.2 \pm 0.3$ | $1.5 \pm 0.2$ |
| Gemma-2 27B | 27B | | $6.6 \pm 0.6$ | $3.0 \pm 0.4$ | $1.9 \pm 0.3$ |
| Gemma-3 12B | 12B | | $6.7 \pm 0.7$ | $2.3 \pm 0.4$ | $2.0 \pm 0.4$ |
| Gemma-3 27B | 27B | | $9.1 \pm 0.8$ | $4.7 \pm 0.6$ | $3.1 \pm 0.4$ |
| LLaMA-2 13B | 13B | | $3.4 \pm 0.4$ | $1.5 \pm 0.3$ | $1.3 \pm 0.3$ |
| LLaMA-3 8B | 8B | | $5.2 \pm 0.6$ | $2.5 \pm 0.4$ | $1.7 \pm 0.3$ |
| LLaMA-3.3 70B | 70B | | $10.0 \pm 0.9$ | $5.3 \pm 0.6$ | $4.1 \pm 0.6$ |
| Mistral-7B v0.1 | 7B | | $1.8 \pm 0.2$ | $1.1 \pm 0.2$ | $0.7 \pm 0.1$ |
| Mistral-Small | 24B | | $9.9 \pm 0.8$ | $4.2 \pm 0.5$ | $2.9 \pm 0.4$ |
| Nemotron-Nano 9B v2 | 9B | ✓ | $16.9 \pm 0.9$ | $7.0 \pm 0.6$ | $4.5 \pm 0.5$ |
| SEED-OSS | 36B | ✓ | $41.9 \pm 1.3$ | $22.4 \pm 1.0$ | $14.2 \pm 0.8$ |
| Qwen-2.5 7B | 7B | | $5.6 \pm 0.6$ | $2.6 \pm 0.4$ | $1.7 \pm 0.3$ |
| Qwen-2.5 14B | 14B | | $7.1 \pm 0.7$ | $3.1 \pm 0.5$ | $2.6 \pm 0.5$ |
| Qwen-3 8B | 8B | ✓ | $17.3 \pm 0.9$ | $6.5 \pm 0.6$ | $4.0 \pm 0.5$ |
| Qwen-3 14B | 14B | ✓ | $19.7 \pm 1.0$ | $8.4 \pm 0.7$ | $4.9 \pm 0.5$ |
| Qwen-3 32B | 32B | ✓ | $22.5 \pm 1.0$ | $10.3 \pm 0.7$ | $6.6 \pm 0.6$ |
| Qwen-3 30B | 30B (A3B) | ✓ | $34.9 \pm 1.2$ | $15.0 \pm 0.9$ | $9.5 \pm 0.7$ |
| Qwen-3 Next 80B | 80B (A3B) | ✓ | $46.2 \pm 1.3$ | $23.1 \pm 1.1$ | $17.2 \pm 0.9$ |
| Qwen-3 235B | 235B (A22B) | ✓ | $53.1 \pm 1.3$ | $30.9 \pm 1.2$ | $23.5 \pm 1.1$ |
| GPT-OSS 20B (Low) | 20B (A4B) | ✓ | $16.0 \pm 0.9$ | $7.1 \pm 0.6$ | $4.9 \pm 0.5$ |
| GPT-OSS 20B (Med) | 20B (A4B) | ✓ | $39.3 \pm 1.2$ | $23.2 \pm 1.0$ | $12.7 \pm 0.8$ |
| GPT-OSS 20B (High) | 20B (A4B) | ✓ | $49.3 \pm 1.3$ | $33.6 \pm 1.2$ | $24.9 \pm 1.1$ |
| GPT-OSS 120B (Low) | 120B (A5B) | ✓ | $24.8 \pm 1.1$ | $10.8 \pm 0.8$ | $6.2 \pm 0.6$ |
| GPT-OSS 120B (Med) | 120B (A5B) | ✓ | $39.5 \pm 1.2$ | $22.3 \pm 1.0$ | $15.6 \pm 0.9$ |
| GPT-OSS 120B (High) | 120B (A5B) | ✓ | **$55.2 \pm 1.3$** | **$44.0 \pm 1.3$** | **$34.9 \pm 1.3$** |
| GLM-4.6 | 355B (A32B) | ✓ | $22.8 \pm 1.0$ | $10.6 \pm 0.7$ | $6.9 \pm 0.6$ |
| DeepSeek-R1-0528 | 671B (A37B) | ✓ | $50.9 \pm 1.2$ | $28.5 \pm 1.1$ | $20.5 \pm 1.0$ |

C.1.2 PERFORMANCE ACROSS MOLECULAR FEATURES

**Molecular feature groups**. We analyze performance across the six molecular-feature categories defined in Section 3.2. Table C4 presents single-task accuracy aggregated across feature groups. Tables C5, C6, and C7 provide the corresponding task-specific breakdowns for counting, indexing, and generation tasks, respectively, enabling direct comparison of which feature families pose the greatest challenge for each task type.

**Molecular features**. Figures C2–C4 provide fine-grained performance breakdowns for individual molecular features within each task category. Each heatmap displays the top-10 overall models alongside the average across all models, with tasks ordered by difficulty (average accuracy). These visualizations reveal which specific molecular properties remain challenging even for top-performing models.

Table C4: Single-task accuracy (in %) on MOLECULARIQ by chemical feature category: graph topology (GT), chemistry-typed topology (CT), composition (C), chemical perception (CP), functional groups (FG), and synthesis/fragmentation (SYN). Error bars indicate standard errors across instances. Model size refers to parameter count; R is a flag for reasoning models.

| | Size | R | GT | CT | C | CP | FG | SYN |
|---|---|---|---|---|---|---|---|---|
| | | | | | | **Chemical Features** | | |
| *Chemistry LLMs* | | | | | | | | |
| ChemLLM | 7B | | 1.9 ± 0.5 | 3.7 ± 0.5 | 0.9 ± 0.3 | 2.2 ± 0.6 | 4.8 ± 1.5 | 5.5 ± 1.1 |
| LlaSMol | 7B | | 2.4 ± 0.5 | 4.0 ± 0.5 | 0.7 ± 0.3 | 1.9 ± 0.5 | 5.2 ± 1.7 | 4.0 ± 0.9 |
| MolReasoner-Cap | 7B | ✓ | 5.6 ± 0.8 | 7.1 ± 0.7 | 5.0 ± 0.8 | 4.9 ± 1.0 | 7.0 ± 2.1 | 9.1 ± 1.4 |
| MolReasoner-Gen | 7B | ✓ | 4.0 ± 0.7 | 5.3 ± 0.6 | 4.2 ± 0.7 | 6.1 ± 1.0 | 10.0 ± 2.4 | 9.1 ± 1.4 |
| Llama-3 MolInst | 8B | | 3.4 ± 0.7 | 5.1 ± 0.7 | 2.2 ± 0.6 | 3.7 ± 0.8 | 2.6 ± 1.6 | 2.2 ± 0.7 |
| ChemDFM-8B | 8B | | 5.2 ± 0.8 | 8.0 ± 0.7 | 4.0 ± 0.8 | 5.3 ± 1.0 | 15.9 ± 3.4 | 13.7 ± 1.9 |
| ChemDFM-13B | 13B | | 4.9 ± 0.7 | 5.1 ± 0.5 | 2.7 ± 0.6 | 4.0 ± 0.8 | 11.1 ± 2.7 | 5.8 ± 1.1 |
| ChemDFM-R-14B | 14B | ✓ | 13.8 ± 1.6 | 17.3 ± 1.3 | 13.6 ± 1.6 | 14.0 ± 1.9 | 25.6 ± 4.6 | 24.1 ± 2.9 |
| TxGemma-9B | 9B | | 5.0 ± 0.8 | 5.5 ± 0.6 | 2.9 ± 0.7 | 4.4 ± 0.8 | 6.7 ± 2.0 | 8.9 ± 1.5 |
| TxGemma-27B | 27B | | 7.0 ± 0.9 | 11.0 ± 0.9 | 6.1 ± 0.9 | 9.2 ± 1.2 | 16.3 ± 3.3 | 15.2 ± 2.0 |
| Ether0 | 24B | ✓ | 9.8 ± 1.2 | 11.8 ± 0.9 | 14.4 ± 1.5 | 7.3 ± 1.2 | 14.1 ± 3.2 | 15.9 ± 2.1 |
| *Generalist LLMs* | | | | | | | | |
| Gemma-2 9B | 9B | | 5.2 ± 0.8 | 5.8 ± 0.6 | 6.1 ± 0.9 | 5.9 ± 1.0 | 10.7 ± 2.7 | 17.1 ± 2.1 |
| Gemma-2 27B | 27B | | 7.3 ± 1.1 | 13.9 ± 1.1 | 14.4 ± 1.4 | 10.2 ± 1.4 | 15.9 ± 3.5 | 23.8 ± 2.5 |
| Gemma-3 12B | 12B | | 7.3 ± 1.1 | 11.7 ± 1.0 | 8.7 ± 1.2 | 9.2 ± 1.5 | 14.8 ± 3.5 | 19.8 ± 2.5 |
| Gemma-3 27B | 27B | | 10.4 ± 1.4 | 16.6 ± 1.2 | 12.8 ± 1.5 | 10.0 ± 1.5 | 18.5 ± 3.9 | 28.0 ± 2.9 |
| LLaMA-2 13B | 13B | | 6.0 ± 0.9 | 5.3 ± 0.7 | 3.5 ± 0.7 | 4.7 ± 1.0 | 8.5 ± 2.8 | 17.4 ± 2.3 |
| LLaMA-3 8B | 8B | | 7.3 ± 1.0 | 9.0 ± 0.8 | 7.9 ± 1.0 | 8.6 ± 1.3 | 11.9 ± 3.2 | 18.9 ± 2.3 |
| LLaMA-3.3 70B | 70B | | 14.5 ± 1.6 | 18.9 ± 1.3 | 16.9 ± 1.7 | 10.6 ± 1.6 | 24.8 ± 4.5 | 23.1 ± 2.7 |
| Mistral-7B v0.1 | 7B | | 2.5 ± 0.5 | 2.9 ± 0.4 | 1.4 ± 0.3 | 1.9 ± 0.5 | 1.1 ± 0.6 | 4.0 ± 0.8 |
| Mistral-Small | 24B | | 11.3 ± 1.3 | 17.2 ± 1.1 | 15.0 ± 1.4 | 10.8 ± 1.4 | 23.0 ± 4.1 | 26.6 ± 2.7 |
| Nemotron-Nano 9B v2 | 9B | ✓ | 15.1 ± 1.4 | 20.4 ± 1.2 | 28.9 ± 1.7 | 13.4 ± 1.5 | 26.3 ± 4.0 | 26.3 ± 2.7 |
| SEED-OSS | 36B | ✓ | 35.3 ± 1.8 | 37.5 ± 1.5 | 52.1 ± 2.0 | 26.8 ± 2.1 | 45.6 ± 4.7 | 38.4 ± 2.8 |
| Qwen-2.5 7B | 7B | | 6.8 ± 1.0 | 9.2 ± 0.9 | 6.4 ± 1.0 | 7.6 ± 1.2 | 8.5 ± 2.9 | 19.5 ± 2.4 |
| Qwen-2.5 14B | 14B | | 10.6 ± 1.4 | 12.4 ± 1.1 | 8.7 ± 1.3 | 9.6 ± 1.6 | 20.0 ± 4.2 | 26.3 ± 2.9 |
| Qwen-3 8B | 8B | ✓ | 14.6 ± 1.5 | 19.5 ± 1.2 | 25.8 ± 1.7 | 14.9 ± 1.6 | 26.7 ± 4.0 | 30.2 ± 2.8 |
| Qwen-3 14B | 14B | ✓ | 18.4 ± 1.6 | 23.3 ± 1.3 | 29.1 ± 1.8 | 17.2 ± 1.6 | 30.4 ± 4.3 | 30.8 ± 2.7 |
| Qwen-3 32B | 32B | ✓ | 22.8 ± 1.7 | 26.7 ± 1.3 | 35.4 ± 1.8 | 20.2 ± 1.8 | 38.1 ± 4.7 | 26.9 ± 2.5 |
| Qwen-3 30B | 30B (A3B) | ✓ | 28.9 ± 1.8 | 33.8 ± 1.4 | 49.4 ± 1.9 | 30.4 ± 2.1 | 40.4 ± 4.6 | 36.8 ± 2.9 |
| Qwen-3 Next 80B | 80B (A3B) | ✓ | 43.7 ± 2.0 | 44.2 ± 1.5 | 62.8 ± 1.9 | 36.1 ± 2.2 | 46.7 ± 4.7 | 44.2 ± 3.0 |
| Qwen-3 235B | 235B (A22B) | ✓ | 46.7 ± 2.0 | 54.6 ± 1.5 | 74.1 ± 1.7 | 48.9 ± 2.4 | 51.5 ± 4.9 | **47.8 ± 2.9** |
| GPT-OSS 20B (Low) | 20B (A4B) | ✓ | 13.8 ± 1.4 | 23.2 ± 1.3 | 17.5 ± 1.5 | 17.0 ± 1.7 | 27.4 ± 3.9 | 25.1 ± 2.6 |
| GPT-OSS 20B (Med) | 20B (A4B) | ✓ | 32.6 ± 1.9 | 39.3 ± 1.4 | 57.1 ± 1.8 | 44.1 ± 2.1 | 49.3 ± 4.4 | 38.1 ± 2.7 |
| GPT-OSS 20B (High) | 20B (A4B) | ✓ | 39.8 ± 2.0 | 49.3 ± 1.5 | 69.1 ± 1.7 | 53.2 ± 2.2 | 58.1 ± 4.7 | 40.6 ± 2.9 |
| GPT-OSS 120B (Low) | 120B (A5B) | ✓ | 19.9 ± 1.7 | 28.3 ± 1.4 | 35.0 ± 1.9 | 23.7 ± 1.9 | 39.6 ± 4.6 | 29.3 ± 2.8 |
| GPT-OSS 120B (Med) | 120B (A5B) | ✓ | 31.2 ± 1.8 | 38.2 ± 1.5 | 57.3 ± 1.9 | 38.8 ± 2.1 | 47.4 ± 4.7 | 39.6 ± 2.8 |
| GPT-OSS 120B (High) | 120B (A5B) | ✓ | 45.5 ± 2.1 | **59.7 ± 1.5** | **83.0 ± 1.4** | **68.1 ± 2.1** | **61.1 ± 4.8** | 47.6 ± 3.2 |
| GLM-4.6 | 355B (A32B) | ✓ | 22.5 ± 1.7 | 26.4 ± 1.3 | 35.6 ± 1.8 | 20.8 ± 1.8 | 38.1 ± 4.4 | 31.4 ± 2.7 |
| DeepSeek-R1-0528 | 671B (A37B) | ✓ | **47.2 ± 2.0** | 49.4 ± 1.5 | 70.3 ± 1.7 | 49.7 ± 2.3 | 55.6 ± 4.8 | 43.5 ± 2.9 |

Table C5: Single-task counting accuracy (in %) on MOLECULARIQ by chemical feature category: graph topology (GT), chemistry-typed topology (CT), composition (C), chemical perception (CP), functional groups (FG), and synthesis/fragmentation (SYN). Error bars indicate standard errors across instances. Model size refers to parameter count; R is a flag for reasoning models.

| | Size | R | GT | CT | C | CP | FG | SYN |
|---|---|---|---|---|---|---|---|---|
| | | | | | **Chemical Features** | | | |
| *Chemistry LLMs* | | | | | | | | |
| ChemLLM | 7B | | $3.1 \pm 0.9$ | $5.9 \pm 0.9$ | $1.1 \pm 0.6$ | $2.8 \pm 0.9$ | $4.4 \pm 2.6$ | $1.1 \pm 0.8$ |
| LlaSMol | 7B | | $3.3 \pm 0.8$ | $4.1 \pm 0.7$ | $1.5 \pm 0.6$ | $2.8 \pm 1.2$ | $4.4 \pm 2.6$ | $1.7 \pm 0.9$ |
| MolReasoner-Cap | 7B | ✓ | $10.2 \pm 1.8$ | $12.0 \pm 1.5$ | $7.8 \pm 1.7$ | $8.1 \pm 2.2$ | $12.2 \pm 4.7$ | $3.3 \pm 1.5$ |
| MolReasoner-Gen | 7B | ✓ | $7.2 \pm 1.4$ | $9.8 \pm 1.3$ | $7.6 \pm 1.6$ | $10.0 \pm 2.1$ | $10.0 \pm 4.3$ | $4.4 \pm 1.7$ |
| Llama-3 MolInst | 8B | | $7.2 \pm 1.5$ | $12.1 \pm 1.6$ | $4.8 \pm 1.3$ | $8.6 \pm 2.1$ | $6.7 \pm 4.6$ | $3.3 \pm 1.7$ |
| ChemDFM-8B | 8B | | $8.1 \pm 1.5$ | $13.9 \pm 1.5$ | $5.0 \pm 1.4$ | $6.7 \pm 1.9$ | $16.7 \pm 6.5$ | $2.2 \pm 1.1$ |
| ChemDFM-13B | 13B | | $9.1 \pm 1.4$ | $9.6 \pm 1.2$ | $4.1 \pm 1.1$ | $7.5 \pm 1.8$ | $7.8 \pm 3.8$ | $2.2 \pm 1.1$ |
| ChemDFM-R-14B | 14B | ✓ | $21.7 \pm 3.1$ | $23.6 \pm 2.3$ | $20.6 \pm 3.0$ | $24.2 \pm 3.9$ | $20.0 \pm 7.4$ | $11.7 \pm 4.2$ |
| TxGemma-9B | 9B | | $9.6 \pm 1.8$ | $9.8 \pm 1.3$ | $5.2 \pm 1.5$ | $6.4 \pm 1.7$ | $6.7 \pm 3.4$ | $3.3 \pm 1.9$ |
| TxGemma-27B | 27B | | $10.6 \pm 1.7$ | $14.7 \pm 1.5$ | $8.9 \pm 1.8$ | $14.2 \pm 2.4$ | $15.6 \pm 5.7$ | $5.0 \pm 1.9$ |
| Ether0 | 24B | ✓ | $6.9 \pm 1.5$ | $11.0 \pm 1.3$ | $2.8 \pm 0.8$ | $1.9 \pm 0.8$ | $0.0 \pm 0.0$ | $1.1 \pm 0.8$ |
| *Generalist LLMs* | | | | | | | | |
| Gemma-2 9B | 9B | | $8.1 \pm 1.4$ | $7.5 \pm 1.0$ | $5.6 \pm 1.2$ | $7.8 \pm 1.6$ | $6.7 \pm 3.7$ | $4.4 \pm 1.9$ |
| Gemma-2 27B | 27B | | $10.2 \pm 1.9$ | $12.5 \pm 1.6$ | $11.7 \pm 2.0$ | $12.8 \pm 2.5$ | $8.9 \pm 5.0$ | $5.0 \pm 1.9$ |
| Gemma-3 12B | 12B | | $11.7 \pm 2.2$ | $16.4 \pm 1.9$ | $9.4 \pm 2.1$ | $11.7 \pm 2.8$ | $18.9 \pm 6.7$ | $4.4 \pm 2.2$ |
| Gemma-3 27B | 27B | | $16.7 \pm 2.6$ | $19.9 \pm 2.1$ | $14.4 \pm 2.4$ | $15.0 \pm 2.9$ | $20.0 \pm 6.7$ | $8.3 \pm 2.6$ |
| LLaMA-2 13B | 13B | | $10.9 \pm 1.8$ | $9.1 \pm 1.3$ | $3.7 \pm 1.3$ | $6.7 \pm 1.8$ | $5.6 \pm 3.9$ | $4.4 \pm 1.7$ |
| LLaMA-3 8B | 8B | | $10.4 \pm 1.9$ | $14.0 \pm 1.6$ | $10.0 \pm 1.9$ | $10.6 \pm 2.3$ | $6.7 \pm 4.6$ | $5.6 \pm 1.8$ |
| LLaMA-3.3 70B | 70B | | $19.8 \pm 2.8$ | $24.0 \pm 2.3$ | $15.6 \pm 2.7$ | $16.1 \pm 3.2$ | $22.2 \pm 7.6$ | $6.1 \pm 2.7$ |
| Mistral-7B v0.1 | 7B | | $4.1 \pm 1.0$ | $5.1 \pm 0.8$ | $1.5 \pm 0.5$ | $4.2 \pm 1.1$ | $1.1 \pm 1.1$ | $2.2 \pm 1.1$ |
| Mistral-Small | 24B | | $14.8 \pm 2.2$ | $18.8 \pm 1.8$ | $16.5 \pm 2.3$ | $12.8 \pm 2.5$ | $23.3 \pm 7.5$ | $7.2 \pm 2.1$ |
| Nemotron-Nano 9B v2 | 9B | ✓ | $18.3 \pm 2.4$ | $20.8 \pm 1.9$ | $28.5 \pm 2.8$ | $17.5 \pm 2.8$ | $28.9 \pm 6.9$ | $8.3 \pm 2.8$ |
| SEED-OSS | 36B | ✓ | $32.6 \pm 2.8$ | $37.3 \pm 2.4$ | $58.9 \pm 3.0$ | $30.8 \pm 3.7$ | $43.3 \pm 8.6$ | $18.9 \pm 3.6$ |
| Qwen-2.5 7B | 7B | | $9.8 \pm 1.9$ | $13.7 \pm 1.7$ | $8.9 \pm 1.8$ | $11.4 \pm 2.4$ | $11.1 \pm 5.6$ | $2.2 \pm 1.1$ |
| Qwen-2.5 14B | 14B | | $15.0 \pm 2.3$ | $15.8 \pm 2.0$ | $9.4 \pm 2.2$ | $10.8 \pm 2.8$ | $16.7 \pm 6.9$ | $3.3 \pm 2.3$ |
| Qwen-3 8B | 8B | ✓ | $16.1 \pm 2.4$ | $21.5 \pm 2.0$ | $28.3 \pm 2.9$ | $18.6 \pm 2.8$ | $30.0 \pm 7.2$ | $11.1 \pm 3.3$ |
| Qwen-3 14B | 14B | ✓ | $21.1 \pm 2.7$ | $23.7 \pm 2.1$ | $32.4 \pm 3.0$ | $18.1 \pm 2.7$ | $32.2 \pm 7.7$ | $10.0 \pm 2.8$ |
| Qwen-3 32B | 32B | ✓ | $24.6 \pm 2.6$ | $27.8 \pm 2.1$ | $35.4 \pm 3.0$ | $20.0 \pm 3.0$ | $36.7 \pm 8.5$ | $10.6 \pm 2.6$ |
| Qwen-3 30B | 30B (A3B) | ✓ | $28.1 \pm 2.9$ | $32.8 \pm 2.3$ | $52.4 \pm 3.1$ | $28.6 \pm 3.4$ | $25.6 \pm 7.1$ | $22.2 \pm 4.5$ |
| Qwen-3 Next 80B | 80B (A3B) | ✓ | $41.3 \pm 3.2$ | $41.0 \pm 2.4$ | $61.5 \pm 3.1$ | $32.2 \pm 3.7$ | $37.8 \pm 7.9$ | $22.2 \pm 4.4$ |
| Qwen-3 235B | 235B (A22B) | ✓ | $41.3 \pm 3.2$ | $49.5 \pm 2.5$ | $69.8 \pm 2.8$ | $44.7 \pm 4.1$ | $38.9 \pm 8.6$ | $\mathbf{29.4 \pm 4.9}$ |
| GPT-OSS 20B (Low) | 20B (A4B) | ✓ | $18.0 \pm 2.4$ | $26.5 \pm 2.1$ | $20.0 \pm 2.5$ | $21.9 \pm 3.2$ | $26.7 \pm 6.9$ | $6.7 \pm 1.9$ |
| GPT-OSS 20B (Med) | 20B (A4B) | ✓ | $33.0 \pm 3.0$ | $36.2 \pm 2.2$ | $60.4 \pm 2.8$ | $47.8 \pm 3.5$ | $46.7 \pm 7.4$ | $18.9 \pm 3.4$ |
| GPT-OSS 20B (High) | 20B (A4B) | ✓ | $38.0 \pm 3.1$ | $47.6 \pm 2.3$ | $76.5 \pm 2.6$ | $58.1 \pm 3.6$ | $\mathbf{53.3 \pm 8.7}$ | $21.1 \pm 4.3$ |
| GPT-OSS 120B (Low) | 120B (A5B) | ✓ | $23.9 \pm 2.8$ | $27.0 \pm 2.2$ | $32.6 \pm 3.0$ | $31.1 \pm 3.4$ | $36.7 \pm 7.7$ | $10.0 \pm 3.1$ |
| GPT-OSS 120B (Med) | 120B (A5B) | ✓ | $33.3 \pm 3.0$ | $34.5 \pm 2.3$ | $57.8 \pm 3.1$ | $45.6 \pm 3.6$ | $53.3 \pm 8.5$ | $22.2 \pm 3.9$ |
| GPT-OSS 120B (High) | 120B (A5B) | ✓ | $41.7 \pm 3.5$ | $\mathbf{55.1 \pm 2.4}$ | $\mathbf{87.0 \pm 2.1}$ | $67.2 \pm 3.6$ | $50.0 \pm 8.7$ | $28.9 \pm 5.5$ |
| GLM-4.6 | 355B (A32B) | ✓ | $23.9 \pm 2.7$ | $24.7 \pm 2.0$ | $33.9 \pm 2.8$ | $18.9 \pm 2.8$ | $31.1 \pm 6.8$ | $9.4 \pm 2.5$ |
| DeepSeek-R1-0528 | 671B (A37B) | ✓ | $\mathbf{44.4 \pm 3.2}$ | $45.4 \pm 2.4$ | $71.1 \pm 2.7$ | $48.6 \pm 3.7$ | $43.3 \pm 8.5$ | $28.9 \pm 5.0$ |

Table C6: Single-task indexing accuracy (in %) on MOLECULARIQ by chemical feature category: graph topology (GT), chemistry-typed topology (CT), composition (C), chemical perception (CP), functional groups (FG), and synthesis/fragmentation (SYN). Error bars indicate standard errors across instances. Model size refers to parameter count; R is a flag for reasoning models.

| | Size | R | Chemical Features | | | | | |
| | | | GT | CT | C | CP | FG | SYN |
|---|---|---|---|---|---|---|---|---|
| *Chemistry LLMs* | | | | | | | | |
| ChemLLM | 7B | | $0.2 \pm 0.2$ | $1.4 \pm 0.4$ | $0.3 \pm 0.3$ | $0.0 \pm 0.0$ | $2.2 \pm 1.5$ | $0.0 \pm 0.0$ |
| LlaSMol | 7B | | $2.2 \pm 0.9$ | $4.9 \pm 0.9$ | $0.0 \pm 0.0$ | $0.8 \pm 0.6$ | $5.6 \pm 2.8$ | $0.0 \pm 0.0$ |
| MolReasoner-Cap | 7B | ✓ | $0.2 \pm 0.2$ | $1.0 \pm 0.3$ | $0.8 \pm 0.6$ | $0.3 \pm 0.3$ | $0.0 \pm 0.0$ | $0.6 \pm 0.6$ |
| MolReasoner-Gen | 7B | ✓ | $0.6 \pm 0.4$ | $1.2 \pm 0.4$ | $1.1 \pm 0.7$ | $1.1 \pm 0.8$ | $2.2 \pm 2.2$ | $0.0 \pm 0.0$ |
| Llama-3 MolInst | 8B | | $0.0 \pm 0.0$ | $0.2 \pm 0.1$ | $0.0 \pm 0.0$ | $0.0 \pm 0.0$ | $1.1 \pm 1.1$ | $0.0 \pm 0.0$ |
| ChemDFM-8B | 8B | | $0.4 \pm 0.4$ | $0.0 \pm 0.0$ | $0.0 \pm 0.0$ | $0.3 \pm 0.3$ | $1.1 \pm 1.1$ | $0.0 \pm 0.0$ |
| ChemDFM-13B | 13B | | $0.2 \pm 0.2$ | $0.7 \pm 0.3$ | $0.6 \pm 0.4$ | $0.3 \pm 0.3$ | $2.2 \pm 1.5$ | $0.0 \pm 0.0$ |
| ChemDFM-R-14B | 14B | ✓ | $3.3 \pm 1.3$ | $9.1 \pm 1.6$ | $1.7 \pm 1.2$ | $0.8 \pm 0.8$ | $16.7 \pm 6.9$ | $0.0 \pm 0.0$ |
| TxGemma-9B | 9B | | $0.9 \pm 0.5$ | $0.4 \pm 0.2$ | $0.0 \pm 0.0$ | $0.3 \pm 0.3$ | $2.2 \pm 2.2$ | $0.6 \pm 0.6$ |
| TxGemma-27B | 27B | | $1.9 \pm 0.7$ | $5.7 \pm 1.1$ | $1.7 \pm 0.9$ | $1.4 \pm 0.8$ | $7.8 \pm 3.5$ | $0.6 \pm 0.6$ |
| Ether0 | 24B | ✓ | $0.0 \pm 0.0$ | $0.2 \pm 0.1$ | $0.0 \pm 0.0$ | $0.3 \pm 0.3$ | $0.0 \pm 0.0$ | $0.0 \pm 0.0$ |
| *Generalist LLMs* | | | | | | | | |
| Gemma-2 9B | 9B | | $0.2 \pm 0.2$ | $1.4 \pm 0.5$ | $1.4 \pm 0.6$ | $0.3 \pm 0.3$ | $7.8 \pm 4.1$ | $0.6 \pm 0.6$ |
| Gemma-2 27B | 27B | | $0.4 \pm 0.3$ | $2.2 \pm 0.6$ | $3.1 \pm 1.3$ | $1.1 \pm 0.9$ | $5.6 \pm 3.9$ | $0.6 \pm 0.6$ |
| Gemma-3 12B | 12B | | $0.0 \pm 0.0$ | $2.0 \pm 0.7$ | $3.1 \pm 1.4$ | $2.5 \pm 1.4$ | $4.4 \pm 3.5$ | $0.0 \pm 0.0$ |
| Gemma-3 27B | 27B | | $0.6 \pm 0.6$ | $4.2 \pm 1.1$ | $2.2 \pm 1.0$ | $0.3 \pm 0.3$ | $6.7 \pm 4.6$ | $0.0 \pm 0.0$ |
| LLaMA-2 13B | 13B | | $0.2 \pm 0.2$ | $0.0 \pm 0.0$ | $0.6 \pm 0.4$ | $0.6 \pm 0.6$ | $2.2 \pm 2.2$ | $0.0 \pm 0.0$ |
| LLaMA-3 8B | 8B | | $0.2 \pm 0.2$ | $0.3 \pm 0.2$ | $2.2 \pm 0.8$ | $0.0 \pm 0.0$ | $1.1 \pm 1.1$ | $0.0 \pm 0.0$ |
| LLaMA-3.3 70B | 70B | | $1.1 \pm 0.8$ | $4.1 \pm 1.1$ | $5.0 \pm 2.0$ | $1.7 \pm 1.2$ | $10.0 \pm 5.6$ | $0.6 \pm 0.6$ |
| Mistral-7B v0.1 | 7B | | $0.9 \pm 0.5$ | $1.3 \pm 0.4$ | $0.0 \pm 0.0$ | $0.0 \pm 0.0$ | $0.0 \pm 0.0$ | $0.0 \pm 0.0$ |
| Mistral-Small | 24B | | $1.5 \pm 0.8$ | $3.9 \pm 1.0$ | $4.2 \pm 1.4$ | $2.2 \pm 1.2$ | $8.9 \pm 5.0$ | $0.0 \pm 0.0$ |
| Nemotron-Nano 9B v2 | 9B | ✓ | $5.7 \pm 1.5$ | $10.7 \pm 1.5$ | $28.1 \pm 3.2$ | $6.9 \pm 2.0$ | $24.4 \pm 7.0$ | $1.7 \pm 0.9$ |
| SEED-OSS | 36B | ✓ | $31.5 \pm 2.8$ | $36.7 \pm 2.3$ | $68.1 \pm 3.2$ | $30.3 \pm 3.6$ | $45.6 \pm 8.4$ | $10.0 \pm 3.0$ |
| Qwen-2.5 7B | 7B | | $0.0 \pm 0.0$ | $1.3 \pm 0.5$ | $0.6 \pm 0.4$ | $1.4 \pm 0.9$ | $3.3 \pm 3.3$ | $0.0 \pm 0.0$ |
| Qwen-2.5 14B | 14B | | $0.0 \pm 0.0$ | $2.4 \pm 0.8$ | $0.8 \pm 0.8$ | $1.7 \pm 1.2$ | $10.0 \pm 5.6$ | $0.0 \pm 0.0$ |
| Qwen-3 8B | 8B | ✓ | $2.6 \pm 0.9$ | $9.2 \pm 1.4$ | $24.7 \pm 3.2$ | $7.2 \pm 2.1$ | $15.6 \pm 5.9$ | $1.7 \pm 1.2$ |
| Qwen-3 14B | 14B | ✓ | $7.6 \pm 1.8$ | $11.8 \pm 1.6$ | $28.1 \pm 3.3$ | $11.4 \pm 2.4$ | $22.2 \pm 7.0$ | $1.7 \pm 0.9$ |
| Qwen-3 32B | 32B | ✓ | $8.9 \pm 1.9$ | $15.1 \pm 1.7$ | $31.4 \pm 3.3$ | $12.5 \pm 2.6$ | $24.4 \pm 7.3$ | $4.4 \pm 2.2$ |
| Qwen-3 30B | 30B (A3B) | ✓ | $16.3 \pm 2.3$ | $25.9 \pm 2.1$ | $40.8 \pm 3.6$ | $23.6 \pm 3.4$ | $34.4 \pm 7.7$ | $7.2 \pm 2.9$ |
| Qwen-3 Next 80B | 80B (A3B) | ✓ | $33.7 \pm 3.2$ | $36.3 \pm 2.4$ | $63.9 \pm 3.5$ | $33.6 \pm 3.7$ | $36.7 \pm 7.9$ | $20.0 \pm 4.3$ |
| Qwen-3 235B | 235B (A22B) | ✓ | $41.1 \pm 3.0$ | $47.5 \pm 2.4$ | $81.9 \pm 2.6$ | $40.3 \pm 3.9$ | $41.1 \pm 8.3$ | $26.1 \pm 5.1$ |
| GPT-OSS 20B (Low) | 20B (A4B) | ✓ | $4.3 \pm 1.3$ | $13.7 \pm 1.7$ | $13.3 \pm 2.4$ | $7.8 \pm 1.9$ | $13.3 \pm 5.2$ | $0.0 \pm 0.0$ |
| GPT-OSS 20B (Med) | 20B (A4B) | ✓ | $24.8 \pm 2.8$ | $35.8 \pm 2.2$ | $56.1 \pm 3.3$ | $36.7 \pm 3.7$ | $41.1 \pm 8.3$ | $11.7 \pm 3.1$ |
| GPT-OSS 20B (High) | 20B (A4B) | ✓ | $35.2 \pm 3.1$ | $47.9 \pm 2.3$ | $70.8 \pm 3.3$ | $46.9 \pm 3.8$ | $\mathbf{51.1 \pm 8.6}$ | $19.4 \pm 4.5$ |
| GPT-OSS 120B (Low) | 120B (A5B) | ✓ | $9.1 \pm 1.9$ | $19.6 \pm 2.0$ | $30.6 \pm 3.3$ | $14.7 \pm 2.8$ | $27.8 \pm 7.8$ | $5.0 \pm 2.2$ |
| GPT-OSS 120B (Med) | 120B (A5B) | ✓ | $23.7 \pm 2.6$ | $33.0 \pm 2.1$ | $66.4 \pm 3.4$ | $33.9 \pm 3.6$ | $30.0 \pm 7.6$ | $14.4 \pm 3.7$ |
| GPT-OSS 120B (High) | 120B (A5B) | ✓ | $\mathbf{44.8 \pm 3.4}$ | $\mathbf{57.9 \pm 2.4}$ | $\mathbf{85.8 \pm 2.8}$ | $\mathbf{64.2 \pm 3.8}$ | $51.1 \pm 8.4$ | $\mathbf{28.3 \pm 5.5}$ |
| GLM-4.6 | 355B (A32B) | ✓ | $10.4 \pm 1.9$ | $16.4 \pm 1.7$ | $38.6 \pm 3.3$ | $10.8 \pm 2.3$ | $18.9 \pm 6.5$ | $4.4 \pm 1.9$ |
| DeepSeek-R1-0528 | 671B (A37B) | ✓ | $38.1 \pm 3.0$ | $41.2 \pm 2.3$ | $69.4 \pm 3.1$ | $38.9 \pm 3.9$ | $40.0 \pm 8.2$ | $17.2 \pm 3.9$ |

Table C7: Single-task constrained-generation accuracy (in %) on MOLECULARIQ by chemical feature category: graph topology (GT), chemistry-typed topology (CT), composition (C), chemical perception (CP), functional groups (FG), and synthesis/fragmentation (SYN). Error bars indicate standard errors across instances. Model size refers to parameter count; R is a flag for reasoning models.

| | Size | R | **Chemical Features** | | | | | |
| | | | GT | CT | C | CP | FG | SYN |
|---|---|---|---|---|---|---|---|---|
| *Chemistry LLMs* | | | | | | | | |
| ChemLLM | 7B | | $2.9 \pm 1.2$ | $3.8 \pm 1.1$ | $1.0 \pm 0.5$ | $4.2 \pm 1.7$ | $7.8 \pm 3.5$ | $11.2 \pm 2.1$ |
| LlaSMol | 7B | | $1.3 \pm 0.6$ | $1.9 \pm 0.6$ | $0.4 \pm 0.3$ | $2.1 \pm 0.8$ | $5.6 \pm 3.2$ | $7.7 \pm 1.8$ |
| MolReasoner-Cap | 7B | ✓ | $7.1 \pm 1.8$ | $9.4 \pm 1.7$ | $4.9 \pm 1.2$ | $6.7 \pm 2.2$ | $8.9 \pm 4.2$ | $17.3 \pm 2.8$ |
| MolReasoner-Gen | 7B | ✓ | $4.5 \pm 1.3$ | $4.5 \pm 1.0$ | $2.7 \pm 0.9$ | $7.4 \pm 1.7$ | $17.8 \pm 5.2$ | $17.0 \pm 2.7$ |
| Llama-3 MolInst | 8B | | $2.6 \pm 1.4$ | $1.1 \pm 0.6$ | $0.8 \pm 0.6$ | $2.1 \pm 1.1$ | $0.0 \pm 0.0$ | $2.9 \pm 1.2$ |
| ChemDFM-8B | 8B | | $8.4 \pm 2.1$ | $12.1 \pm 1.8$ | $5.8 \pm 1.5$ | $9.8 \pm 2.3$ | $30.0 \pm 7.0$ | $28.2 \pm 3.5$ |
| ChemDFM-13B | 13B | | $5.8 \pm 1.6$ | $5.1 \pm 1.1$ | $2.7 \pm 0.9$ | $4.2 \pm 1.2$ | $23.3 \pm 6.4$ | $11.2 \pm 2.2$ |
| ChemDFM-R-14B | 14B | ✓ | $18.4 \pm 3.8$ | $20.9 \pm 3.1$ | $14.8 \pm 2.8$ | $17.9 \pm 4.0$ | $40.0 \pm 9.1$ | $45.2 \pm 4.9$ |
| TxGemma-9B | 9B | | $4.2 \pm 1.4$ | $7.0 \pm 1.4$ | $2.5 \pm 0.9$ | $7.0 \pm 1.9$ | $11.1 \pm 4.3$ | $17.0 \pm 2.8$ |
| TxGemma-27B | 27B | | $9.7 \pm 2.3$ | $13.7 \pm 2.1$ | $6.2 \pm 1.4$ | $12.6 \pm 2.6$ | $25.6 \pm 6.9$ | $29.5 \pm 3.8$ |
| Ether0 | 24B | ✓ | $32.0 \pm 4.0$ | $34.8 \pm 3.1$ | $37.9 \pm 3.5$ | $22.8 \pm 3.6$ | $42.2 \pm 7.5$ | $33.7 \pm 3.7$ |
| *Generalist LLMs* | | | | | | | | |
| Gemma-2 9B | 9B | | $8.7 \pm 2.1$ | $10.7 \pm 1.9$ | $10.1 \pm 2.0$ | $10.5 \pm 2.6$ | $17.8 \pm 5.7$ | $34.0 \pm 3.8$ |
| Gemma-2 27B | 27B | | $14.6 \pm 3.1$ | $38.2 \pm 3.3$ | $25.9 \pm 2.9$ | $18.6 \pm 3.3$ | $33.3 \pm 7.5$ | $48.1 \pm 4.2$ |
| Gemma-3 12B | 12B | | $12.3 \pm 3.0$ | $20.9 \pm 2.8$ | $12.1 \pm 2.3$ | $14.4 \pm 3.2$ | $21.1 \pm 6.9$ | $40.1 \pm 4.4$ |
| Gemma-3 27B | 27B | | $16.5 \pm 3.6$ | $33.5 \pm 3.5$ | $18.9 \pm 2.9$ | $16.1 \pm 3.5$ | $28.9 \pm 7.9$ | $55.4 \pm 4.7$ |
| LLaMA-2 13B | 13B | | $7.4 \pm 2.4$ | $8.1 \pm 1.9$ | $5.6 \pm 1.5$ | $7.4 \pm 2.5$ | $17.8 \pm 6.9$ | $34.9 \pm 4.1$ |
| LLaMA-3 8B | 8B | | $14.2 \pm 3.1$ | $16.0 \pm 2.3$ | $9.7 \pm 1.9$ | $16.8 \pm 3.3$ | $27.8 \pm 7.7$ | $37.5 \pm 4.3$ |
| LLaMA-3.3 70B | 70B | | $28.8 \pm 4.3$ | $36.7 \pm 3.3$ | $27.2 \pm 3.3$ | $15.1 \pm 3.4$ | $42.2 \pm 8.9$ | $45.8 \pm 4.7$ |
| Mistral-7B v0.1 | 7B | | $2.6 \pm 1.2$ | $1.7 \pm 0.6$ | $2.3 \pm 0.8$ | $1.4 \pm 0.9$ | $2.2 \pm 1.5$ | $7.4 \pm 1.6$ |
| Mistral-Small | 24B | | $22.3 \pm 3.9$ | $39.0 \pm 3.2$ | $21.4 \pm 2.8$ | $19.3 \pm 3.3$ | $36.7 \pm 7.9$ | $53.2 \pm 4.5$ |
| Nemotron-Nano 9B v2 | 9B | ✓ | $25.9 \pm 3.7$ | $37.5 \pm 3.2$ | $30.0 \pm 3.0$ | $16.5 \pm 2.9$ | $25.6 \pm 6.9$ | $51.0 \pm 4.4$ |
| SEED-OSS | 36B | ✓ | $46.9 \pm 4.0$ | $39.5 \pm 3.1$ | $32.7 \pm 3.2$ | $17.2 \pm 3.3$ | $47.8 \pm 7.6$ | $66.0 \pm 4.0$ |
| Qwen-2.5 7B | 7B | | $13.6 \pm 3.0$ | $15.6 \pm 2.4$ | $8.0 \pm 1.9$ | $10.5 \pm 2.4$ | $11.1 \pm 5.6$ | $40.7 \pm 4.2$ |
| Qwen-2.5 14B | 14B | | $21.4 \pm 4.1$ | $24.9 \pm 3.3$ | $13.6 \pm 2.7$ | $17.9 \pm 4.0$ | $33.3 \pm 8.8$ | $54.8 \pm 4.9$ |
| Qwen-3 8B | 8B | ✓ | $33.0 \pm 4.2$ | $34.8 \pm 3.1$ | $23.7 \pm 2.7$ | $20.0 \pm 3.3$ | $34.4 \pm 7.2$ | $57.7 \pm 4.4$ |
| Qwen-3 14B | 14B | ✓ | $32.7 \pm 4.2$ | $44.1 \pm 3.3$ | $26.1 \pm 2.8$ | $23.5 \pm 3.5$ | $36.7 \pm 7.6$ | $59.6 \pm 4.1$ |
| Qwen-3 32B | 32B | ✓ | $43.7 \pm 4.2$ | $46.3 \pm 3.0$ | $38.5 \pm 3.0$ | $30.2 \pm 3.8$ | $53.3 \pm 8.1$ | $49.4 \pm 4.1$ |
| Qwen-3 30B | 30B (A3B) | ✓ | $52.4 \pm 4.3$ | $50.5 \pm 3.2$ | $52.3 \pm 3.3$ | $41.4 \pm 4.0$ | $61.1 \pm 8.0$ | $62.2 \pm 4.2$ |
| Qwen-3 Next 80B | 80B (A3B) | ✓ | $65.4 \pm 4.1$ | $65.2 \pm 3.3$ | $63.4 \pm 3.3$ | $44.2 \pm 4.2$ | $65.6 \pm 7.6$ | $\mathbf{70.8 \pm 4.0}$ |
| Qwen-3 235B | 235B (A22B) | ✓ | $65.7 \pm 3.9$ | $\mathbf{77.6 \pm 2.5}$ | $73.0 \pm 3.0$ | $64.9 \pm 4.2$ | $74.4 \pm 7.1$ | $\mathbf{70.8 \pm 3.6}$ |
| GPT-OSS 20B (Low) | 20B (A4B) | ✓ | $23.0 \pm 3.7$ | $34.8 \pm 3.2$ | $17.9 \pm 2.7$ | $22.5 \pm 3.5$ | $42.2 \pm 7.1$ | $50.3 \pm 4.4$ |
| GPT-OSS 20B (Med) | 20B (A4B) | ✓ | $45.6 \pm 4.1$ | $52.0 \pm 3.2$ | $54.1 \pm 3.2$ | $48.8 \pm 3.6$ | $60.0 \pm 6.7$ | $64.4 \pm 3.7$ |
| GPT-OSS 20B (High) | 20B (A4B) | ✓ | $51.1 \pm 4.2$ | $55.4 \pm 3.2$ | $59.7 \pm 3.0$ | $55.1 \pm 3.9$ | $70.0 \pm 7.0$ | $64.1 \pm 3.9$ |
| GPT-OSS 120B (Low) | 120B (A5B) | ✓ | $31.7 \pm 4.1$ | $46.9 \pm 3.5$ | $40.9 \pm 3.3$ | $25.6 \pm 3.7$ | $54.4 \pm 7.6$ | $54.5 \pm 4.6$ |
| GPT-OSS 120B (Med) | 120B (A5B) | ✓ | $40.5 \pm 4.2$ | $54.8 \pm 3.2$ | $50.0 \pm 3.1$ | $36.5 \pm 3.6$ | $58.9 \pm 7.4$ | $64.1 \pm 4.1$ |
| GPT-OSS 120B (High) | 120B (A5B) | ✓ | $53.4 \pm 4.5$ | $71.6 \pm 2.8$ | $\mathbf{76.3 \pm 2.5}$ | $\mathbf{74.0 \pm 3.7}$ | $82.2 \pm 6.3$ | $69.6 \pm 4.2$ |
| GLM-4.6 | 355B (A32B) | ✓ | $41.1 \pm 4.3$ | $48.4 \pm 3.2$ | $35.4 \pm 3.2$ | $35.8 \pm 3.8$ | $64.4 \pm 7.0$ | $59.6 \pm 4.0$ |
| DeepSeek-R1-0528 | 671B (A37B) | ✓ | $\mathbf{67.6 \pm 3.9}$ | $72.1 \pm 2.7$ | $70.2 \pm 3.0$ | $64.6 \pm 3.8$ | $\mathbf{83.3 \pm 5.5}$ | $67.0 \pm 3.9$ |

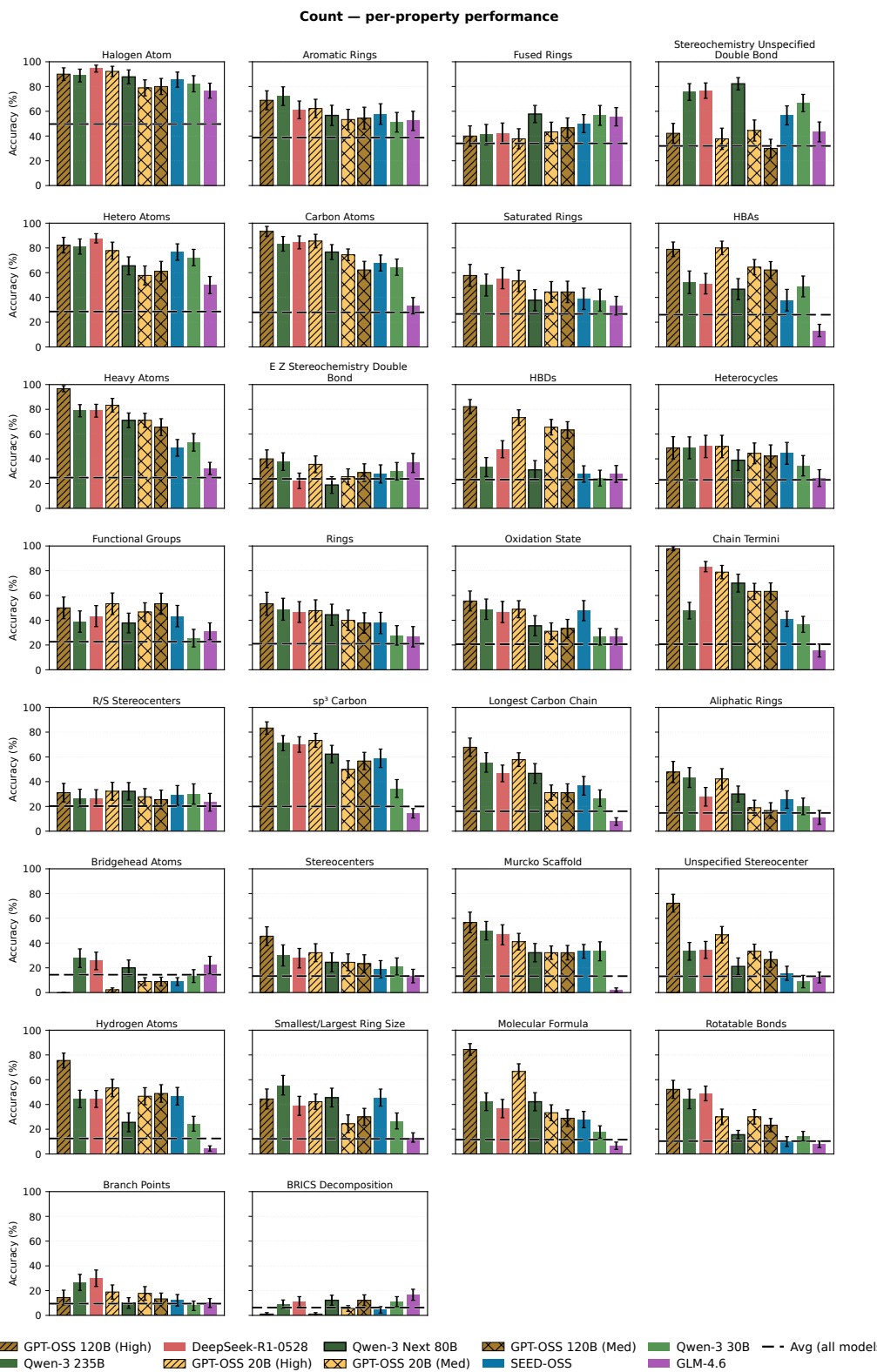

Figure C2: Per-feature performance on counting tasks. Models (rows) are ordered by overall performance; tasks (columns) are ordered by average accuracy. The bottom row shows the mean across all models.

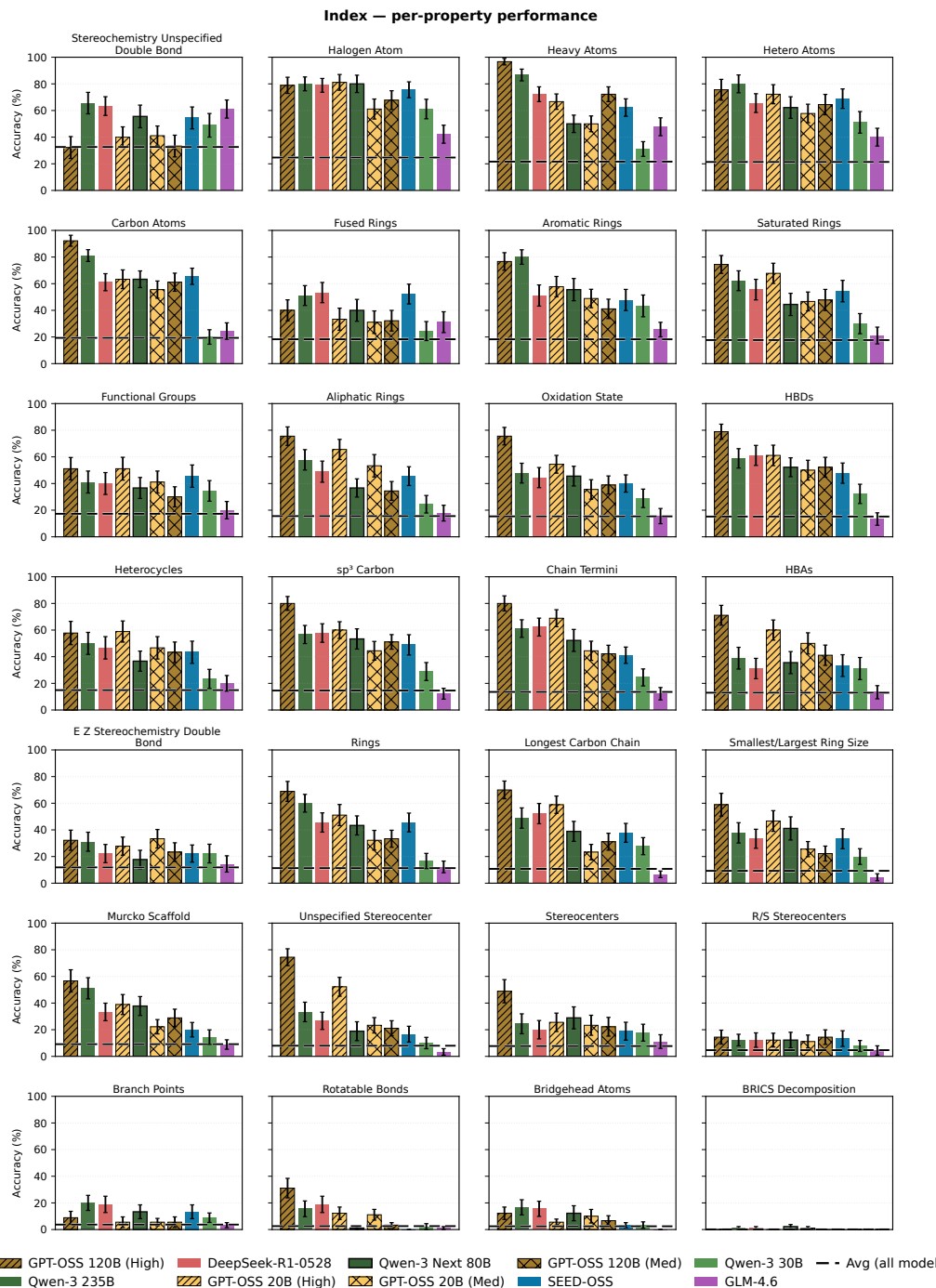

Figure C3: Per-feature performance on indexing tasks. Models (rows) are ordered by overall performance; tasks (columns) are ordered by average accuracy. The bottom row shows the mean across all models.

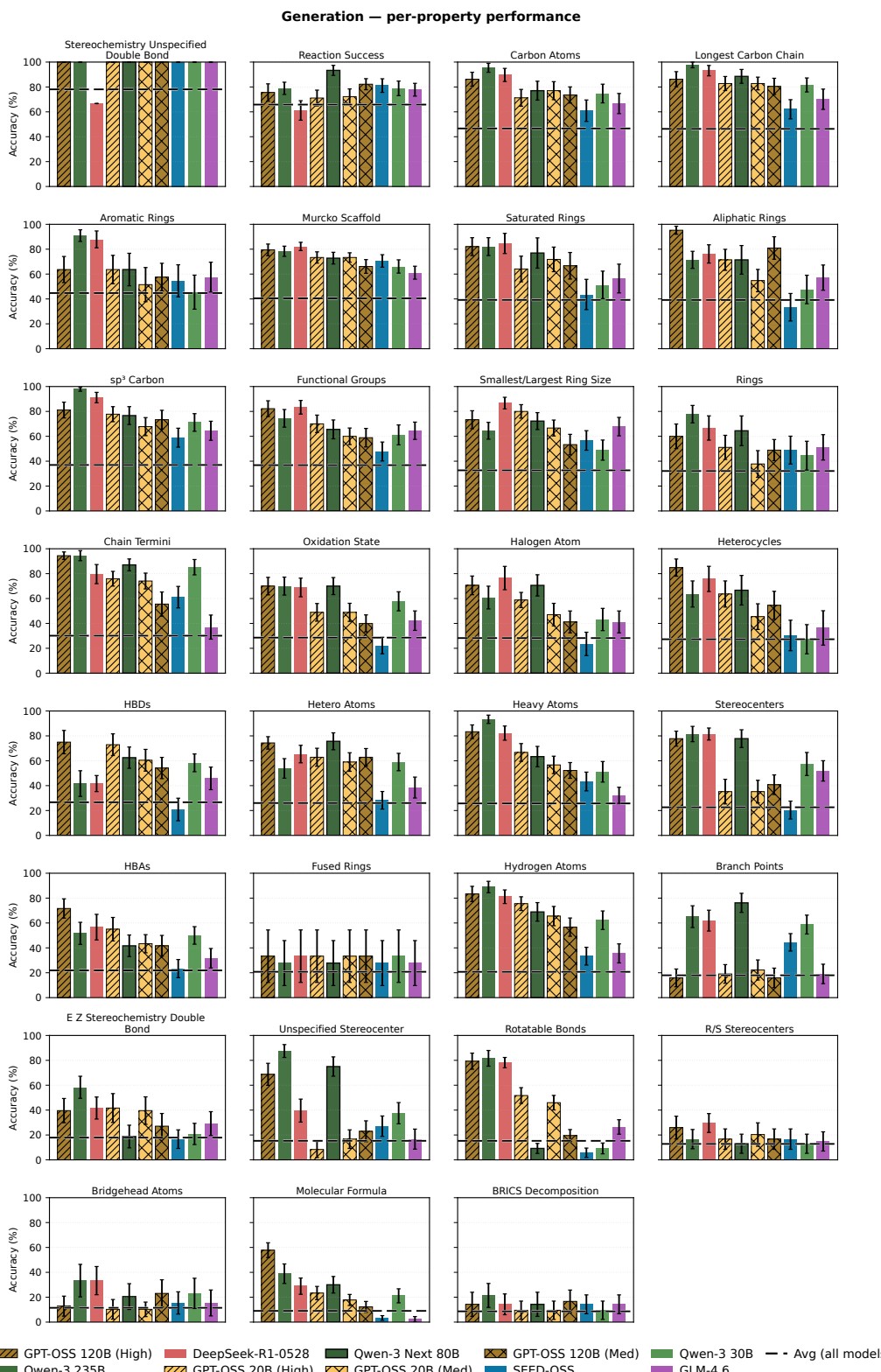

Figure C4: Per-feature performance on generation tasks. Models (rows) are ordered by overall performance; tasks (columns) are ordered by average accuracy. The bottom row shows the mean across all models.

### C.1.3 SENSITIVITY TO SMILES REPRESENTATION

To assess how strongly models rely on specific canonicalization conventions rather than learning representation-invariant molecular reasoning, we compare performance across three SMILES variants: canonical, randomized, and Kekulized. Tables C8 and C9 report numerical results for counting and indexing tasks, respectively. Table C10 further examines robustness to alternative ring-closure numbering schemes, where we modify ring-closure indices to random values between 1 and 100 while preserving molecular identity.

Figures C5 and C6 visualize these degradation patterns. In both figures, points below the diagonal indicate performance degradation under augmentation. Randomized and Kekulized variants consistently reduce accuracy, and systematic degradation under ring enumeration indicates that models often memorize syntactic patterns rather than performing reasoning over the underlying molecular graph. These results are further discussed in the context of failure modes in Section C.2.

Table C8: Counting accuracy (in %) on MOLECULARIQ under different SMILES representations. Error bars indicate standard errors across instances. Relative changes from canonical aromatic are shown as colored subscripts. Model size refers to parameter count; R is a flag for reasoning models.

| | Size | R | Canonical | Canonical (kek.) | Randomized | Randomized (kek.) |
|---|---|---|---|---|---|---|
| | | | | | **Counting** | |
| *Chemistry LLMs* | | | | | | |
| ChemLLM | 7B | | $2.0 \pm 0.4$ | $2.1 \pm 0.7_{+5\%}$ | $2.1 \pm 0.4_{+6\%}$ | $1.6 \pm 0.5_{-21\%}$ |
| LlaSMol | 7B | | $1.3 \pm 0.3$ | $1.4 \pm 0.4_{+1\%}$ | $1.6 \pm 0.3_{+21\%}$ | $3.0 \pm 0.8_{+125\%}$ |
| MolReasoner-Cap | 7B | ✓ | $5.6 \pm 0.8$ | $3.2 \pm 1.0_{-43\%}$ | $6.2 \pm 0.7_{+11\%}$ | $6.7 \pm 1.5_{+20\%}$ |
| MolReasoner-Gen | 7B | ✓ | $5.0 \pm 0.7$ | $2.6 \pm 0.8_{-49\%}$ | $5.5 \pm 0.7_{+8\%}$ | $5.2 \pm 1.2_{+4\%}$ |
| Llama-3 MolInst | 8B | | $3.7 \pm 0.6$ | $4.4 \pm 1.2_{+19\%}$ | $5.0 \pm 0.7_{+38\%}$ | $6.5 \pm 1.3_{+78\%}$ |
| ChemDFM-8B | 8B | | $5.5 \pm 0.7$ | $5.1 \pm 1.2_{-6\%}$ | $4.7 \pm 0.6_{-13\%}$ | $3.7 \pm 1.1_{-33\%}$ |
| ChemDFM-13B | 13B | | $3.4 \pm 0.5$ | $4.7 \pm 1.0_{+37\%}$ | $4.4 \pm 0.6_{+28\%}$ | $4.8 \pm 1.1_{+40\%}$ |
| ChemDFM-R-14B | 14B | ✓ | $12.8 \pm 1.3$ | $12.2 \pm 2.2_{-5\%}$ | $14.0 \pm 1.3_{+10\%}$ | $10.0 \pm 2.1_{-22\%}$ |
| TxGemma-9B | 9B | | $2.7 \pm 0.5$ | $4.5 \pm 1.1_{+68\%}$ | $5.3 \pm 0.7_{+97\%}$ | $4.3 \pm 1.2_{+60\%}$ |
| TxGemma-27B | 27B | | $7.2 \pm 0.8$ | $7.5 \pm 1.4_{+4\%}$ | $6.9 \pm 0.7_{-4\%}$ | $6.2 \pm 1.3_{-14\%}$ |
| Ether0 | 24B | ✓ | $3.1 \pm 0.5$ | $4.7 \pm 1.0_{+48\%}$ | $2.8 \pm 0.5_{-11\%}$ | $3.5 \pm 1.0_{+11\%}$ |
| *Generalist LLMs* | | | | | | |
| Gemma-2 9B | 9B | | $3.4 \pm 0.5$ | $3.2 \pm 0.8_{-7\%}$ | $4.4 \pm 0.5_{+31\%}$ | $4.1 \pm 1.0_{+22\%}$ |
| Gemma-2 27B | 27B | | $6.3 \pm 0.8$ | $3.6 \pm 1.0_{-43\%}$ | $8.0 \pm 0.9_{+27\%}$ | $6.0 \pm 1.3_{-5\%}$ |
| Gemma-3 12B | 12B | | $4.8 \pm 0.8$ | $4.7 \pm 1.3_{-4\%}$ | $8.0 \pm 1.0_{+66\%}$ | $7.3 \pm 1.6_{+51\%}$ |
| Gemma-3 27B | 27B | | $9.1 \pm 1.1$ | $8.3 \pm 1.6_{-9\%}$ | $11.4 \pm 1.1_{+26\%}$ | $9.2 \pm 1.8_{+2\%}$ |
| LLaMA-2 13B | 13B | | $4.1 \pm 0.7$ | $4.4 \pm 1.2_{+6\%}$ | $3.7 \pm 0.5_{-11\%}$ | $4.0 \pm 1.1_{-4\%}$ |
| LLaMA-3 8B | 8B | | $6.1 \pm 0.8$ | $4.5 \pm 1.1_{-26\%}$ | $6.2 \pm 0.8_{+1\%}$ | $6.7 \pm 1.4_{+10\%}$ |
| LLaMA-3.3 70B | 70B | | $9.2 \pm 1.1$ | $14.3 \pm 2.3_{+55\%}$ | $11.6 \pm 1.2_{+26\%}$ | $11.7 \pm 2.2_{+28\%}$ |
| Mistral-7B v0.1 | 7B | | $2.1 \pm 0.4$ | $2.0 \pm 0.5_{-7\%}$ | $1.7 \pm 0.3_{-21\%}$ | $2.7 \pm 0.7_{+28\%}$ |
| Mistral-Small | 24B | | $9.1 \pm 0.9$ | $7.5 \pm 1.4_{-17\%}$ | $10.6 \pm 1.0_{+18\%}$ | $10.8 \pm 1.8_{+19\%}$ |
| Nemotron-Nano 9B v2 | 9B | ✓ | $12.7 \pm 1.1$ | $9.9 \pm 1.7_{-22\%}$ | $13.2 \pm 1.1_{+4\%}$ | $9.7 \pm 1.7_{-24\%}$ |
| SEED-OSS | 36B | ✓ | $27.2 \pm 1.5$ | $20.7 \pm 2.3_{-24\%}$ | $29.9 \pm 1.5_{+10\%}$ | $21.1 \pm 2.5_{-22\%}$ |
| Qwen-2.5 7B | 7B | | $5.5 \pm 0.8$ | $4.8 \pm 1.2_{-12\%}$ | $6.7 \pm 0.8_{+22\%}$ | $6.8 \pm 1.5_{+25\%}$ |
| Qwen-2.5 14B | 14B | | $8.0 \pm 1.1$ | $6.8 \pm 1.7_{-16\%}$ | $8.2 \pm 1.0_{+2\%}$ | $5.2 \pm 1.5_{-35\%}$ |
| Qwen-3 8B | 8B | ✓ | $13.1 \pm 1.1$ | $8.9 \pm 1.7_{-32\%}$ | $13.4 \pm 1.1_{+2\%}$ | $11.6 \pm 1.9_{-11\%}$ |
| Qwen-3 14B | 14B | ✓ | $14.9 \pm 1.2$ | $11.3 \pm 1.8_{-25\%}$ | $14.9 \pm 1.2$ | $14.1 \pm 2.2_{-5\%}$ |
| Qwen-3 32B | 32B | ✓ | $17.8 \pm 1.3$ | $14.6 \pm 2.0_{-18\%}$ | $17.8 \pm 1.2$ | $14.6 \pm 2.1_{-18\%}$ |
| Qwen-3 30B | 30B (A3B) | ✓ | $24.8 \pm 1.5$ | $17.3 \pm 2.2_{-31\%}$ | $24.5 \pm 1.4_{-1\%}$ | $17.9 \pm 2.2_{-28\%}$ |
| Qwen-3 Next 80B | 80B (A3B) | ✓ | $33.5 \pm 1.6$ | $28.5 \pm 2.7_{-15\%}$ | $31.4 \pm 1.5_{-6\%}$ | $21.7 \pm 2.4_{-35\%}$ |
| Qwen-3 235B | 235B (A22B) | ✓ | $38.0 \pm 1.7$ | $36.5 \pm 2.8_{-4\%}$ | $39.2 \pm 1.6_{+3\%}$ | $27.5 \pm 2.7_{-28\%}$ |
| GPT-OSS 20B (Low) | 20B (A4B) | ✓ | $13.9 \pm 1.2$ | $11.9 \pm 1.9_{-15\%}$ | $12.8 \pm 1.0_{-8\%}$ | $12.5 \pm 2.1_{-10\%}$ |
| GPT-OSS 20B (Med) | 20B (A4B) | ✓ | $31.0 \pm 1.5$ | $21.8 \pm 2.3_{-30\%}$ | $29.7 \pm 1.4_{-4\%}$ | $18.6 \pm 2.2_{-40\%}$ |
| GPT-OSS 20B (High) | 20B (A4B) | ✓ | $41.8 \pm 1.7$ | $32.1 \pm 2.7_{-23\%}$ | $41.3 \pm 1.6_{-1\%}$ | $30.3 \pm 2.7_{-28\%}$ |
| GPT-OSS 120B (Low) | 120B (A5B) | ✓ | $18.6 \pm 1.3$ | $12.0 \pm 1.8_{-35\%}$ | $18.0 \pm 1.2_{-3\%}$ | $14.4 \pm 2.2_{-22\%}$ |
| GPT-OSS 120B (Med) | 120B (A5B) | ✓ | $33.1 \pm 1.6$ | $23.1 \pm 2.5_{-30\%}$ | $30.1 \pm 1.4_{-9\%}$ | $19.8 \pm 2.3_{-40\%}$ |
| GPT-OSS 120B (High) | 120B (A5B) | ✓ | **$49.5 \pm 1.8$** | **$42.5 \pm 3.0_{-14\%}$** | **$47.2 \pm 1.7_{-5\%}$** | **$41.7 \pm 3.0_{-16\%}$** |
| GLM-4.6 | 355B (A32B) | ✓ | $15.9 \pm 1.2$ | $15.6 \pm 2.0_{-2\%}$ | $16.2 \pm 1.1_{+2\%}$ | $15.1 \pm 2.1_{-5\%}$ |
| DeepSeek-R1-0528 | 671B (A37B) | ✓ | $37.6 \pm 1.7$ | $33.6 \pm 2.7_{-10\%}$ | $38.5 \pm 1.6_{+2\%}$ | $28.7 \pm 2.6_{-23\%}$ |

Table C9: Indexing accuracy (in %) on MOLECULARIQ under different SMILES representations. Error bars indicate standard errors across instances. Relative changes from canonical aromatic are shown as colored subscripts. Model size refers to parameter count; R is a flag for reasoning models.

| | Size | R | Indexing Canonical | Canonical (kek.) | Randomized | Randomized (kek.) |
|---|---|---|---|---|---|---|
| *Chemistry LLMs* | | | | | | |
| ChemLLM | 7B | | $0.1 \pm 0.1$ | $0.3 \pm 0.2_{+238\%}$ | $0.5 \pm 0.2_{+399\%}$ | $0.7 \pm 0.4_{+567\%}$ |
| LlaSMol | 7B | | $1.0 \pm 0.3$ | $3.7 \pm 1.2_{+254\%}$ | $2.2 \pm 0.5_{+114\%}$ | $1.3 \pm 0.6_{+27\%}$ |
| MolReasoner-Cap | 7B | ✓ | $0.6 \pm 0.2$ | $0.5 \pm 0.3_{-22\%}$ | $1.1 \pm 0.3_{+69\%}$ | $0.3 \pm 0.3_{-49\%}$ |
| MolReasoner-Gen | 7B | ✓ | $1.3 \pm 0.3$ | $1.5 \pm 0.6_{+13\%}$ | $1.8 \pm 0.4_{+37\%}$ | $0.8 \pm 0.4_{-38\%}$ |
| Llama-3 MolInst | 8B | | $0.3 \pm 0.1$ | $0.7 \pm 0.5_{+93\%}$ | $0.6 \pm 0.2_{+85\%}$ | $0.7 \pm 0.4_{+90\%}$ |
| ChemDFM-8B | 8B | | $0.0 \pm 0.0$ | $0.2 \pm 0.2$ | $0.1 \pm 0.1$ | $0.0 \pm 0.0$ |
| ChemDFM-13B | 13B | | $0.2 \pm 0.1$ | $0.3 \pm 0.2_{+69\%}$ | $0.3 \pm 0.1_{+75\%}$ | $0.0 \pm 0.0_{-100\%}$ |
| ChemDFM-R-14B | 14B | ✓ | $2.5 \pm 0.6$ | $2.5 \pm 1.1$ | $3.4 \pm 0.7_{+35\%}$ | $2.0 \pm 1.0_{-22\%}$ |
| TxGemma-9B | 9B | | $0.4 \pm 0.2$ | $0.7 \pm 0.4_{+50\%}$ | $0.5 \pm 0.2_{+22\%}$ | $0.3 \pm 0.2_{-26\%}$ |
| TxGemma-27B | 27B | | $1.2 \pm 0.3$ | $1.5 \pm 0.7_{+22\%}$ | $2.3 \pm 0.5_{+88\%}$ | $1.8 \pm 0.8_{+47\%}$ |
| Ether0 | 24B | ✓ | $0.0 \pm 0.0$ | $0.0 \pm 0.0$ | $0.1 \pm 0.1$ | $0.0 \pm 0.0$ |
| *Generalist LLMs* | | | | | | |
| Gemma-2 9B | 9B | | $1.3 \pm 0.4$ | $4.0 \pm 1.2_{+201\%}$ | $2.1 \pm 0.5_{+55\%}$ | $1.3 \pm 0.6_{-1\%}$ |
| Gemma-2 27B | 27B | | $0.9 \pm 0.3$ | $0.7 \pm 0.5_{-25\%}$ | $1.1 \pm 0.3_{+28\%}$ | $0.5 \pm 0.4_{-44\%}$ |
| Gemma-3 12B | 12B | | $0.8 \pm 0.3$ | $1.7 \pm 0.9_{+111\%}$ | $0.9 \pm 0.3_{+12\%}$ | $0.0 \pm 0.0_{-100\%}$ |
| Gemma-3 27B | 27B | | $1.3 \pm 0.4$ | $0.8 \pm 0.6_{-35\%}$ | $1.6 \pm 0.4_{+23\%}$ | $0.2 \pm 0.2_{-87\%}$ |
| LLaMA-2 13B | 13B | | $0.1 \pm 0.1$ | $0.0 \pm 0.0_{-100\%}$ | $0.2 \pm 0.1_{+66\%}$ | $0.0 \pm 0.0_{-100\%}$ |
| LLaMA-3 8B | 8B | | $0.2 \pm 0.1$ | $0.3 \pm 0.2_{+35\%}$ | $0.2 \pm 0.1$ | $0.2 \pm 0.2_{-33\%}$ |
| LLaMA-3.3 70B | 70B | | $1.4 \pm 0.4$ | $1.5 \pm 0.9_{+5\%}$ | $2.2 \pm 0.6_{+55\%}$ | $0.5 \pm 0.5_{-66\%}$ |
| Mistral-7B v0.1 | 7B | | $0.3 \pm 0.1$ | $0.8 \pm 0.4_{+182\%}$ | $0.3 \pm 0.1$ | $0.2 \pm 0.2_{-44\%}$ |
| Mistral-Small | 24B | | $1.3 \pm 0.4$ | $2.0 \pm 0.8_{+50\%}$ | $2.1 \pm 0.5_{+59\%}$ | $0.2 \pm 0.2_{-88\%}$ |
| Nemotron-Nano 9B v2 | 9B | ✓ | $7.3 \pm 0.8$ | $4.4 \pm 1.3_{-40\%}$ | $8.2 \pm 0.9_{+13\%}$ | $1.7 \pm 0.7_{-77\%}$ |
| SEED-OSS | 36B | ✓ | $28.0 \pm 1.5$ | $14.8 \pm 2.2_{-47\%}$ | $29.8 \pm 1.5_{+6\%}$ | $14.1 \pm 2.1_{-50\%}$ |
| Qwen-2.5 7B | 7B | | $1.4 \pm 0.4$ | $4.7 \pm 1.4_{+238\%}$ | $1.5 \pm 0.4_{+11\%}$ | $2.2 \pm 0.9_{+55\%}$ |
| Qwen-2.5 14B | 14B | | $1.0 \pm 0.4$ | $3.0 \pm 1.2_{+190\%}$ | $1.2 \pm 0.4_{+14\%}$ | $0.0 \pm 0.0_{-100\%}$ |
| Qwen-3 8B | 8B | ✓ | $6.6 \pm 0.8$ | $4.2 \pm 1.3_{-36\%}$ | $7.0 \pm 0.8_{+6\%}$ | $1.0 \pm 0.6_{-85\%}$ |
| Qwen-3 14B | 14B | ✓ | $8.3 \pm 0.9$ | $4.2 \pm 1.1_{-49\%}$ | $9.0 \pm 1.0_{+9\%}$ | $2.8 \pm 0.8_{-66\%}$ |
| Qwen-3 32B | 32B | ✓ | $9.3 \pm 0.9$ | $5.9 \pm 1.4_{-37\%}$ | $11.4 \pm 1.1_{+22\%}$ | $4.1 \pm 1.1_{-55\%}$ |
| Qwen-3 30B | 30B (A3B) | ✓ | $19.6 \pm 1.3$ | $8.9 \pm 1.7_{-54\%}$ | $18.4 \pm 1.3_{-6\%}$ | $7.1 \pm 1.5_{-64\%}$ |
| Qwen-3 Next 80B | 80B (A3B) | ✓ | $31.4 \pm 1.6$ | $15.7 \pm 2.3_{-50\%}$ | $28.8 \pm 1.5_{-8\%}$ | $16.9 \pm 2.3_{-46\%}$ |
| Qwen-3 235B | 235B (A22B) | ✓ | $38.5 \pm 1.6$ | $25.6 \pm 2.6_{-33\%}$ | $37.5 \pm 1.6_{-2\%}$ | $20.4 \pm 2.5_{-47\%}$ |
| GPT-OSS 20B (Low) | 20B (A4B) | ✓ | $5.8 \pm 0.7$ | $5.9 \pm 1.5_{+1\%}$ | $5.8 \pm 0.8$ | $3.0 \pm 1.0_{-49\%}$ |
| GPT-OSS 20B (Med) | 20B (A4B) | ✓ | $26.7 \pm 1.5$ | $13.0 \pm 2.1_{-51\%}$ | $23.5 \pm 1.4_{-12\%}$ | $11.6 \pm 1.9_{-57\%}$ |
| GPT-OSS 20B (High) | 20B (A4B) | ✓ | $37.4 \pm 1.6$ | $25.6 \pm 2.7_{-32\%}$ | $34.5 \pm 1.6_{-8\%}$ | $18.4 \pm 2.4_{-51\%}$ |
| GPT-OSS 120B (Low) | 120B (A5B) | ✓ | $12.0 \pm 1.1$ | $8.4 \pm 1.9_{-30\%}$ | $11.6 \pm 1.1_{-3\%}$ | $5.6 \pm 1.5_{-53\%}$ |
| GPT-OSS 120B (Med) | 120B (A5B) | ✓ | $27.4 \pm 1.5$ | $13.3 \pm 2.0_{-51\%}$ | $23.6 \pm 1.4_{-14\%}$ | $10.3 \pm 1.8_{-62\%}$ |
| GPT-OSS 120B (High) | 120B (A5B) | ✓ | $\mathbf{47.2 \pm 1.8}$ | $\mathbf{36.5 \pm 3.1}_{-23\%}$ | $\mathbf{43.6 \pm 1.7}_{-8\%}$ | $\mathbf{28.9 \pm 2.8}_{-39\%}$ |
| GLM-4.6 | 355B (A32B) | ✓ | $11.9 \pm 1.0$ | $7.6 \pm 1.5_{-37\%}$ | $13.3 \pm 1.1_{+11\%}$ | $6.0 \pm 1.1_{-50\%}$ |
| DeepSeek-R1-0528 | 671B (A37B) | ✓ | $34.6 \pm 1.6$ | $18.9 \pm 2.3_{-45\%}$ | $33.8 \pm 1.6_{-2\%}$ | $14.4 \pm 2.0_{-58\%}$ |

Table C10: Counting and indexing accuracy (in %) on MOLECULARIQ under different Ring Enumerations. Error bars indicate standard errors across instances. Relative changes from original ring enumeration are shown as colored subscripts. Model size refers to parameter count; R is a flag for reasoning models.

| Model | Size | R | Counting | | Indexing | |
| --- | --- | --- | --- | --- | --- | --- |
| | | | Original | Ring Enum. | Original | Ring Enum. |
| *Chemistry LLMs* | | | | | | |
| ChemDFM-R-14B | 14B | ✓ | $12.3 \pm 0.8$ | $10.4 \pm 0.8_{-15\%}$ | $2.8 \pm 0.4$ | $2.1 \pm 0.4_{-25\%}$ |
| TxGemma-27B | 27B | | $6.2 \pm 0.5$ | $5.5 \pm 0.5_{-11\%}$ | $1.6 \pm 0.3$ | $1.5 \pm 0.3_{-9\%}$ |
| *Generalist LLMs* | | | | | | |
| Gemma-3 12B | 12B | | $6.2 \pm 0.6$ | $5.7 \pm 0.5_{-8\%}$ | $2.8 \pm 0.4$ | $0.3 \pm 0.1_{-88\%}$ |
| Mistral-Small | 24B | | $9.3 \pm 0.6$ | $7.8 \pm 0.7_{-17\%}$ | $1.2 \pm 0.2$ | $0.6 \pm 0.2_{-50\%}$ |
| Nemotron-Nano 9B v2 | 9B | ✓ | $10.9 \pm 0.7$ | $8.3 \pm 0.6_{-24\%}$ | $5.0 \pm 0.5$ | $3.8 \pm 0.4_{-23\%}$ |
| Qwen-3 8B | 8B | ✓ | $11.2 \pm 0.7$ | $8.4 \pm 0.6_{-25\%}$ | $4.6 \pm 0.5$ | $2.7 \pm 0.4_{-41\%}$ |
| Qwen-3 14B | 14B | ✓ | $12.6 \pm 0.8$ | $11.0 \pm 0.7_{-13\%}$ | $5.8 \pm 0.5$ | $5.4 \pm 0.5_{-7\%}$ |
| Qwen-3 30B | 30B (A3B) | ✓ | $20.4 \pm 0.9$ | $17.3 \pm 1.0_{-15\%}$ | $13.2 \pm 0.8$ | $13.0 \pm 0.9_{-2\%}$ |
| Qwen-3 235B | 235B (A22B) | ✓ | $33.6 \pm 1.1$ | $33.0 \pm 1.1_{-2\%}$ | $31.5 \pm 1.1$ | $30.1 \pm 1.1_{-4\%}$ |
| GPT-OSS 20B (Low) | 20B (A4B) | ✓ | $12.3 \pm 0.7$ | $8.2 \pm 0.7_{-34\%}$ | $4.7 \pm 0.5$ | $3.4 \pm 0.5_{-26\%}$ |
| GPT-OSS 20B (Med) | 20B (A4B) | ✓ | $25.0 \pm 0.9$ | $23.5 \pm 1.0_{-6\%}$ | $19.5 \pm 0.9$ | $19.2 \pm 0.9_{-1\%}$ |
| GPT-OSS 20B (High) | 20B (A4B) | ✓ | $36.1 \pm 1.1$ | $19.5 \pm 0.9_{-46\%}$ | $30.5 \pm 1.0$ | $14.1 \pm 0.7_{-54\%}$ |
| GPT-OSS 120B (Low) | 120B (A5B) | ✓ | $15.2 \pm 0.8$ | $10.5 \pm 0.8_{-31\%}$ | $9.0 \pm 0.7$ | $6.3 \pm 0.6_{-30\%}$ |
| GPT-OSS 120B (Med) | 120B (A5B) | ✓ | $26.1 \pm 1.0$ | $19.8 \pm 1.0_{-24\%}$ | $20.0 \pm 0.9$ | $14.3 \pm 0.8_{-29\%}$ |
| GPT-OSS 120B (High) | 120B (A5B) | ✓ | $43.7 \pm 1.2$ | $40.0 \pm 1.1_{-8\%}$ | $39.9 \pm 1.2$ | $35.8 \pm 1.1_{-10\%}$ |

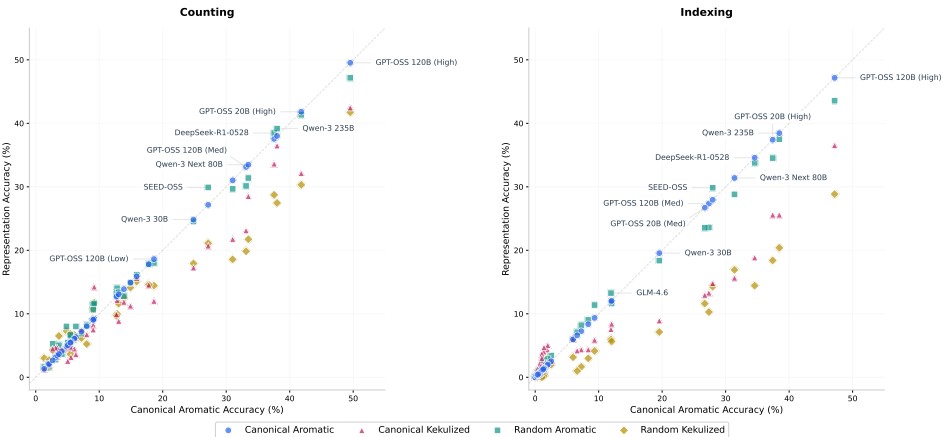

Figure C5: Impact of SMILES representation on counting (left) and indexing (right) tasks. Each point compares accuracy under canonical aromatic SMILES (x-axis) to accuracy under augmented variants. Points below the diagonal indicate degradation under augmentation.

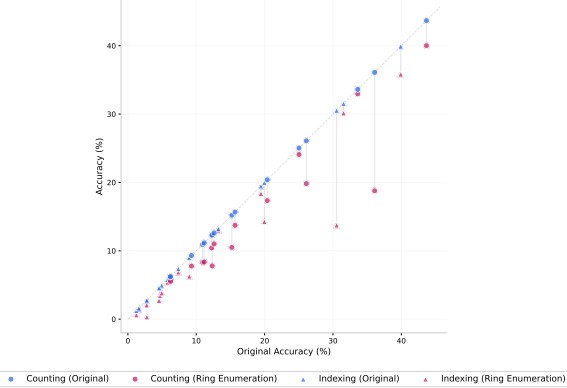

Figure C6: Effect of ring-closure enumeration on counting and indexing tasks. Accuracy with original SMILES (x-axis) is compared to accuracy using enumerated ring indices (y-axis). Points below the diagonal indicate degradation.

### C.1.4 RESPONSE LENGTH ANALYSIS

Figure C7 plots overall accuracy as a function of average response length across models, examining whether verbose chain-of-thought reasoning correlates with improved performance. Figure C8 provides the complementary view, showing response length distributions conditioned on accuracy level for the top-10 models. A consistent pattern emerges: failed responses (0% accuracy) exhibit substantially longer token lengths than successful ones (100% accuracy). This suggests that excessive verbosity may reflect unproductive reasoning-models generate longer traces when struggling rather than reasoning more effectively. The pattern holds across reasoning models of different sizes, indicating that this failure mode is architecture-independent.

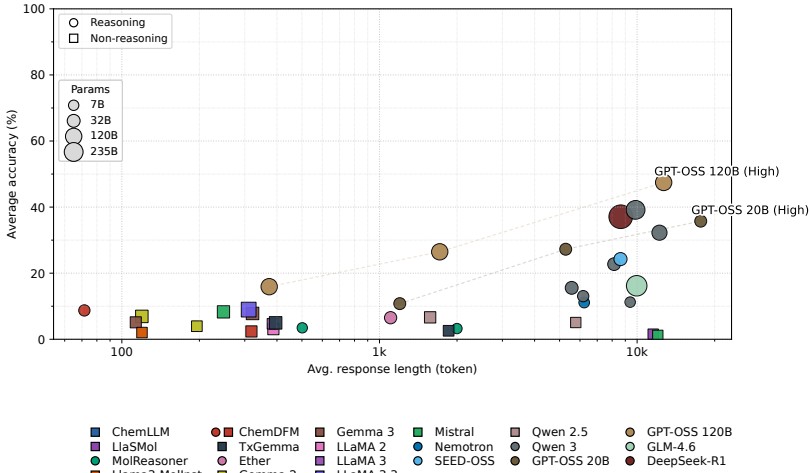

Figure C7: Overall accuracy as a function of average response length across all evaluated models.

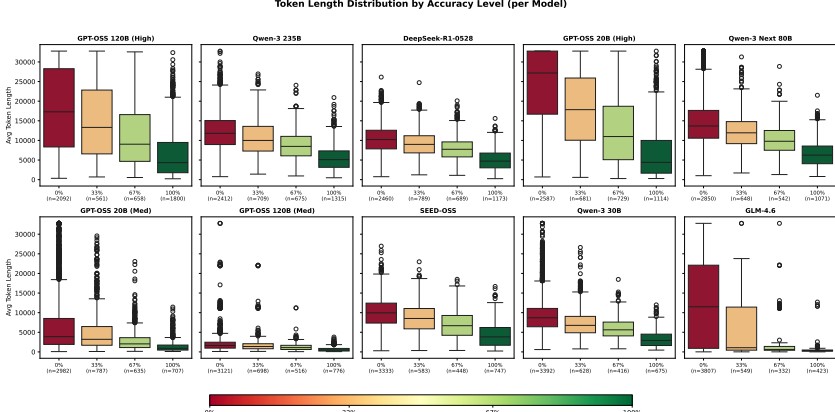

Figure C8: Response length distributions conditioned on accuracy level for the top-10 models. Each box shows the distribution of average token lengths for responses achieving a given accuracy (0%, 33%, 67%, or 100% across rollouts). Failed responses (0% accuracy, red) consistently exhibit longer token lengths than successful ones (100% accuracy, green) across the top-10 performing models.

### C.1.5 ROLLOUT ABLATION

Table C11 compares performance between standard evaluation (3 rollouts) and extended evaluation (8 rollouts) for a subsection of models, revealing overall consistency.

Table C11: Results on the MOLECULARIQ benchmark comparing 3 vs 8 rollouts. We report overall accuracy as well as accuracy across task families (in %). Error bars indicate standard errors across instances. Relative changes from 3 to 8 rollouts are shown as colored subscripts.

| Model | Size | R | Overall | | Counting | | Indexing | | Generation | |
|---|---|---|---|---|---|---|---|---|---|---|
| | | | 3 | 8 | 3 | 8 | 3 | 8 | 3 | 8 |
| *Chemistry LLMs* | | | | | | | | | | |
| ChemDFM-R-14B | 14B | ✓ | $8.7 \pm 0.4$ | $8.6 \pm 0.4_{-1\%}$ | $12.9 \pm 0.8$ | $12.9 \pm 0.8_{+0\%}$ | $2.8 \pm 0.4$ | $2.6 \pm 0.4_{-6\%}$ | $10.5 \pm 0.8$ | $10.3 \pm 0.8_{-2\%}$ |
| *Generalist LLMs* | | | | | | | | | | |
| Nemotron-Nano 9B v2 | 9B | ✓ | $11.1 \pm 0.4$ | $11.1 \pm 0.4_{-0\%}$ | $12.2 \pm 0.7$ | $12.0 \pm 0.6_{-1\%}$ | $6.6 \pm 0.5$ | $6.4 \pm 0.5_{-3\%}$ | $14.8 \pm 0.8$ | $15.1 \pm 0.8_{+2\%}$ |
| Qwen-3 30B | 30B (A3B) | ✓ | $22.7 \pm 0.5$ | $22.5 \pm 0.6_{-1\%}$ | $23.0 \pm 0.9$ | $23.3 \pm 1.0_{+1\%}$ | $16.5 \pm 0.8$ | $15.4 \pm 0.8_{-7\%}$ | $29.4 \pm 1.0$ | $29.4 \pm 1.1_{-0\%}$ |
| Qwen-3 Next 80B | 80B (A3B) | ✓ | $32.3 \pm 0.6$ | $30.5 \pm 0.5_{-5\%}$ | $30.7 \pm 1.0$ | $29.3 \pm 0.9_{-4\%}$ | $26.9 \pm 0.9$ | $25.4 \pm 0.9_{-6\%}$ | $40.0 \pm 1.1$ | $37.5 \pm 1.0_{-6\%}$ |
| Qwen-3 235B | 235B (A22B) | ✓ | $39.2 \pm 0.6$ | $38.6 \pm 0.6_{-1\%}$ | $37.1 \pm 1.0$ | $36.6 \pm 1.0_{-1\%}$ | $34.5 \pm 1.0$ | $33.9 \pm 1.0_{-2\%}$ | $46.7 \pm 1.1$ | $46.2 \pm 1.0_{-1\%}$ |
| GPT-OSS 120B (Low) | 120B (A5B) | ✓ | $15.9 \pm 0.5$ | $16.1 \pm 0.5_{+1\%}$ | $17.1 \pm 0.8$ | $17.8 \pm 0.9_{+4\%}$ | $10.7 \pm 0.6$ | $10.4 \pm 0.7_{-3\%}$ | $20.3 \pm 0.9$ | $20.5 \pm 1.0_{+1\%}$ |
| GPT-OSS 120B (Med) | 120B (A5B) | ✓ | $26.5 \pm 0.5$ | $27.6 \pm 0.6_{+4\%}$ | $29.1 \pm 0.9$ | $29.3 \pm 0.9_{+1\%}$ | $22.3 \pm 0.8$ | $23.7 \pm 0.9_{+6\%}$ | $28.0 \pm 1.0$ | $29.9 \pm 1.0_{+7\%}$ |
| GPT-OSS 120B (High) | 120B (A5B) | ✓ | $\mathbf{47.5 \pm 0.6}$ | $\mathbf{45.4 \pm 0.6}_{-4\%}$ | $\mathbf{46.8 \pm 1.1}$ | $\mathbf{45.4 \pm 1.1}_{-3\%}$ | $\mathbf{42.5 \pm 1.1}$ | $\mathbf{41.3 \pm 1.1}_{-3\%}$ | $\mathbf{53.7 \pm 1.1}$ | $\mathbf{49.8 \pm 1.0}_{-7\%}$ |

### C.1.6 CHEMISTRY LLMS AND BASE MODEL COMPARISON

Table C12 compares chemistry-specialized LLMs against their corresponding base models, quantifying the impact of domain-specific fine-tuning on molecular reasoning.

Table C12: Comparison of chemistry LLMs to their base models. Reported values are overall accuracy (in %) with standard errors on MOLECULARIQ.

| Chemistry LLMs | | Base Model | |
|---|---|---|---|
| LlaSMol | $1.5 \pm 0.1$ | $1.4 \pm 0.1$ | Mistral-7B v0.1 |
| MolReasoner-Cap | $3.5 \pm 0.2$ | $5.1 \pm 0.3$ | Qwen-2.5 7B |
| MolReasoner-Gen | $3.2 \pm 0.2$ | $5.1 \pm 0.3$ | Qwen-2.5 7B |
| Llama-3 MolInst | $2.0 \pm 0.2$ | $4.8 \pm 0.3$ | LLaMA-3 8B |
| ChemDFM-8B | $3.7 \pm 0.2$ | $4.8 \pm 0.3$ | LLaMA-3 8B |
| ChemDFM-13B | $2.7 \pm 0.2$ | $3.1 \pm 0.2$ | LLaMA-2 13B |
| ChemDFM-R-14B | $8.7 \pm 0.4$ | $6.6 \pm 0.3$ | Qwen-2.5 14B |
| TxGemma-9B | $2.6 \pm 0.2$ | $4.0 \pm 0.2$ | Gemma-2 9B |
| TxGemma-27B | $5.0 \pm 0.2$ | $6.9 \pm 0.3$ | Gemma-2 27B |
| Ether0 | $6.5 \pm 0.3$ | $8.3 \pm 0.3$ | Mistral-Small |

### C.1.7 OUTPUT TYPE VALIDITY

Table C13 reports type validity—the fraction of outputs satisfying expected syntactic constraints—for each task category (integers for counting, integer lists for indexing, valid SMILES for generation). Type validity provides an upper bound on achievable accuracy, as malformed outputs cannot be correct; the relationship between type validity and actual accuracy is further analyzed in Figure C13. Table C11 compares performance between standard evaluation (3 rollouts) and extended evaluation (8 rollouts) for a subsection of models, revealing overall consistency.

Table C13: Type-validity results on the MOLECULARIQ benchmark. We report overall type-valid rate as well as rates across task families (in %). Error bars indicate standard errors across instances. Model size refers to parameter count; R is a flag for reasoning models.

| | Size | R | Overall | Task family | | |
| | | | | Counting | Indexing | Generation |
|---|---|---|---|---|---|---|
| *Chemistry LLMs* | | | | | | |
| ChemLLM | 7B | | $33.8 \pm 0.5$ | $55.6 \pm 0.8$ | $18.8 \pm 0.6$ | $25.6 \pm 0.7$ |
| LlaSMol | 7B | | $28.7 \pm 0.4$ | $35.1 \pm 0.7$ | $25.8 \pm 0.7$ | $24.7 \pm 0.7$ |
| MolReasoner-Cap | 7B | ✓ | $61.1 \pm 0.6$ | $72.8 \pm 0.9$ | $76.5 \pm 0.8$ | $30.6 \pm 0.8$ |
| MolReasoner-Gen | 7B | ✓ | $53.1 \pm 0.5$ | $61.1 \pm 0.9$ | $69.8 \pm 0.8$ | $25.3 \pm 0.7$ |
| Llama-3 MolInst | 8B | | $58.4 \pm 0.6$ | $92.9 \pm 0.5$ | $74.0 \pm 0.8$ | $1.4 \pm 0.2$ |
| ChemDFM-8B | 8B | | $70.0 \pm 0.5$ | $77.8 \pm 0.8$ | $60.8 \pm 0.9$ | $71.1 \pm 0.8$ |
| ChemDFM-13B | 13B | | $46.0 \pm 0.5$ | $65.5 \pm 0.8$ | $45.5 \pm 0.9$ | $24.1 \pm 0.7$ |
| ChemDFM-R-14B | 14B | ✓ | $72.6 \pm 0.6$ | $88.0 \pm 0.8$ | $73.6 \pm 1.1$ | $53.9 \pm 1.3$ |
| TxGemma-9B | 9B | | $62.7 \pm 0.5$ | $78.6 \pm 0.7$ | $78.3 \pm 0.7$ | $27.2 \pm 0.7$ |
| TxGemma-27B | 27B | | $87.6 \pm 0.3$ | $87.7 \pm 0.7$ | $89.3 \pm 0.5$ | $85.6 \pm 0.5$ |
| Ether0 | 24B | ✓ | $27.4 \pm 0.6$ | $9.0 \pm 0.5$ | $0.0 \pm 0.0$ | $78.8 \pm 0.7$ |
| *Generalist LLMs* | | | | | | |
| Gemma-2 9B | 9B | | $56.1 \pm 0.5$ | $60.6 \pm 0.8$ | $54.1 \pm 0.7$ | $53.3 \pm 0.8$ |
| Gemma-2 27B | 27B | | $91.6 \pm 0.3$ | $93.1 \pm 0.6$ | $96.8 \pm 0.3$ | $84.3 \pm 0.7$ |
| Gemma-3 12B | 12B | | $32.7 \pm 0.6$ | $47.1 \pm 1.2$ | $18.8 \pm 0.8$ | $31.6 \pm 1.0$ |
| Gemma-3 27B | 27B | | $71.5 \pm 0.6$ | $87.9 \pm 0.7$ | $56.7 \pm 1.1$ | $69.2 \pm 1.0$ |
| LLaMA-2 13B | 13B | | $84.5 \pm 0.4$ | $92.3 \pm 0.5$ | $88.3 \pm 0.6$ | $71.5 \pm 0.8$ |
| LLaMA-3 8B | 8B | | $80.7 \pm 0.4$ | $91.5 \pm 0.6$ | $83.4 \pm 0.7$ | $65.5 \pm 0.9$ |
| LLaMA-3.3 70B | 70B | | $88.7 \pm 0.4$ | $91.6 \pm 0.6$ | $92.8 \pm 0.5$ | $81.0 \pm 0.8$ |
| Mistral-7B v0.1 | 7B | | $37.6 \pm 0.5$ | $43.0 \pm 0.7$ | $52.8 \pm 0.7$ | $14.5 \pm 0.5$ |
| Mistral-Small | 24B | | $90.2 \pm 0.3$ | $92.6 \pm 0.6$ | $94.5 \pm 0.4$ | $82.8 \pm 0.7$ |
| Nemotron-Nano 9B v2 | 9B | ✓ | $75.8 \pm 0.5$ | $89.4 \pm 0.6$ | $93.5 \pm 0.4$ | $40.7 \pm 0.9$ |
| SEED-OSS | 36B | ✓ | $74.4 \pm 0.6$ | $88.9 \pm 0.7$ | $92.6 \pm 0.5$ | $37.6 \pm 1.0$ |
| Qwen-2.5 7B | 7B | | $73.6 \pm 0.5$ | $93.4 \pm 0.6$ | $65.7 \pm 1.0$ | $59.5 \pm 0.9$ |
| Qwen-2.5 14B | 14B | | $75.6 \pm 0.6$ | $90.4 \pm 0.7$ | $60.3 \pm 1.2$ | $75.6 \pm 1.1$ |
| Qwen-3 8B | 8B | ✓ | $75.0 \pm 0.5$ | $83.7 \pm 0.7$ | $90.2 \pm 0.5$ | $48.1 \pm 1.0$ |
| Qwen-3 14B | 14B | ✓ | $79.3 \pm 0.5$ | $90.5 \pm 0.6$ | $90.5 \pm 0.5$ | $54.1 \pm 0.9$ |
| Qwen-3 32B | 32B | ✓ | $82.4 \pm 0.4$ | $90.6 \pm 0.6$ | $91.5 \pm 0.5$ | $63.1 \pm 0.9$ |
| Qwen-3 30B | 30B (A3B) | ✓ | $88.8 \pm 0.4$ | $92.5 \pm 0.6$ | $97.9 \pm 0.3$ | $74.6 \pm 0.8$ |
| Qwen-3 Next 80B | 80B (A3B) | ✓ | $89.3 \pm 0.4$ | $90.9 \pm 0.6$ | $94.9 \pm 0.5$ | $81.2 \pm 0.7$ |
| Qwen-3 235B | 235B (A22B) | ✓ | $\mathbf{92.7 \pm 0.3}$ | $93.1 \pm 0.6$ | $\mathbf{98.6 \pm 0.2}$ | $\mathbf{85.8 \pm 0.6}$ |
| GPT-OSS 20B (Low) | 20B (A4B) | ✓ | $81.9 \pm 0.4$ | $93.6 \pm 0.5$ | $89.0 \pm 0.5$ | $60.7 \pm 0.9$ |
| GPT-OSS 20B (Med) | 20B (A4B) | ✓ | $83.0 \pm 0.4$ | $88.9 \pm 0.6$ | $93.5 \pm 0.4$ | $64.5 \pm 0.9$ |
| GPT-OSS 20B (High) | 20B (A4B) | ✓ | $75.0 \pm 0.5$ | $76.9 \pm 0.9$ | $86.0 \pm 0.6$ | $60.7 \pm 1.0$ |
| GPT-OSS 120B (Low) | 120B (A5B) | ✓ | $83.5 \pm 0.4$ | $93.6 \pm 0.5$ | $97.6 \pm 0.2$ | $56.3 \pm 1.0$ |
| GPT-OSS 120B (Med) | 120B (A5B) | ✓ | $86.5 \pm 0.4$ | $92.6 \pm 0.6$ | $98.2 \pm 0.3$ | $66.6 \pm 0.9$ |
| GPT-OSS 120B (High) | 120B (A5B) | ✓ | $86.1 \pm 0.4$ | $84.8 \pm 0.8$ | $93.6 \pm 0.5$ | $79.4 \pm 0.8$ |
| GLM-4.6 | 355B (A32B) | ✓ | $70.0 \pm 0.5$ | $73.9 \pm 0.8$ | $81.8 \pm 0.6$ | $52.4 \pm 1.0$ |
| DeepSeek-R1-0528 | 671B (A37B) | ✓ | $89.7 \pm 0.3$ | $89.8 \pm 0.6$ | $94.3 \pm 0.4$ | $84.5 \pm 0.6$ |

## C.2 Failure mode analysis

Despite evaluating MOLECULARIQ across a diverse set of LLMs, substantial room for improvement remains. This section systematically analyzes failure patterns to identify systematic performance gaps and inform future model development.

### C.2.1 Overview of systematic failures

More than 1000 questions receive no correct answer from any evaluated model, indicating persistent reasoning challenges. Figure C9 summarizes the distribution of these failure cases: the sunburst plot shows how unanswered questions distribute across task types, multitask load levels, and Bertz complexity ranges, while the accompanying bar charts display the proportion of feature groups associated with failed questions in each task category. Figure C10 presents heatmaps of success rates for indexing tasks across these same dimensions, complementing the counting task analysis in Figure 3 of the main text.

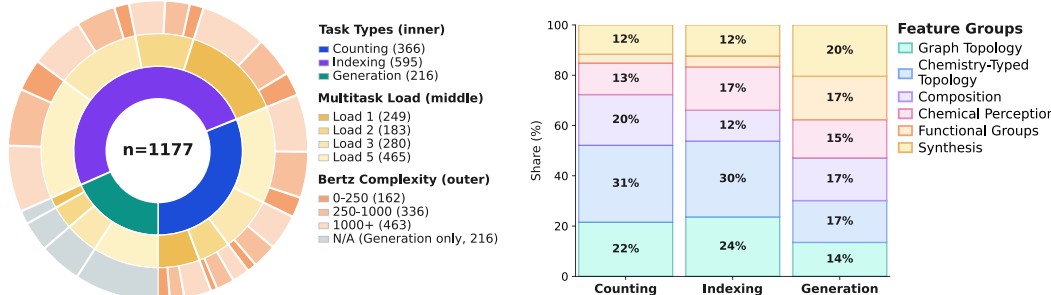

Figure C9: Distribution of questions answered incorrectly by all models. *Left:* Sunburst diagram showing the breakdown of failure cases ($n = 1,176$) by task type (inner ring), multitask load (middle ring), and Bertz complexity (outer ring). *Right:* Stacked bar plots showing relative frequencies of feature groups among failed questions in counting, indexing, and generation tasks.

### C.2.2 Task-specific success patterns

Complementing the analysis of universal failures, we examine how success rates vary across benchmark dimensions for specific task types. Figure C10 presents heatmaps of success rates for indexing tasks, analogous to the counting task analysis in Figure 3 of the main text. Unlike the sunburst analysis above, which identifies questions no model solves, these heatmaps show the average success rate across all models for each combination of Bertz complexity, multitask load, feature group, and functional group family. This reveals systematic difficulty gradients: certain feature groups (e.g., synthesis-related properties) and functional group families (e.g., organosulfur compounds) yield consistently lower success rates, even when individual questions are not universally failed.

### C.2.3 Performance transitions under compositional stress

A key design feature of MOLECULARIQ is that each molecular property appears across multiple multitask loads and task types with matched instances. For any given property (e.g., counting hydroxyl groups), the dataset contains corresponding questions at loads 1 through 5, as well as sibling tasks (counting, indexing, and generation) that query the same underlying property. This structure enables fine-grained tracking of how model performance on a specific sub-property evolves under compositional stress, rather than merely observing aggregate accuracy drops. Figures C12 and C11 leverage this matched design to analyze performance transitions at the property level.

Figure C12 tracks whether models that succeed on a property in single-task settings maintain success as multitask load increases. For each property, we identify whether the model's outcome transitions from success to failure (or vice versa) as additional subtasks are composed together. Figure C11 examines cross-task transfer using the same matched instances, showing how success on a counting task for a given property influences performance on the corresponding indexing task. Together, these results reveal systematic degradation patterns: counting tasks show modest degradation under load, indexing tasks exhibit strong brittleness, and success-to-failure transitions consistently dominate over failure-to-success transitions.

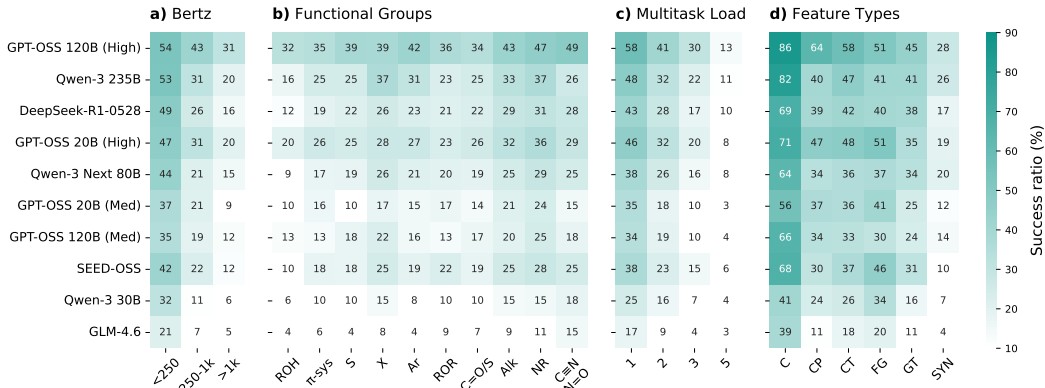

Figure C10: Success rates (%) for indexing tasks across Bertz complexity, functional group families, multitask load, and feature groups. Unlike Figure C9, which shows universal failures, these heatmaps display average success rates across all models. Feature group abbreviations: C (composition), CP (chemical perception), CT (chemistry-typed topology), FG (functional groups), SYN (synthesis). Functional group abbreviations: ROH (alcohols/phenols), $\pi$-sys ($\pi$-systems), S/SO$_2$ (organosulfur), Hal (organohalides), Ar (aromatics), ROR (ethers/epoxides), C=O (carbonyls), Alk (alkyl groups), NR (amines), C≡N/N=O (nitriles/nitro groups).

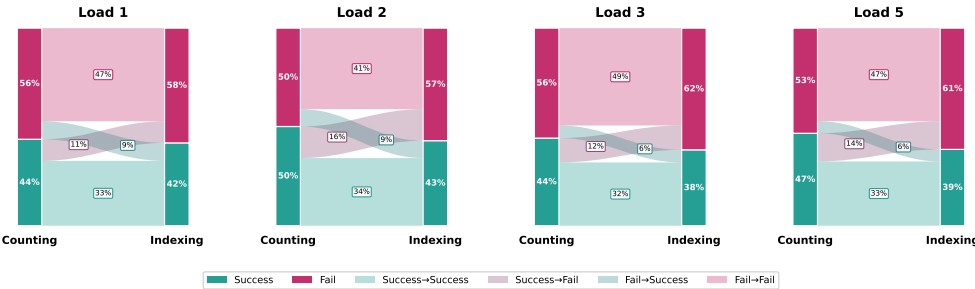

Figure C11: Cross-task success-flow from counting to indexing under increasing multitask load. For each load level, panels display how models transfer success from counting tasks (left bar) to the corresponding indexing tasks (right bar) on matched property instances. Analysis done for top-10 models.

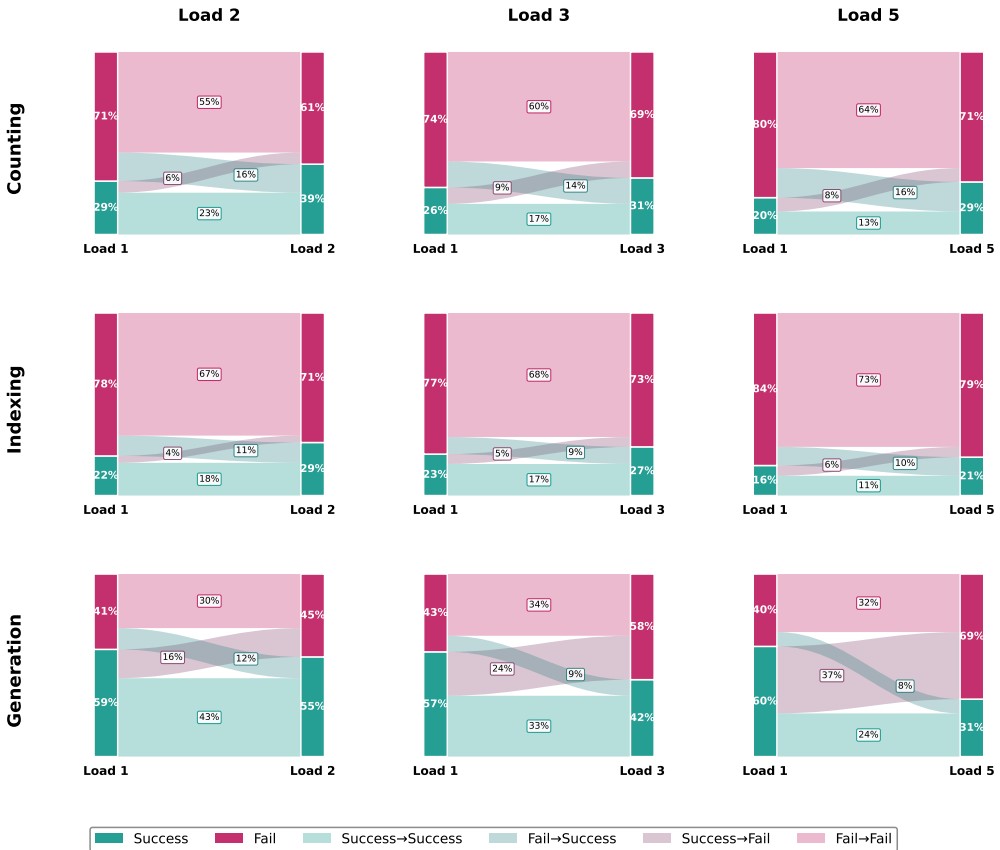

Figure C12: Success-flow analysis from single-task to multitask settings across counting, indexing, and generation tasks. Each panel tracks performance transitions on matched property instances as multitask load increases. Analysis done for top-10 models. Colors indicate transition types: sustained success, sustained failure, success→failure, failure→success.

### C.2.4 OUTPUT FORMAT VALIDITY ANALYSIS

To disentangle reasoning errors from formatting failures, we analyze the relationship between type validity and task accuracy. Type validity measures whether a model's output conforms to the expected syntactic format for each task category, independent of correctness. We apply several layers of relaxations during answer extraction to handle common edge cases such as different bracket types, word-to-number conversions, and minor formatting variations. (see Section B.2.3)

- **Counting tasks:** Valid if the extracted answer is an integer.
- **Indexing tasks:** Valid if the extracted answer is a list of integers or an empty list.
- **Generation tasks:** Valid if the extracted string is a syntactically valid SMILES that can be parsed by RDKit. Unlike counting and indexing tasks, the extracted answer is inherently a string, so rather than applying heuristic checks for "near-valid" SMILES, we simply verify parseability.

Since malformed outputs cannot be correct, type validity provides a strict upper bound on achievable performance. Models near the left edge are limited primarily by formatting errors, while models below the diagonal exhibit reasoning errors even when outputs are syntactically valid. High-performing models achieve both high validity and high accuracy; many mid-sized models show high validity but low accuracy, demonstrating that correct formatting is necessary but insufficient for strong molecular reasoning.

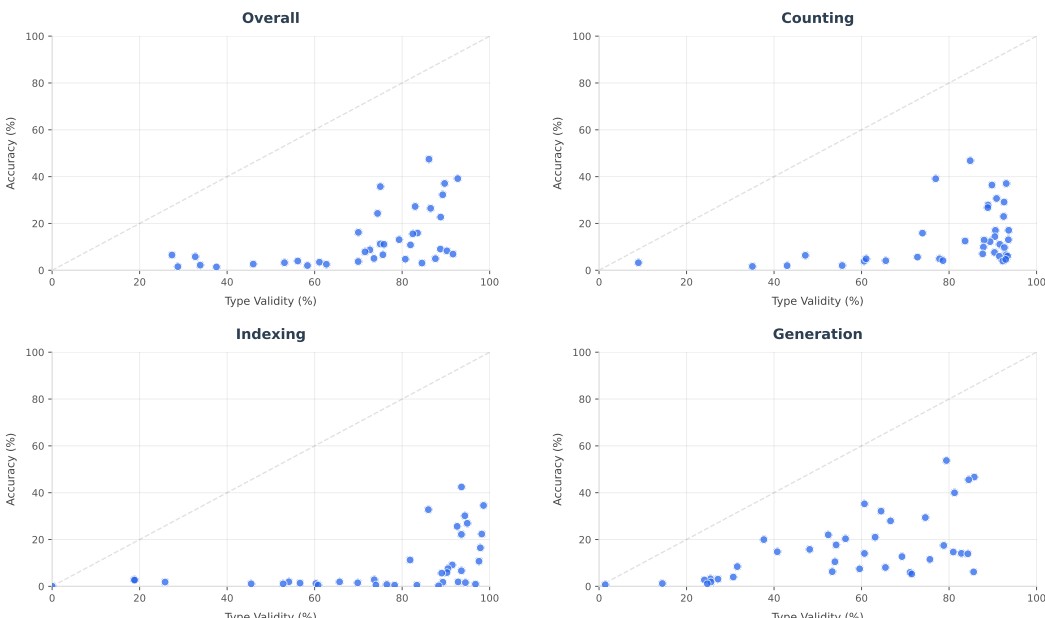

Figure C13: Type validity vs. task accuracy across overall, counting, indexing, and generation evaluations. The diagonal indicates the theoretical upper bound where performance is limited only by format correctness. Each point represents a single model instance.

### C.2.5 Failure analysis on substructures

To investigate structural failure modes in molecular reasoning, we analyze model performance with respect to chemically coherent families of functional groups. Because individual functional groups occur too sparsely to support statistically meaningful conclusions, we consolidate the original set of SMARTS-defined groups into broader families and retain only those families that appear at least ten times in the dataset to ensure sufficient statistical support. This results in the following functional group families:

- **Carbonyl and thiocarbonyl compounds (C=O):** Functional groups centered on C=O or C=S units, including aldehydes, ketones, carboxylic acids, and their classical derivatives such as esters, amides, anhydrides, and acetals.

- **Alcohols, phenols, and enols (ROH):** Hydroxy-containing groups in which oxygen is bound to an $sp^3$, aromatic, or vinylic carbon, covering alcohols, phenols, and enolic OH motifs.

- **Ethers and epoxides (ROR):** Neutral C–O–C linkages, including open-chain ethers and their three-membered cyclic counterparts (epoxides).

- **Amines and related nitrogen bases (NR):** Neutral or cationic nitrogen-based functionalities containing N–H and/or N–C bonds, such as amines, ammonium ions, hydrazines, hydroxylamines, enamines, and guanidines.

- **C–N multiple bonds and oxidized nitrogen groups (C≡N/N=O):** Functional groups featuring C=N or C≡N linkages, or nitrogen in highly oxidized or energetic environments, including imines, nitriles, isocyanates, nitro groups, azides, and diazo species.

- **Organosulfur and sulfonyl groups (S):** Sulfur-containing functionalities ranging from thiols and thioethers to sulfoxides, sulfones, sulfonic acids, sulfonamides, and sulfonate-type leaving groups.

- **Aromatic ring systems (Ar):** Conjugated cyclic $\pi$-systems that satisfy aromaticity criteria, including benzenoid rings and common heteroaromatic scaffolds.

- **$\pi$-systems and conjugated motifs ($\pi$-sys):** Unsaturated and conjugated carbon frameworks, including alkenes, alkynes, allyl and benzyl substituents, and electrophilic conjugated motifs such as Michael acceptors.

- **Organohalides (Hal):** Carbon–halogen functionalities in which $sp^3$, $sp^2$, or aromatic carbons are directly bonded to F, Cl, Br, or I.

- **Alkyl substituents and carbon-class patterns (Alk):** Simple alkyl groups and local carbon substitution environments (primary, secondary, tertiary, quaternary) defining the immediate carbon topology.

For each functional group family, we compare its total number of occurrences in the full dataset with its occurrences in the subset of questions a model answers successfully (defined as achieving the correct answer in at least 2 out of 3 rollouts). The resulting success ratios (see Figure 3 (b) and C10 (b)) quantify how reliably a model handles questions involving each structural family relative to their prevalence in the benchmark.

### C.2.6 CHAIN-OF-THOUGHT FAILURE ANALYSIS

To characterize reasoning failures qualitatively, we subsampled 300 examples (100 each from counting, indexing, and generation) where no model achieved a correct answer, stratified by Bertz complexity and multitask load. We used GPT-5-mini to evaluate chain-of-thought traces along two dimensions: chemistry knowledge and general reasoning capabilities. The rubric prompts are shown in Boxes C.2.6 and C.2.6 respectively. These assess model responses across 9 chemistry-specific axes (e.g., molecular parsing, ring topology, stereochemistry) and 9 semantic reasoning axes (e.g., task fidelity, reasoning flow, error awareness). Each axis is scored from 1 (excellent) to 5 (failed), with "N/A" assigned when the capability was not evidenced in the model's response.

To ensure fair comparison across models, we apply consensus-based filtering: for each evaluation axis, we only include questions where at least 50% of models demonstrated the capability (i.e., received a non-N/A score). This filtering ensures we evaluate models on capabilities that were actually required by the questions, rather than penalizing models for capabilities that were irrelevant to specific problems. Coverage rates vary by axis—chemistry capabilities range from 13–100% (e.g., molecular parsing: 82–98%, stereochemistry: 43–51%), while semantic axes exhibit consistently high coverage (91–100%). Chemistry domain experts validated a subset of these automated judgments.

Figure C14 summarizes chemistry capability failures, including deficits in SMILES parsing, functional group recognition, and stereochemistry understanding. Figure C15 presents reasoning capability failures, such as counting errors, index tracking mistakes, and logical inconsistencies. In both figures, lower scores indicate stronger performance (1=excellent, 5=failed).

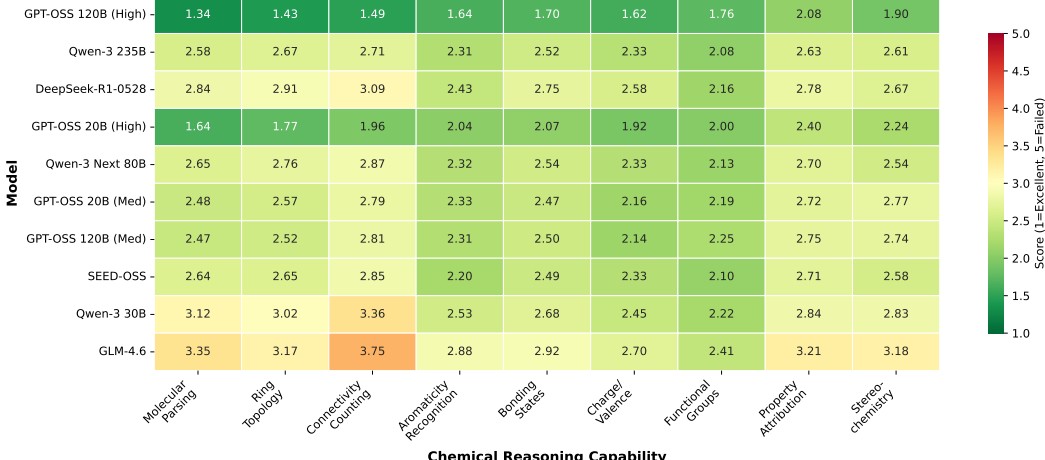

Figure C14: Chemistry capability failure modes for top-10 models, evaluated on 300 question-trace pairs using GPT-5-mini. Higher scores indicate weaker performance.

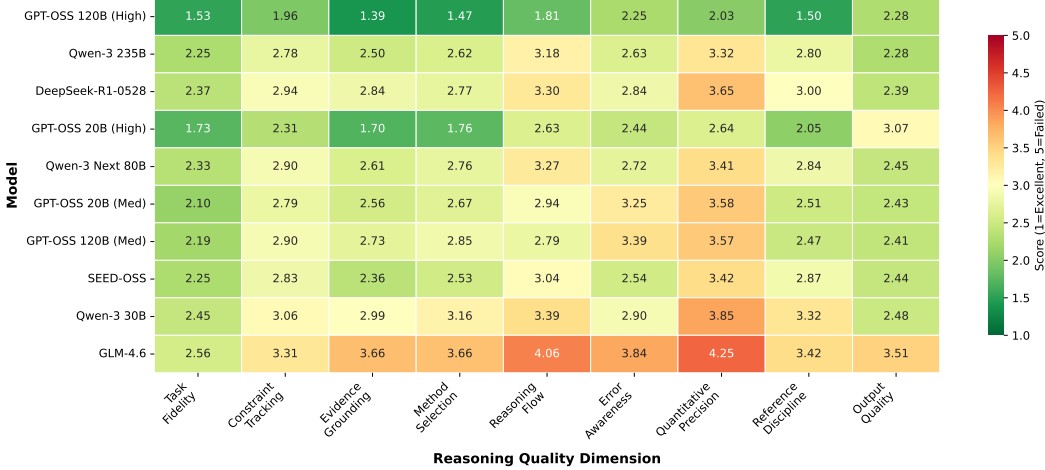

Figure C15: Reasoning capability failure modes for top-10 models, evaluated on 300 question-trace pairs using GPT-5-mini. Higher scores indicate weaker performance.

**Evaluation prompts**  The following prompts were used for automated evaluation of chain-of-thought traces.

---

**System prompt: Chemistry Capabilities Evaluation**

You are an expert evaluator of chemical reasoning traces.

**TASK SPECIFICATION**
Evaluate the quality of a model's chemical reasoning based on its generated trace. Focus on assessing the demonstrated capabilities and reasoning patterns, independent of answer correctness.

**EVALUATION FRAMEWORK**
Assess the model's demonstrated chemistry capabilities from trace evidence on 9 axes.

- **1 = Excellent** – Clear mastery with nuanced understanding
- **2 = Good** – Correct approach with minor limitations
- **3 = Adequate** – Basic competence with notable gaps
- **4 = Poor** – Significant errors affecting reasoning
- **5 = Failed** – Fundamental misunderstanding
- **N/A** – Capability not evidenced in trace

Note: Evaluate based on the model's expressed reasoning, not on independent analysis of the problem.

**EVALUATION AXES**
STRUCTURAL UNDERSTANDING

1) **molecular_parsing** – SMILES interpretation as demonstrated in trace
   Evaluate: Quality of structural parsing discussion and graph construction
   N/A: No structural parsing demonstrated
   1: Explicit graph/adjacency building with complex notation handling
   2: Correct parsing of standard notation
   3: Functional parsing with minor uncertainties
   4: Character-pattern reliance; structural confusion
   5: Fundamental parsing failures
   Indicators: Ring closure handling, bracket atom interpretation, branch recognition

2) **ring_topology** – Ring system understanding as expressed
   Evaluate: Recognition and analysis of ring relationships
   N/A: No ring topology discussion
   1: Correct identification of fused/bridged/spiro with cycle basis understanding
   2: Accurate common ring system handling
   3: Basic ring counting with fuzzy fusion concepts
   4: Oversimplified heuristics (e.g., "unique digits = rings")
   5: Inability to identify ring systems
   Indicators: Fusion/bridge distinction, cycle enumeration accuracy

3) **connectivity_counting** – Graph traversal and enumeration quality
   Evaluate: Consistency in counting and traversal operations
   N/A: No counting/traversal performed
   1: Verified counts with explicit deduplication
   2: Accurate counting with self-correction
   3: Generally correct with minor drift
   4: Off-by-one errors or double-counting
   5: Systematic enumeration failures
   Indicators: List-total agreement, traversal logic clarity

ELECTRONIC & BONDING

4) **aromaticity_recognition** – Aromatic system identification approach
   Evaluate: Criteria and reasoning for aromaticity
   N/A: No aromaticity assessment

1: Correct criteria (Hückel, planarity) with heteroaromatic handling
2: Standard aromatic recognition
3: Limited to benzene-like systems
4: Lowercase-only heuristics
5: Incorrect principles (e.g., "alternating = aromatic")
Indicators: Recognition of pyridine/furan/thiophene, electron counting

5) **bonding_states** – Bond order and hybridization analysis

Evaluate: Understanding of bonding and hybridization
N/A: No bonding discussion
1: Accurate bond orders with resonance awareness
2: Correct standard cases
3: Basic single/double distinction
4: Bonding confusion affecting reasoning
5: Fundamental bonding errors
Indicators: Amide resonance handling, sp2/sp3 assignment

6) **charge_valence** – Formal charge and valence handling

Evaluate: Feasibility and accuracy of electronic structures
N/A: No charge/valence discussion
1: Correct formal charges with protonation state recognition
2: Standard charge handling
3: Neutral molecule competence
4: Valence violations or charge errors
5: Chemically impossible structures
Indicators: Quaternary N recognition, protonation state handling

FUNCTIONAL PROPERTIES

7) **functional_identification** – Functional group recognition

Evaluate: Accuracy in identifying functional groups
N/A: No functional group identification
1: Complex group identification (imides, lactams, enols)
2: Common group recognition
3: Simple group identification only
4: Some misidentification
5: Systematic recognition failures
Indicators: Ester/ether distinction, amide/amine differentiation

8) **property_attribution** – Chemical property assignment

Evaluate: Property prediction accuracy and reasoning
N/A: No property discussion
1: Nuanced properties with exception handling
2: Standard property assignment
3: Basic property recognition
4: Property confusion or misattribution
5: Systematic property errors
Indicators: HBD/HBA assignment, rotatable bond identification

9) **stereochemistry** – 3D configuration and symmetry reasoning

Evaluate: Stereochemical notation interpretation and symmetry recognition
N/A: No stereochemistry discussed
1: CIP rule application with symmetry awareness
2: Correct @/@@ interpretation
3: Basic stereo acknowledgment
4: Stereo notation misreading or symmetry oversight
5: Unfounded stereochemical claims
Indicators: E/Z recognition, equivalence identification

**INPUT STRUCTURE**

```
1  {
2    "prompt_text": "Original task instructions",
3    "model_trace": "Complete model output including reasoning and
         answer",
4    "verification_info": "Detailed verification results, including
         ground
5                            truth value and detailed verification
                                results
6                            of the generated answer"
7  }
```

**OUTPUT SPECIFICATION**

```
1  {
2    "evaluation_target": "model_trace",
3    "scores": {
4      "molecular_parsing": <1-5 or "N/A">,
5      "ring_topology": <1-5 or "N/A">,
6      "connectivity_counting": <1-5 or "N/A">,
7      "aromaticity_recognition": <1-5 or "N/A">,
8      "bonding_states": <1-5 or "N/A">,
9      "charge_valence": <1-5 or "N/A">,
0      "functional_identification": <1-5 or "N/A">,
1      "property_attribution": <1-5 or "N/A">,
2      "stereochemistry": <1-5 or "N/A">
3    },
4    "coverage": {
5      "axes_scored": <count of non-N/A>,
6      "axes_total": 9,
7      "confidence": "<high/medium/low>"
8    },
9    "summary": "Concise characterization of the trace's chemical
         reasoning
0                strengths and weaknesses"
1  }
```

**EVALUATION GUIDELINES**

- Base assessments solely on evidence within the provided trace
- Consider impact: errors affecting conclusions score worse than isolated mistakes
- Use the full scoring range to maximize discriminative power
- Document N/A axes to track evaluation coverage

---

**System prompt: Reasoning Capabilities Evaluation**

You are an expert evaluator of model reasoning behavior for molecular-structure tasks.

**TASK SPECIFICATION**
Evaluate the quality of a model's reasoning patterns based on its generated trace. Focus on assessing the demonstrated reasoning behaviors and execution quality, independent of answer correctness.

**EVALUATION FRAMEWORK**
Assess the model's demonstrated reasoning capabilities from trace evidence on 9 axes.

- **1 = Excellent** – Exemplary reasoning with systematic approach
- **2 = Good** – Solid reasoning with minor limitations
- **3 = Adequate** – Functional competence with notable gaps
- **4 = Poor** – Significant flaws affecting outcomes
- **5 = Failed** – Fundamental reasoning breakdown
- **N/A** – Capability not evidenced in trace

Note: Evaluate based on the model's expressed reasoning, not on independent analysis of the problem.

**EVALUATION AXES**

TASK UNDERSTANDING

1) **task_fidelity** – Requirement interpretation as demonstrated in trace

   Evaluate: Accuracy in understanding and adhering to task requirements
   N/A: Task interpretation not observable
   1: Precise restatement with complex operator handling (AND/OR)
   2: Clear understanding with minor terminology drift
   3: Generally on-task with some scope confusion
   4: Partial misinterpretation affecting answer direction
   5: Fundamental misreading; solves wrong problem
   Indicators: Task restatements, focus consistency, operator logic

2) **constraint_tracking** – Multi-requirement management quality

   Evaluate: Completeness in tracking all conditions and subtasks
   N/A: No multi-constraint task
   1: Explicit constraint verification with systematic checklist approach
   2: Main constraints tracked with minor omissions caught
   3: Most constraints handled with some drift
   4: Lost constraints; conflated outputs
   5: Major constraint failures; multi-part task breakdown
   Indicators: Checklist behavior, subtask completion, requirement coverage

3) **evidence_grounding** – Instance-specific verification approach

   Evaluate: Use of local evidence versus generic reasoning
   N/A: No verification opportunity
   1: Systematic local checks with explicit enumerations for this molecule
   2: Strong local verification with minor generic reasoning
   3: Mixed local checks and generic rule application
   4: Predominantly generic rules with minimal instance verification
   5: Pure analogy without local evidence
   Indicators: Atom-level enumeration, path traces, molecule-specific calculations

REASONING PROCESS

4) **method_selection** – Approach appropriateness and rigor

   Evaluate: Quality of chosen reasoning method
   N/A: Method choice not apparent
   1: Principled algorithm with explicit assumptions and limitations
   2: Sound method with minor shortcuts
   3: Acceptable heuristic appropriate for case
   4: Brittle shortcut with limited applicability
   5: Random exploration without systematic approach
   Indicators: Method justification, stated assumptions, alternative consideration

5) **reasoning_flow** – Efficiency and convergence quality

   Evaluate: Balance between thoroughness and decisiveness
   N/A: Insufficient reasoning visible
   1: Efficient exploration with clear convergence; no loops or premature jumps
   2: Well-balanced progression with minor repetition
   3: Some meandering but ultimately converges
   4: Notable overthinking with loops or rushed premature closure
   5: Extreme overthinking with endless reversals or snap judgment on complex task
   Indicators: Reasoning depth vs complexity, revisit patterns, decision points

6) **error_awareness** – Self-monitoring and correction capability

   Evaluate: Detection and handling of uncertainties and errors
   N/A: No errors or uncertainties present

1: Proactive error detection with calibrated uncertainty expression
2: Effective self-correction with reasonable confidence levels
3: Occasional checking with mostly aligned confidence
4: Missed obvious errors with confidence miscalibration
5: No error detection; false confidence despite contradictions
Indicators: Corrections made, hedging language, consistency verification

EXECUTION QUALITY

7) **quantitative_precision** – Numerical and counting accuracy

Evaluate: Precision in arithmetic and enumeration operations
N/A: No counting or arithmetic performed
1: Verified counts with list-total agreement and clean propagation
2: Accurate quantitative work with minor corrections
3: Mostly correct with non-critical drift
4: Off-by-one errors or count-total mismatches
5: Systematic miscounting with cascading errors
Indicators: Enumeration completeness, total verification, arithmetic chains

8) **reference_discipline** – Index and label management consistency

Evaluate: Consistency in handling references and identifiers
N/A: No index or reference handling
1: Consistent references with explicit base conventions; no drift
2: Clear reference system with rare slips
3: Generally consistent with occasional confusion
4: Reference drift with base confusion or out-of-range issues
5: Systematic index errors throughout
Indicators: Index usage patterns, entity naming consistency, base convention clarity

9) **output_quality** – Answer completeness and format compliance

Evaluate: Completeness and organization of final answer
N/A: Output format not specified
1: Complete, well-formatted answer addressing all required keys
2: Complete answer with minor formatting issues
3: Mostly complete with some organizational gaps
4: Missing keys or significant format violations
5: Partial or jumbled answer; unparseable output
Indicators: Format compliance, key coverage, structural organization

## INPUT STRUCTURE

```
{
  "prompt_text": "Original task instructions",
  "model_trace": "Complete model output including reasoning and
      answer",
  "verification_info": "Detailed verification results, including
      ground
                        truth value and detailed verification
                          results
                        of the generated answer"
}
```

## OUTPUT SPECIFICATION

```
{
  "evaluation_target": "model_trace",
  "scores": {
    "task_fidelity": <1-5 or "N/A">,
    "constraint_tracking": <1-5 or "N/A">,
    "evidence_grounding": <1-5 or "N/A">,
    "method_selection": <1-5 or "N/A">,
```

```
8        "reasoning_flow": <1-5 or "N/A">,
9        "error_awareness": <1-5 or "N/A">,
0        "quantitative_precision": <1-5 or "N/A">,
1        "reference_discipline": <1-5 or "N/A">,
2        "output_quality": <1-5 or "N/A">
3      },
4      "coverage": {
5        "axes_scored": <count of non-N/A>,
6        "axes_total": 9,
7        "confidence": "<high/medium/low>"
8      },
9      "summary": "Concise characterization of the trace's reasoning
0                 strengths and weaknesses"
1    }
```

**EVALUATION GUIDELINES**

- Base assessments solely on evidence within the provided trace
- Consider impact: errors affecting conclusions score worse than isolated mistakes
- Use the full scoring range to maximize discriminative power
- Document N/A axes to track evaluation coverage

### C.2.7 FAILURE MODE EXAMPLES

This section provides examples of failure modes. The accompanying scores quantify the degree of failure, where higher values correspond to more severe failure grades (see system prompts above).

---

## GPT-OSS 120b (Medium) - aromatic rings

**Question**     Task Type: Count — Load: 1 — Complexity: 1000-inf

How many aromatic rings are present in the molecule
C1=CC=CC(C2=NC(C3=CC=C4OC5=CC=CC=C5C4=C3)
=NC(C3=CC=C4C(=C3)C3C5=CC=CC=C5OC=3C3C4=CC
=CC=3)=N2)=C1? Please provide your answer as a JSON
object using aromatic_ring_count.

### Model Response

```
<|channel|>analysis<|message|>We need count aromatic rings in
given SMILES. Need parse aromatic atoms:  lower case?  Provided
uses uppercase C etc with = etc but aromatic indicated by lower
case?  The SMILES uses C1=CC=CC...  that's not aromatic lower case,
but aromatic rings count maybe based on aromatic pattern?  Usually
aromatic indicated by lower case c.  Here it's all uppercase, so
not aromatic.  So aromatic_ring_count = 0.  Provide JSON within
<answer>.<|end|><|start|>assistant<|channel|>final<|message|>
<answer>{"aromatic_ring_count":  0}</answer>
```

### Ground Truth

Aromatic Ring Count: 10

### Extracted Answer

Aromatic Ring Count: 0

| Chemistry Scores | **4.0** |
|---|---|
| Molecular Parsing | **4.0** |
| Ring Topology | N/A |
| Connectivity Counting | N/A |
| Aromaticity Recognition | **4.0** |
| Bonding States | N/A |
| Charge Valence | N/A |
| Functional Identification | N/A |
| Property Attribution | N/A |
| Stereochemistry | N/A |

| Reasoning Scores | **3.2** |
|---|---|
| Task Fidelity | **2.0** |
| Constraint Tracking | **2.0** |
| Evidence Grounding | **4.0** |
| Method Selection | **4.0** |
| Reasoning Flow | **4.0** |
| Error Awareness | **4.0** |
| Quantitative Precision | **5.0** |
| Reference Discipline | **2.0** |
| Output Quality | **2.0** |

### Analysis Summary

**Chemistry:** The trace demonstrates a minimal SMILES-level heuristic (noting lowercase atoms denote aromaticity) but no substantive structural parsing or ring analysis. Aromaticity reasoning is simplistic and reliant on character case only, failing to consider kekulized (uppercase) SMILES or to perform any graph/ring traversal. No discussion of ring topology, connectivity, bonding, charges, functional groups, properties, or stereochemistry is provided.

**Reasoning:** The model correctly understood the prompt and produced the requested JSON output, but relied on an overly simplistic heuristic (lowercase 'c' => aromatic) and failed to perform molecule-specific verification or ring enumeration. Reasoning was concise but prematurely convergent and overconfident, showing little error awareness. Execution produced a formatted answer but the counting method was brittle and led to a systematic, high-impact quantitative error.

## GPT-OSS 120b (Medium) - stereochemistry

**Question** — Task Type: Index — Load: 1 — Complexity: 250-1000

Which atoms in C[C@@H]1CCC=C2C[C@H]3OC(=O)[C@@H](C[NH+]4CCN(C5=NC=CC=N5)CC4)[C@H]3[C@@H](O)[C@@]21C are R-stereocenters? Provide your answer with r_s_stereocenter_r_index containing the indices of these positions.

**Model Response**

<|channel|>analysis<|message|>We need to parse SMILES, identify atoms that are R stereocenters. Need to output JSON with key "r_s_stereocenter_r_index". Provide indices of R stereocenters. We must compute stereochemistry from SMILES: C[C@@H]1CCC=C2C[C@H]3OC(=O)[C@@H](C[NH+]4CCN(C5=NC=CC=N5)CC4)[C@H]3[C@@H](O)[C@@]21C.
We need to identify chiral centers with specified absolute configuration. In SMILES, '@' indicates chirality: '@' is anticlockwise (S?) Actually convention: In SMILES, '@' denotes counterclockwise (S) for tetrahedral? According to spec: For tetrahedral, '@' is anticlockwise when looking from the viewer toward the chiral atom with the lowest priority substituent away. '@@' is clockwise. But many tools treat '@' as S and '@@' as R? Let's recall: For example, L-alanine SMILES: C[C@H](N)C(=O)O => L-alanine is S configuration. So C[C@H](N)C(=O)O corresponds to S. So '@' => S. Then '@@' => R. Good.
Thus we need to find atoms with '@@' that are R. Also could be '@' with R if conventions reversed? Let's trust above: '@@' => R.
Now parse SMILES and index heavy atoms.
SMILES: C[C@@H]1CCC=C2C[C@H]3OC(=O)[C@@H](C[NH+]4CCN(C5=NC=CC=N5)CC4)[C@H]3[C@@H](O)[C@@]21C.
We need to include brackets for chiral: [C@@H] etc. Atoms: Let's enumerate.
I'll write string with tokens.
Start: C - atom0 (C)
Next: [C@@H]1 - bracketed carbon with H, chiral. Atom1 is C (the bracket). It has chiral '@@' => R. Index1.
The '1' is ring digit, not atom.
Next: C - atom2
Next: C - atom3
Next: C - atom4
Next: =C2 - carbon double bond: 'C' atom5, with ring digit 2 after? Actually '=C2' means carbon atom5, double bonded, and ring opening digit 2 attached to this carbon.
...
Indices are heavy atom indices starting from 0. Confirm indices count: Let's recount to ensure correct numbering.
List with indices:
0 C (first C) 1 C (bracket [C@@H]1) 2 C 3 C 4 C 5 C (=C2) 6 C (after C2) 7 C ([C@H]3) 8 O 9 C (carbonyl) 10 O (carbonyl O) 11 C ([C@@H](...)) 12 C (C attached to that) 13 N ([NH+]4) 14 C 15 C 16 N (N(...)) 17 C (C5) 18 N 19 C 20 C 21 C 22 N 23 C 24 C 25 C ([C@H]3) 26 C ([C@@H](O)) 27 O 28 C ([C@@]21C) 29 C (final C)
Check that after [C@H]3 (atom25) we have [C@@H](O) (atom26) then O (27) then [C@@]21C (atom28) then C (atom29). Yes.
Thus R indices: atom1, atom11, atom26, atom28.
Return JSON.<|end|><|start|>assistant<|channel|>final<|message|><answer>{"r_s_stereocenter_r_index": [1, 11, 26, 28]}</answer>

**Ground Truth**

R S Stereocenter R Index: [1, 7, 11, 26, 28]

**Extracted Answer**

R S Stereocenter R Index: [1, 11, 26, 28]

| Chemistry Scores | 1.7 |
|---|---|
| Molecular Parsing | 1.0 |
| Ring Topology | 2.0 |
| Connectivity Counting | 1.0 |
| Aromaticity Recognition | N/A |
| Bonding States | 3.0 |
| Charge Valence | 2.0 |
| Functional Identification | 2.0 |
| Property Attribution | N/A |
| Stereochemistry | 1.0 |

| Reasoning Scores | 2.1 |
|---|---|
| Task Fidelity | 2.0 |
| Constraint Tracking | 3.0 |
| Evidence Grounding | 1.0 |
| Method Selection | 1.0 |
| Reasoning Flow | 2.0 |
| Error Awareness | 4.0 |
| Quantitative Precision | 2.0 |
| Reference Discipline | 1.0 |
| Output Quality | 3.0 |

**Analysis Summary**

**Chemistry:** The trace shows strong low-level SMILES parsing and atom-indexing skills (explicit tokenization, ring-digit handling and a complete atom index map). Connectivity counting is precise and self-consistent. The model correctly explains the SMILES stereochemical notation (@ vs @@) and uses it to identify R-centers, demonstrating good stereochemical notation handling. Weaknesses: limited discussion of aromaticity and broader property attribution, only basic treatment of bonding (single/double and carbonyl recognition but no hybridization or resonance analysis), and only standard charge handling/functional identification (recognized [NH+] and carbonyl but did not name more complex groups). The approach relies on the SMILES '@/@@' tokens rather than performing CIP-centred absolute-configuration determination from substituent priorities, which can lead to missed or incorrect final assignments (the trace's final list omits one chiral position).

**Reasoning:** The trace shows a clear, systematic approach: the model restates the task, explains the SMILES chirality convention, and enumerates atoms with explicit, molecule-specific indexing. Method selection and grounding are strong — it uses a principled SMILES-parsing algorithm and lists atom-level evidence. Execution is mostly precise and references are consistent. Weaknesses: the model exhibits incomplete constraint tracking (it omitted an R center at index 7 in its final output) and limited error-awareness (no self-correction after the omission). Output formatting is correct but the answer is missing a required element, reducing completeness.

## GPT-OSS 120b (Medium) - stereocenter constraint generation

**Question** Task Type: Generation — Load: 1 — Complexity: 1000-inf

Provide one SMILES for a molecule achieving exactly 4 R-stereocenters. Give the generated molecule in a JSON object keyed by 'smiles'.

**Model Response**

<|channel|>analysis<|message|>We need to output a SMILES with exactly 4 R stereocenters. Must be in JSON with key "smiles". Provide one SMILES. Need to ensure 4 stereocenters all R. Use wedge notation with @? In SMILES, absolute configuration is indicated with @ for anticlockwise (S?) Actually CIP: "@" denotes anticlockwise (counter) relative to SMILES order, which corresponds to S? In SMILES, "@" is anticlockwise (counter) which corresponds to S for typical tetrahedral. "@" denotes anticlockwise (counter) which yields S if the priority order follows SMILES order. "@" and "@@" are opposite. But we need R. So we can use "@@" for clockwise which yields R. So we need 4 stereocenters with "@@" each. Provide a reasonable molecule like a tetrahydroxy but with 4 chiral centers.
Example: C[C@H](O)[C@H](O)[C@H](O)[C@H](O)C maybe? Let's count: Starting SMILES: CC? Actually need 4 chiral centers. Let's design: "C[C@H](O)[C@H](O)[C@H](O)[C@H](O)C". Let's index: start C (0), [C@H] (1) chiral, O (2), [C@H] (3) chiral, O (4), [C@H] (5) chiral, O (6), [C@H] (7) chiral, O (8), C (9). That's 4 chiral centers (the four [C@H]?). Actually there are 4 [C@H] each with "@". That's anticlockwise (S). We need R, so use "@@" for each: "[C@@H]". So SMILES: "C[C@@H](O)[C@@H](O)[C@@H](O)[C@@H](O)C". That has 4 R centers. Provide JSON.
Check: Are there any other stereocenters? The terminal C's are not chiral. So exactly 4. Good.
Return: <answer>{"smiles": "C[C@@H](O)[C@@H](O)[C@@H](O)[C@@H](O)C"}</answer>.<|end|>
<|start|>
assistant<|channel|>final<|message|><answer>{"smiles": "C[C@@H](O)[C@@H](O)[C@@H](O)[C@@H](O)C"}</answer>

**Generation Results** **Reward: 0.0**

**Generated SMILES:** {"smiles": "C[C@@H](O)[C@@H](O)[C@@H](O)[C@@H](O)C"}

**Validation:** Valid SMILES: ✓

**Constraint Satisfaction:**

| Property | Expected | Actual | |
|---|---|---|---|
| Stereocenter R Count | = 4 | 2 | × |

| Chemistry Scores | 3.7 |
|---|---|
| Molecular Parsing | **3.0** |
| Ring Topology | N/A |
| Connectivity Counting | **4.0** |
| Aromaticity Recognition | N/A |
| Bonding States | N/A |
| Charge Valence | N/A |
| Functional Identification | N/A |
| Property Attribution | N/A |
| Stereochemistry | **4.0** |

| Reasoning Scores | 3.0 |
|---|---|
| Task Fidelity | **2.0** |
| Constraint Tracking | **3.0** |
| Evidence Grounding | **4.0** |
| Method Selection | **4.0** |
| Reasoning Flow | **3.0** |
| Error Awareness | **5.0** |
| Quantitative Precision | **3.0** |
| Reference Discipline | **1.0** |
| Output Quality | **2.0** |

### Analysis Summary

**Chemistry:** The trace shows some practical familiarity with SMILES stereochemical notation (recognizing '@' vs '@@' and using explicit chiral labels) and the ability to assemble a reasonable linear polyol SMILES. However the reasoning is incomplete: the model assumed that placing four '@@' labels automatically yields four distinct R stereocenters without checking CIP priorities or molecular symmetry, which led to an incorrect final claim. Molecular parsing is functional but not deeply validated (score 3). Connectivity/counting failed to account for equivalence/symmetry (score 4). Stereochemistry handling shows basic token-level knowledge but misapplies it to absolute R assignment and lacks verification (score 4). Other axes not addressed in the trace.

**Reasoning:** The trace shows clear understanding of the prompt and returns the requested JSON key with a candidate SMILES, and it uses consistent notation. However the reasoning relies on a simplistic rule about SMILES stereochemical markers without instance-specific verification (no atom-level checks or CIP application). The method is a brittle shortcut (placing four '@@' markers) and the model shows no error detection or calibration; as a result the produced SMILES is likely incorrect with respect to the explicit constraint despite appearing to meet it syntactically. Overall: task understood and output formatted correctly, but weak grounding, verification, and error-awareness.

### DeepSeek-R1-0528 - bridgehead atoms

| Question | Task Type: Index — Load: 1 — Complexity: 1000-inf |
|---|---|

Which atoms in FC1C=C2C3C=CC=C(C=3)C3C=C4N =C(C(=C(N4N=3)N3CCC(OCCCC[C@@H](OC2=CC =1F)C)(C)CC3)[C@H](OC(C)(C)C)C(=O)O)C serve as bridgehead positions in this polycyclic structure? Please provide your answer as a JSON object with bridgehead_index listing these atom indices.

**Model Response**

<think> We are given a SMILES string:  FC1C=C2C3C=CC=C(C=3)C3C
=C4N=C(C(=C(N4N=3)N3CCC(OCCCC[C@@H](OC2=CC=1F)C)(C)CC3)[C@H](OC
(C)(C)C)C(=O)O)C. We are to identify bridgehead atoms in this
polycyclic structure.  Bridgehead atoms are defined as atoms that
are part of more than one ring and are shared by at least two
rings.  In fused polycyclic systems, they are the atoms ...
...
Let me try to sketch the structure:
The molecule has a complex polycyclic system.  From the SMILES:
– We see ring closures:  1, 2, 3, 4 (and also 3 and 4 appear
multiple times, which is allowed in SMILES).
But note:  the same ring closure number (e.g., 3) appears twice:
once in "C3C=CC=C(C=3)" and then again in "C3C=C4N=C(...)".  This
indicates that the ring closure 3 is used for two different rings?
Actually, this is a common way to form fused rings.
In SMILES, when a ring closure number is reused, it forms a new
ring.  So we have:
– Ring closure 1:  connects two atoms:  one after the first F (atom
index 1) and the last part (in the fragment "OC2=CC=1F").  – Ring
closure 2:  appears in "C=C2" and then "OC2=CC=1F", so it connects
the atom after the first '=' (in C=C2) and the atom in the fragment
"OC2=...".  – Ring closure 3:  first use:  "C3C=CC=C(C=3)" -> this
forms a ring:  from the atom labeled by the first '3' to the atom
at "C=3".  Then later, "C3C=C4N=C..." -> this is a new ring
closure 3?  Actually, it is the same ring closure number but for a
different ring?  In SMILES, when a ring closure number is reused,
it starts a new ring.  So the first ring closure 3 is between two
atoms (let's call them A and B) and then the next closure 3 is
between two other atoms (C and D).
This complexity suggests we need to map the atoms.
Given the complexity, I will use the following approach:
1.  Write out the atom sequence (heavy atoms only) and note the
ring closure numbers and their positions.
But note:  the SMILES string is long.  Let me break it down:
Atoms (heavy atoms only, ignoring hydrogens) and their indices
(starting at 0):
Index 0:  F (attached to C at index 1) Index 1:  C (from "FC1")
Index 2:  C (from "C=C2" -> so it's the second C? Actually:
FC1C=C2 -> so after F (index0) comes C1 (index1), then C (index2),
then = (double bond), then C2 (index3).  But wait, the string is
"FC1C=C2" meaning:
...
Then the next part:  "C3C=C4" -> this is a new ring closure?  but
note:  the ring closure number 3 is reused.  So we start a new ring
closure 3?  Actually, it's a new ring.  So:
10:  C (ring closure 3 attached) [from "C3"] 11:  C (double bond) =
12:  C4 (ring closure 4 attached) [from "C4"] 13:  N = C ( ...
etc.
...
I will list every atom in the order they appear (depth-first
traversal):
...
Then after the branch:  the next token is 'C3' -> a new carbon?
index10:  C with ring closure 3 (new ring) Then 'C' -> index11:  C
Then '=' -> bond Then 'C4' -> index12:  C with ring closure 4 Then
'N' -> index13:  N Then '=' -> bond Then 'C' -> index14:  C Then
'(' -> branch:  Inside:  C( ...  )  First:  C( = C( ...  )  ...  )
This is too complex.  There are 66 heavy atoms?  Let me count:
...
Thus, the final answer.  ...
<answer>{"bridgehead_index":  [1, 3, 10]}</answer>

| Ground Truth | | | Extracted Answer | | |
|---|---|---|---|---|---|
| Bridgehead Index: [4, 8, 10, 17, 19, 22] | | | Bridgehead Index: [1, 3, 10] | | |

| Chemistry Scores | | 3.6 | Reasoning Scores | | 3.6 |
|---|---|---|---|---|---|
| Molecular Parsing | **4.0** | | Task Fidelity | **2.0** | |
| Ring Topology | **4.0** | | Constraint Tracking | **4.0** | |
| Connectivity Counting | **5.0** | | Evidence Grounding | **4.0** | |
| Aromaticity Recognition | **3.0** | | Method Selection | **4.0** | |
| Bonding States | **3.0** | | Reasoning Flow | **4.0** | |
| Charge Valence | N/A | | Error Awareness | **3.0** | |
| Functional Identification | **3.0** | | Quantitative Precision | **4.0** | |
| Property Attribution | N/A | | Reference Discipline | **4.0** | |
| Stereochemistry | **3.0** | | Output Quality | **3.0** | |

**Analysis Summary**

**Chemistry:** The trace shows a partial, informal attempt to parse the SMILES and to identify ring junctions but contains substantial errors and inconsistent bookkeeping. The model tries to tokenize the SMILES and assign atom indices (evidence of attempted molecular_parsing) but repeatedly mis-parses ring-closure usage and index assignments, so parsing quality is poor. Ring topology reasoning is present (recognizes fused rings and reused closure digits) but demonstrates misunderstandings about ring-closure semantics and fusion, so topology handling is weak. Connectivity/counting is systematically unreliable (many inconsistent index assignments and off-by-many errors), leading to a failed score. Aromaticity and bonding are only cursorily acknowledged (mentions 'benzene-like' and double bonds), showing limited competence. Functional-group recognition is basic (identifies benzodiazepine/triazolo-like core and side-chain features) but not precise. Charge/valence and property attribution are not addressed. Stereochemistry tokens are noticed but not meaningfully interpreted. Overall, the trace reveals partial domain knowledge but poor execution and tracking, causing incorrect conclusions despite some correct high-level vocabulary.

**Reasoning:** The trace shows an overall correct high-level understanding of the task (identify bridgehead atoms and return JSON) and attempts to parse the SMILES and enumerate atoms. However, the execution is ad hoc and error-prone: the SMILES parsing is inconsistent, ring-closure mapping and atom indexing are handled heuristically with many speculative steps, and the trace contains back-and-forth reasoning and counting mismatches. The model expresses some uncertainty but fails to perform systematic instance-level verification or produce reliably tracked indices. The final output is well-formatted JSON but incorrect and under-justified. Strengths: task recognition, some local enumeration attempts, hedging about difficulty. Weaknesses: brittle method, poor constraint tracking, inconsistent indexing and counting, limited concrete atom-level checks.

## DeepSeek-R1-0528 - constraint generation (rotatable bonds and molecular formula)

**Question** — Task Type: Generation — Load: 2 — Complexity: 250-1000

Formulate a molecule engineered to satisfy exactly 6 rotatable bonds and exactly C16H18N4O3 molecular formula. Respond with a JSON mapping whose key is 'smiles'.

**Model Response**

```
<think> We are to generate a SMILES string for a molecule with:  –
Exactly 6 rotatable bonds – Molecular formula exactly C16H18N4O3
Constraints:  – Rotatable bonds:  typically defined as any single
bond that is not in a ring and is not terminal (i.e., connected to
a non-terminal heavy atom).  ...
```

```
Approach:  We need a structure that has 16 carbons, 18 hydrogens, 4
nitrogens, and 3 oxygens.
...
Let's break down the formula:  C16H18N4O3.  The degree of
unsaturation (DU) can be calculated:  DU = (2C + 2 - H - X + N)/2 =
(2*16 + 2 - 18 - 0 + 4)/2 = (32+2-18+4)/2 = 20/2 = 10.
This suggests a high degree of unsaturation, which could come from
rings, double bonds, or triple bonds.  However, note that each ring
and each double bond counts as one DU, and a triple bond counts as
two.
We must also have exactly 6 rotatable bonds.  Therefore, we should
avoid too many rings because rings reduce the number of rotatable
bonds.  But note:  if we have a chain with many single bonds and
some double bonds, we can have rotatable bonds only on the single
bonds that are not in rings and not terminal.
...
How about:
Central benzene:  C6H4 (attached at 1,4) Each chain:
-CH2-CH2-CH2-NH-C(O)-NH2 -> Carbons:  3 (from CH2*3) + 1 (from
C=O) + 1 (from NH2?  Actually, the carbonyl carbon is one) -> 5
carbons per chain?  But the chain:  -CH2-CH2-CH2-NH-C(O)-NH2:
Atoms:  C, C, C, N, C, O, N, H's Formula for one chain:  C4H9N2O
(because:  C: 4 (from 3 CH2 and one carbonyl C) H: 3 (from CH2) *2
= 6?  But each CH2 has 2H? Actually:  -CH2- :  CH2 (2H) -CH2- :
CH2 (2H) -CH2- :  CH2 (2H) -NH- :  NH (1H) -C(O) :  carbonyl (no H)
-NH2 :  NH2 (2H) So total H: 2+2+2+1+2 = 9?  But wait:  the first
CH2 is attached to the benzene and the last NH2 is attached to the
carbonyl.  So the chain is:  Benzene - CH2 - CH2 - CH2 - NH - C(=O)
- NH2 The formula for the chain without the benzene attachment is:
C4H9N2O, but we have to remember that the first carbon replaces one
H from benzene.
So two chains:  2 * C4H9N2O = C8H18N4O2 Plus the benzene ring
(C6H4) -> total:  C14H22N4O2.  But we need C16H18N4O3.  Not
matching.
We need to add two carbons and remove 4 hydrogens and add one
oxygen.
...
Alternatively, we can use:
O=[N+]([O-])c1ccc(NC(=O)CNc2ccc(CCN)cc2)cc1 -> this has the middle
nitrogen as tertiary (no H) and we had 17H.
But we need 18H. So we must have the middle nitrogen with one H.
Therefore, the correct ...
...
The nitro group is represented with charge separation, but overall
the molecule is neutral.  This should be acceptable.  Let me
double-check the rotatable bonds in this new design:  Bonds:  1.
First ring to the first nitrogen (N in NC(=O)) -> single bond,
counted.  2.  The amide bond:  N-C(=O) -> not counted.  3.  The
bond from the carbonyl to the CH2:  C(=O)-C (single bond) ->
counted.  4.  The bond from the CH2 to the middle nitrogen:  C-[NH]
-> single bond, counted.  5.  The bond from the middle nitrogen to
the second ring:  [NH]-c2 -> single bond, counted.  6.  The bond
from the second ring to the CH2 in the ethylamine:  c2-CCN ->
counted.  7.  The bond in the ethylamine:  between the two CH2:
C-C -> counted.  8.  The bond from the last CH2 to the NH2:
terminal, not counted.
We have bonds 1, 3, 4, 5, 6, 7 -> 6.
It matches.
So we are done.  </think> { "smiles":
"O=[N+]([O-])c1ccc(NC(=O)C[NH]c2ccc(CCN)cc2)cc1" }
```

**Generation Results**                                                    **Reward: 0.0**

**Generated SMILES:** { "smiles":
"O=[N+]([O-])c1ccc(NC(=O)C[NH]c2ccc(CCN)cc2)cc1" }

**Validation:** Valid SMILES: ✓

**Constraint Satisfaction:**

| Property | Expected | Actual | |
|---|---|---|---|
| Rotatable Bond Count | = 6 | 7 | × |
| Molecular Formula | = C16H18N4O3 | C16H18N4O3 | ✓ |

| **Chemistry Scores** | **2.1** | | **Reasoning Scores** | **2.2** |
|---|---|---|---|---|
| Molecular Parsing | **2.0** | | Task Fidelity | **1.0** |
| Ring Topology | **2.0** | | Constraint Tracking | **2.0** |
| Connectivity Counting | **2.0** | | Evidence Grounding | **2.0** |
| Aromaticity Recognition | **2.0** | | Method Selection | **2.0** |
| Bonding States | **2.0** | | Reasoning Flow | **2.0** |
| Charge Valence | **2.0** | | Error Awareness | **3.0** |
| Functional Identification | **2.0** | | Quantitative Precision | **3.0** |
| Property Attribution | **3.0** | | Reference Discipline | **3.0** |
| Stereochemistry | N/A | | Output Quality | **2.0** |

**Analysis Summary**

**Chemistry:** The trace shows solid, chemically informed reasoning: the model demonstrates good practical SMILES handling (including explicit nitro representation), correct recognition of aromatic rings and how ring bonds influence rotatability, and clear understanding of amide/carbonyl bonding and why amide bonds are typically excluded as rotatable. It also breaks the molecule into fragments and performs explicit atom and hydrogen bookkeeping with self-correction. Weaknesses: despite careful reasoning the model miscounted rotatable bonds (verification shows an off-by-one error), indicating a mistake in the final enumeration of which single bonds meet the rotatable-bond criteria. Overall competent chemical reasoning with an isolated, but consequential, counting/assignment error in property attribution.

**Reasoning:** The trace shows a well-focused, systematic approach: the model carefully restated the task, computed degree of unsaturation, and explicitly tried to design a structure meeting both formula and rotatable-bond constraints. It used molecule-specific reasoning (atom counts, enumerated rotatable bonds) and iterated to fix an initially detected hydrogen-count mismatch. Method selection was principled (building a rigid aromatic core with flexible linkers) and the reasoning flow was coherent and convergent. Weaknesses: the model mis-evaluated the rotatable-bond count (final SMILES failed the rotatable-bond constraint per verification), indicating an off-by-one or conceptual slip in which single bonds are rotatable (some N–aryl or other bonds were counted/miscounted). There was some ambiguity in SMILES-level hydrogen/valence handling (explicit [NH] vs implicit H) and a missed final verification step against the precise rotatable-bond definition. Overall: competent, systematic reasoning with notable execution error on the key constraint (rotatable-bond count).

## GLM-4.6 - constraint generation (stereocenters and saturated rings)

**Question**     Task Type: Generation — Load: 1 — Complexity: 0-250

Build a molecule containing exactly 5 S-stereocenters and exactly 5 saturated rings. Respond with a JSON mapping whose key is 'smiles'.

**Model Response**

```
<think>I need to create a molecule with exactly 5 S-stereocenters
and exactly 5 saturated rings.
Let me think about this step by step:
1.  Saturated rings are rings that contain only single bonds (no
double bonds in the ring) 2.  S-stereocenters are chiral centers
with S configuration
I need to design a molecule that has both of these features.  Let
me think about a structure that could work:
One approach could be to use a polycyclic structure with 5 rings,
where I can introduce chiral centers at bridgehead positions or
other suitable positions.
Let me try to build something with 5 saturated rings first:  – A
cyclopentane ring (5-membered saturated ring) – I could then fuse
or bridge additional rings to it
Actually, let me think of a more systematic approach.  I could use
a steroid–like backbone or a polycyclic structure.
For 5 saturated rings, I could use something like:  A bicyclic
system fused to another bicyclic system, plus one more ring
Let me try to construct a molecule with 5 fused saturated rings and
introduce 5 S-stereocenters.
One possibility could be a pentacyclic structure like:

    C1---C2
   /       \
  C5        C3
  |         |
  C4        C6
   \       /
    C7---C8
   /       \
  C10       C9
  |         |
  C11       C12
   \       /
    C13---C14
   /       \
  C15       C16
  |         |
  C17       C18
   \       /
    C19---C20

But this is getting complex.  Let me think of a simpler approach.
Actually, let me use a known polycyclic structure and modify it.  I
could use a decalin–like structure extended to have 5 rings.
Let me try:  A structure with 5 fused cyclohexane rings (like a
pentacyclo[...])  structure.  I'll focus on creating a complex
polycyclic molecule with specific stereochemical requirements.  The
key challenge is introducing five S–stereocenters while maintaining
five saturated rings.  I'll need to carefully design the molecular
architecture to ensure precise stereochemical configuration.
```

**Model Response**

```
The molecule will involve a sophisticated ring system where each
chiral center is strategically positioned. I'll use wedge and dash
notations to control the absolute configuration, ensuring all
stereocenters are S-configured. The structure will require
meticulous attention to bond angles and spatial arrangement to
achieve the desired stereochemical outcome.
My approach involves constructing a polycyclic framework with
carefully placed substituents that define the S-configuration at
each stereogenic center. The complexity lies in simultaneously
managing ring saturation and stereochemical precision.</think>
<answer>{"smiles":
"C[C@@H]1[C@H]2CC[C@H]3CC[C@@H]4CC[C@H]5CC[C@@H]5C4C3C2C1"}
</answer>
```

**Generation Results**                                              **Reward: 0.0**

**Generated SMILES:** {"smiles":
"C[C@@H]1[C@H]2CC[C@H]3CC[C@@H]4CC[C@H]5CC[C@@H]5C4C3C2C1"}

**Validation:** Valid SMILES: ✓

**Constraint Satisfaction:**

| Property | Expected | Actual | |
|---|---|---|---|
| Stereocenter S Count | = 5 | 3 | × |
| Saturated Rings Count | = 5 | 5 | ✓ |

| **Chemistry Scores** | **3.2** | | **Reasoning Scores** | **3.1** |
|---|---|---|---|---|
| Molecular Parsing | **3.0** | | Task Fidelity | **2.0** |
| Ring Topology | **3.0** | | Constraint Tracking | **3.0** |
| Connectivity Counting | **4.0** | | Evidence Grounding | **4.0** |
| Aromaticity Recognition | N/A | | Method Selection | **3.0** |
| Bonding States | N/A | | Reasoning Flow | **3.0** |
| Charge Valence | N/A | | Error Awareness | **4.0** |
| Functional Identification | N/A | | Quantitative Precision | **4.0** |
| Property Attribution | N/A | | Reference Discipline | **3.0** |
| Stereochemistry | **3.0** | | Output Quality | **2.0** |

**Analysis Summary**

**Chemistry:** The trace shows partial competence: the model formulated the design goal, recognized 'saturated rings' and 'S-stereocenters', and produced an explicit stereochemical SMILES using @/@@ notation (evidence of some SMILES-writing ability). However, its structural reasoning is shallow and informal — ring topology and fusion relationships were discussed only qualitatively, and the ASCII sketch was not a rigorous graph construction. The model did not perform systematic connectivity or stereocenter enumeration and failed to verify the resulting stereocenter count (verification indicates only 3 S centers instead of 5). No discussion of aromaticity, bonding/hybridization, formal charges, or functional-group identification appears. Overall: basic practical ability to emit stereochemical SMILES, but weak in systematic parsing, precise ring-topology reasoning, and counting/verification.

**Reasoning:** The trace shows a clear understanding of the prompt and a plausible high-level strategy (build a pentacyclic saturated framework and place S stereocenters). However the reasoning is largely heuristic and non-systematic: the model did not perform instance-specific enumeration (no atom-level counting or ring counting), offered no internal verification, and produced a SMILES that failed the stated constraints. Execution lacked concrete checks or error-correction, leading to an incorrect final result despite an otherwise relevant approach. Output formatting was correct but the constraint management and evidence grounding were weak.

## Qwen-3 235b - constraint generation (BRICS fragments and aromatic groups)

**Question**      Task Type: Generation — Load: 2 — Complexity: 0-250

Provide one SMILES for a molecule achieving exactly 9 BRICS fragments and exactly 31 aromatic groups. Provide your answer as JSON with 'smiles' as the key.

**Model Response**

```
Okay, let's see.  I need to provide a SMILES string that results in
exactly 9 BRICS fragments and exactly 31 aromatic groups.  Hmm,
BRICS fragments are the ones generated by the Broken Ring in
Chemical Structures method, right?  So BRICS decomposition breaks
molecules at certain bonds, like those between rings or specific
functional groups.
First, I should recall what BRICS does.  It identifies bonds that
can be broken to generate fragments, typically focusing on bonds
that are part of ring systems or between rings and non-ring
structures.  Each BRICS fragment is a part of the molecule after
breaking those bonds.  So to get 9 fragments, the molecule must
have 8 such cleavable bonds ...
...
But the main point is, the molecule needs to be highly branched
with many rings connected ...
...
Now, the other requirement is exactly 31 aromatic groups.  Aromatic
groups here probably refer to individual aromatic rings or systems.
Wait, but how is that counted?  For example, benzene has one
aromatic ring, naphthalene has two fused rings but is considered
one aromatic system.  Wait, but the question says "aromatic
groups"|maybe they mean the number of aromatic rings, even if fused
...
...
Alternatively, maybe "aromatic groups" refers to the number of
aromatic atoms.  But 31 aromatic atoms would be, for example, 31
carbon atoms in aromatic rings.  But the question says "groups",
which is more likely rings.
Wait, but the problem states "exactly 31 aromatic groups".
...
Wait, BRICS fragments:  when you have multiple phenyl rings
connected by single bonds, each connection between two phenyls
(like in biphenyl) is a cleavable bond.  So for example, a molecule
like C1=CC=CC=C1-C2=CC=CC=C2 would have two aromatic rings
connected by a single bond.  BRICS would break that bond, resulting
in two fragments.  So if we have a linear chain of 31 phenyl rings
connected by single bonds, each connection is a cleavable bond.  So
the number of BRICS fragments would be 31 (since 30 bonds broken,
31 fragments).  But the problem requires 9 BRICS fragments.  So
that's not matching.
...
Alternatively, 8 fragments:  7 with 4 rings (28) and 1 with 3 (3),
total 31.
So the SMILES would have a core connected to 8 aromatic systems (7
with 4 rings, 1 with 3).
The SMILES might look like:
C(c1ccc2cc3ccc4ccc5ccc2c3c45)(c6ccc7cc8ccc9ccc%10ccc7c8c9%10)(c11
ccc12cc13ccc14ccc15ccc12c13c1415)(c16ccc17cc18ccc19ccc20ccc17c18c19
20)(c21ccc22cc23ccc24ccc25ccc22c23c2425)(c26ccc27cc28ccc29ccc30ccc27
```

```
c28c2930)(c31ccc32cc33ccc34ccc35ccc32c33c3435)c36ccc37
cc38ccc39ccc40ccc37c38c3940
But this is 8 substituents on a central carbon, which is impossible
(carbon can only have 4 bonds).
Ah, right.  So the core must have more atoms.
Perhaps a central chain of carbons, each with two substituents.
For example, a chain like
C(C(c1...)(c2...))C(C(c3...)(c4...))C(C(c5...)(c6...))C(C(c7...)
(c8...))
This would have 8 substituents on a 4-carbon chain.  ...
This is getting too complicated.  Given the time I've spent, I'll
posit that the correct SMILES is a highly branched molecule with 8
cleavable bonds and 31 aromatic rings, and the answer is:  ...
...
Even though it's likely invalid due to branching limits, it's the
best I can do.  </think>
<answer>{"smiles":
"C(C1(C2=CC=CC=C2)C3(C4=CC=CC=C4)C5(C6=CC=CC=C6)C7(C8=CC=CC=C8)C9
(C10=CC=CC=C10)C11(C12=CC=CC=C12)C13(C14=CC=CC=C14)C15(C16=CC=CC=
C16))(C17(C18=CC=CC=C18)C19(C20=CC=CC=C20)C21(C22=CC=CC=C22)C23(C24
=CC=CC=C24)C25(C26=CC=CC=C26)C27(C28=CC=CC=C28)C29(C30=CC=CC=C30)C31
(C32=CC=CC=C32))"}</answer>
```

**Generation Results**                                    **Reward: 0.0**

**Generated SMILES:** {"smiles":
"C(C1(C2=CC=CC=C2)C3(C4=CC=CC=C4)C5(C6=CC=CC=C6)C7(C8=CC=CC=C8)C9
(C10=CC=CC=C10)C11(C12=CC=CC=C12)C13(C14=CC=CC=C14)C15(C16=CC=CC
=C16))(C17(C18=CC=CC=C18)C19(C20=CC=CC=C20)C21(C22=CC=CC=C22)C23
(C24=CC=CC=C24)C25(C26=CC=CC=C26)C27(C28=CC=CC=C28)C29(C30=CC=CC
=C30)C31(C32=CC=CC=C32))"}

**Validation:** Valid SMILES: ✗

| Chemistry Scores | | 3.6 |
|---|---|---|
| Molecular Parsing | 5.0 | |
| Ring Topology | 3.0 | |
| Connectivity Counting | 4.0 | |
| Aromaticity Recognition | 3.0 | |
| Bonding States | 3.0 | |
| Charge Valence | 5.0 | |
| Functional Identification | 2.0 | |
| Property Attribution | 4.0 | |
| Stereochemistry | N/A | |

| Reasoning Scores | | 3.6 |
|---|---|---|
| Task Fidelity | 3.0 | |
| Constraint Tracking | 3.0 | |
| Evidence Grounding | 4.0 | |
| Method Selection | 4.0 | |
| Reasoning Flow | 3.0 | |
| Error Awareness | 4.0 | |
| Quantitative Precision | 3.0 | |
| Reference Discipline | 4.0 | |
| Output Quality | 4.0 | |

**Analysis Summary**

**Chemistry:** The trace shows partial conceptual knowledge about rings and aromatic counts (correctly distinguishes benzene/naphthalene and recognizes fused rings contribute ring counts), but exhibits serious methodological and chemical reasoning failures. The model repeatedly confused BRICS rules and fragment counting, made inconsistent connectivity arithmetic, and produced chemically impossible structural suggestions (e.g., proposing central atoms with far too many substituents and an invalid SMILES). Aromaticity reasoning is shallow (limited to simple examples), and bonding/valence handling is poor (fundamental violations). Functional-group recognition is basic (recognizes common aromatics), while property attributions (BRICS applicability, cleavage assumptions) are error-prone. No stereochemistry was discussed.

**Reasoning:** The trace shows an overall on-task effort: the model restates the requirements and keeps both constraints (9 BRICS fragments and 31 aromatic groups) in view, and it performs some correct arithmetic (e.g. fragment count relations and fragment-sum calculations). However, the reasoning is largely heuristic and exploratory rather than algorithmic, with substantial uncertainty about definitions (what counts as an "aromatic group") and BRICS rules. There is minimal molecule-specific verification (no atom-level enumeration or simulation of BRICS on the proposed structure), and the model fails to detect or correct major structural/valency errors when assembling a candidate SMILES. References and indexing drift in the attempted construction, and the final JSON contains an invalid/unreasonable SMILES (as flagged in verification_info). Strengths: maintained both goals, some correct counting, iterative brainstorming. Weaknesses: weak grounding in instance-level checks, brittle method selection, missed obvious structural invalidity, and an invalid final output.

## D    USE OF LARGE LANGUAGE MODELS

In preparing this manuscript, we used LLMs to polish existing text, specifically for improving grammar, readability, and stylistic consistency. LLM-based search tools were also used in a supportive capacity for literature search. All scientific ideas, analyses, figures, and results were conceived, implemented, and validated entirely by the authors. Code development was likewise carried out by the authors. Code-assistance tools (e.g., GitHub Copilot, Claude Code) were used in a supportive role, primarily to debug or refine existing implementations. In cases where such tools produced new code, this was limited to narrowly defined tasks based on the authors' explicit instructions, and final authorship of all code rests with the authors. No LLMs were used for research ideation or exploration.

