# OpenReview forum: "MolecularIQ: Characterizing Chemical Reasoning Capabilities Through Symbolic Verification on Molecular Graphs"
_ICLR.cc/2026/Conference — ICLR 2026 Poster_

### Official Review · Reviewer_9zuz · 2025-10-31

**Soundness:** 3
**Presentation:** 3
**Contribution:** 3
**Rating:** 4
**Confidence:** 3

**Summary:**

The paper introduces MOLECULARIQ, a benchmark aimed at chemical structure reasoning with symbolically verifiable ground truths. Tasks are built directly on molecular graphs (via RDKit solvers) and span three orthogonal axes: reasoning category (feature counting, index attribution, constrained generation), multitask load (1, 2, 3, or 5 simultaneous requirements), and molecular complexity (Bertz bins). The authors also provide a dynamic variant (MOLECULARIQD) for regenerating fresh test sets and preventing saturation/overfitting. Evaluation is integrated into the lm-evaluation-harness, with hierarchical answer extraction and semantic (key-agnostic) comparison; accuracy is averaged over three independent rollouts. A large-scale study (34 models) finds that recent MoE/generalist reasoning LLMs dominate; chemistry-tuned instruction/RL models often underperform their bases; constrained generation is easiest at low constraint counts but degrades sharply as constraints stack; and multitask load harms performance more than molecular complexity. The paper argues symbolically verifiable tasks reduce leakage/bias and produce capability fingerprints localizing failure modes.

**Strengths:**

Tight problem statement & clear contribution. A fully symbolically verifiable chemistry benchmark focused on structure-grounded reasoning (not factual recall) is timely and well-motivated.

Three-axis profiling. Disentangling reasoning type, multitask load, and molecular complexity provides diagnostic granularity and actionable error localization.

Index-based tasks. Pairing counting with index attribution helps distinguish genuine graph reasoning from pattern-matching/shortcut counts.

Solid evaluation infrastructure. Harness integration, hierarchical extraction, and semantic answer comparison address brittle formatting issues; multi-rollout accuracy mitigates sampling variance.

Dynamic benchmark (MOLECULARIQD). A path to refreshable evaluations that can evolve with the field and support RL with verifiable rewards.

Empirical insights. Consistent trends (MoE leads; chemistry tuning can hurt; canonical/aromatic SMILES easier; multitask load dominates difficulty) are useful for both modeling and benchmark design.

Transparent limitations section. Clear articulation of the current scope (2D graphs, single-molecule tasks, symbolic-only feature set).

**Weaknesses:**

2D-only scope. Restricting to graph connectivity omits 3D stereoelectronic/conformational effects that matter for realistic chemical reasoning; several stereochemistry tasks may still be fragile under a purely 2D treatment.

Verifier dependence & edge cases. Heavy reliance on RDKit rules brings corner-case risk (e.g., aromaticity/kekulization, tautomers, undefined stereocenters). Clear auditing and unit tests for borderline cases would strengthen claims.

Dataset scale & coverage. The main static benchmark uses hundreds of molecules / thousands of questions—adequate for signal but small relative to chemical space. It is unclear how representative the selected feature distributions and Bertz bins are for downstream applications.

Generation tasks may be gameable. Low-constraint prompts (e.g., “has two rings”) can be satisfied by template snippets; evidence that models aren’t exploiting canonical pattern banks (beyond SMILES randomization) would be welcome.

Per-model configuration fairness. “Tailored configs” improve each model’s score but may complicate cross-model fairness. A fixed canonical configuration alongside tailored ones would help disambiguate.

Rollout averaging & significance. Averaging over three stochastic runs may be thin for close comparisons; confidence intervals and multiple seeds per model/config would improve robustness.

Leaderboard/process details deferred. Several artifacts (e.g., public leaderboard link, full configs) are promised camera-ready; reproducibility would benefit from making the dynamic generator and verifier immediately available.

Limited assessment of prompt/extraction shaping. Although hierarchical extraction is a strength, further stress tests against format-shaping and overfitting to extraction heuristics would increase trust.

**Questions:**

Verifier audits: How do you handle aromaticity/kekulization mismatches, tautomerism, and unspecified stereochemistry during indexing and generation checks? Any published test suite of adversarial edge cases?

SMILES perturbations: Beyond canonical/aromatic randomization, did you try token perturbations (ring index relabeling, randomized branches) to further separate pattern recall from graph reasoning?

Fairness controls: Can you report both fixed (uniform decoding and temperature) and tailored configs for all models to separate capability from tuning sensitivity?

Rollouts & variance: Why three rollouts? Do results meaningfully change with 5–10 rollouts or multiple seeds? Please add per-model variance bars in the main text.

Constraint hardness: For constrained generation, can you provide a calibrated hardness ladder (e.g., constraint sets with matched feasibility rates) and show how models scale as we move up the ladder?

Dynamic set governance: How will MOLECULARIQD updates be versioned (to avoid moving goalposts) and how will you prevent train–test leakage as the community begins to tune on the benchmark?

3D extension: What’s the roadmap for 3D-aware, symbolically verifiable tasks (e.g., CIP resolution, ring puckers, distance constraints) and for multi-molecule tasks (reaction stoichiometry, scaffold ranking) while maintaining verifiability?

Failure mode taxonomy: Can you release capability fingerprint templates and diagnostic exemplars so users can map a model’s errors to specific graph-perception or compositional failures?

---

> ### Author Response · Authors · 2025-11-14
> **Request for Clarification**
>
> Dear reviewer, thank you for your feedback and the many detailed suggestions. Before addressing your specific comments, we would like to clarify one point:
>
> (Q2) We are unfamiliar with the term “branch perturbation.”
>
> Do you refer to keeping the molecular backbone fixed while modifying side-chain positions in the SMILES? If so, wouldn’t this already be implicitly evaluated through SMILES randomization?

---

> > ### Comment · Reviewer_9zuz · 2025-11-22
> >
> > No. SMILES randomization is different string representations of the same molecule. Branch perturbations change the molecule by perturbing its structure. Please look this up how it is typically done. I expect you to be familiar with such basic concepts.

---

> ### Author Response · Authors · 2025-11-28
> **Respose to Reviewer 9zuz (Part 1)**
>
> We thank you for your constructive and detailed feedback. Below, we address each of the weaknesses and questions you raised:
>
> - **(W1 - 2D-only scope)** and **(Q7 - 3D extension)** You are correct that our benchmark currently uses SMILES representations. This is a deliberate choice, as a SMILES-based benchmark best matches the present capabilities of chemistry-capable LLMs. We agree that 3D information is increasingly important, and we have added an outlook on MolecularIQ3D, a future extension with 3D-aware, symbolically verifiable tasks (Sec. 5), informed by your concrete task suggestions. For more discussion, see second point of the general answer.
> - **(W2 – verifier dependence and edge cases)** and **(Q1 – choice of solver)** Our symbolic solver is built on RDKit for reproducibility and compatibility with existing cheminformatics workflows; Sec. B.2 now makes this design choice explicit. We prioritize internal self-consistency over exhaustively resolving chemically ambiguous edge cases: tautomers are treated as distinct molecules, aromatic and Kekulé SMILES yield identical outputs, and undefined stereochemistry is explicitly tracked and surfaced through dedicated tasks rather than silently resolved.
>
>     To assess robustness, we validated the solver on 10,000 ****randomly sampled molecules, checking that (i) stereocenter counts and related quantities are internally consistent across tasks, (ii) property calculations are deterministic, and (iii) all ~50 solver methods produce valid outputs without crashes. These checks give us high confidence that solver-induced artifacts do not drive the reported model rankings.
>
> - **(W3 - dataset scale & coverage)** We agree that 849 molecules are small relative to the full chemical space. However, as detailed in Appendix Sec. B.2, we took deliberate steps to ensure broad coverage: molecules were selected using an inverse-frequency sampling scheme to match the feature distributions observed in PubChem while avoiding over-representation of common motifs. Regarding molecular complexity, defining “complexity” itself is non-trivial. Our current Bertz-based bins (Figure B1) were chosen to represent clear strata of *simple*, *intermediate*, and *complex* small organic molecules. The resulting Bertz statistics closely match the original PubChem pool: ours (780 ± 599) vs. PubChem (778 ± 402).
> - **(W4 - generation tasks may be gameable)** and **(Q2 - SMILES perturbations).** It was a design choice to include different task complexity levels, and clearly, the number of constraints in generation tasks is such a complexity dimension. Comparing model behavior across low- and heavy-constraint tasks would allow us to detect exactly such behavior, and notably our extended failure mode analysis detects such general behavior (Fig. D11). As requested, we added a SMILES-augmentation analysis (Figs. D2, D3).
> - **(W5 - per-model configuration fairness)** and **(Q3 - fairness controls)** This is the only point where we respectfully disagree. We aim to evaluate each model under the usage conditions intended by its authors rather than a single unified configuration. In practice, forcing a shared prompt and decoding setup can disadvantage some models and obscure performance:
>     1. Some models are trained for relatively long completions and perform poorly when forced into very short reasoning budgets.
>     2. Models are highly sensitive to system prompts; for example, on *ether0* omitting the recommended system prompt caused about a 10 percentage point drop in the generation subdomain.
>     3. Technical interfaces differ: GPT-OSS models, for instance, require a developer prompt channel that other APIs do not support, so a single canonical configuration is infeasible.
>
>     For transparency, we therefore document all model-specific settings and prompts, and we regard evaluating each model “as recommended” as the fairest and most realistic comparison.
>
> - **(W6 - rollout averaging & significance)** and **(Q4 - rollouts & variance)** We agree that more rollouts or seeds would, in principle, improve robustness. In practice, our design balances dataset size, task-space dimensionality, and rollout budget under substantial compute costs: evaluating the largest models already requires up to 380 GPUh on H100 hardware, and simply rerunning all experiments would cost about 2000 GPUh. To our knowledge, we are already the only chemistry benchmark to use multiple rollouts per question (ChemIQ [1] and ChemCOTBench [2] use a single rollout). We also share the reviewer’s concern about uncertainty quantification and therefore report error bars (standard errors) for all tables included.
>
>     To directly assess the impact of additional rollouts, on a subset of models, we increased the budget to 8 rollouts and obtained very similar scores and rankings to 3 rollouts (Tab. D13), supporting our chosen rollout budget as a reasonable compromise between robustness and practicality.

---

> ### Author Response · Authors · 2025-11-28
> **Respose to Reviewer 9zuz (Part 2)**
>
> - **(W7 - leaderboard/process details deferred)** To preserve anonymity, we do not include the live leaderboard link in the manuscript. In the updated supplementary material, we now provide a zipped version of the leaderboard. The full model configs and symbolic verifiers were already included, and we have now also added the dynamic generator. After the review process, we will release the public leaderboard link, a GitHub repository with all code, and a PR to lm-evaluation-harness.
>     - Because the live leaderboard depends on Hugging Face datasets that are not yet accessible to you, we provide an offline variant with the leaderboard data and code. To launch it, please: (i) navigate into the leaderboard folder, (ii) build the conda environment from `leaderboard.yaml` or install the requirements from `requirements.txt`, and (iii) run `python app.py`. We also include screenshots for easy inspection of the frontend.
>     - For MolecularIQD, we provide an example file that introduces the core features of the module. To run it, please: (i) navigate into the MolecularIQD folder and (ii) run `python example_usage.py`.
> - **(W8 - limited assessment of prompt/extraction shaping)** We appreciate that you view our hierarchical extraction pipeline as a strength. Extraction is a critical challenge in LLM evaluation, and we invested considerable effort to make it robust and transparent.
>
>     Following your suggestion, we further analyzed format dependence by quantifying type validity (whether the extracted answer matches the requested structure or type) and relating it to task accuracy. As discussed in the general response and in Sec. D.2.4, top models typically achieve high type validity (80 to 90%) while accuracy remains substantially lower, indicating that most errors stem from semantic reasoning rather than formatting.
>
> - **(Q2- SMILES perturbations, add on)**  Thank you for clarifying your point on branch perturbation. We initially misunderstood it as another way to generate alternative SMILES for the same molecule (analogous to randomization, kekulization, or ring labeling) to probe robustness to varied string representations. We are familiar with branch perturbations that modify the molecular structure itself, but did not consider such changes here, as they would not help isolate pattern recall from structural reasoning.
>
>     In addition to the existing SMILES perturbation results in Tabs. D8–D9 and Fig. D2, we have added, as you suggested, a ring re-enumeration analysis (Tab. D10, Fig. D3). The results reinforce our earlier findings that current models partially exploit canonical token and ring-numbering regularities rather than performing fully representation-invariant graph reasoning.
>
> - **(Q5 - constraint hardness)** For single-constraint generation, Fig. D4 details which constraints are easy versus difficult; for example, reaction success, carbon atom count, and longest carbon chain are easier, while bridgehead atoms, molecular formula, and BRICCS decomposition are harder. For multi-constraint generation, Fig. D1 shows accuracy as a function of constraint prevalence, with rarer constraints consistently harder to satisfy, so prevalence effectively defines a hardness ladder.
> - **(Q6 - dynamic set governance)** MolecularIQD will use a simple, transparent versioning scheme: each released dynamic set receives a unique version identifier, and leaderboard entries are tagged with the corresponding version. Older versions remain available so results stay reproducible and the “goalposts” do not shift.
>
>     We do not plan a fixed release schedule, but will introduce new dynamic sets once models begin to saturate on the current one. To limit leakage, the test pool remains non-public and new molecules and tasks are sampled from it. While it is theoretically possible to preprocess the entire PubChem database in our exact way, we regard this as technically infeasible at present and, as with most benchmarks, also rely on the community’s goodwill.
>
> - **(Q8 failure mode taxonomy)** We agree that a failure taxonomy is important for model development. We now provide one, derived from manual analysis of failing chain-of-thought traces (Secs. D.2.6–D.2.7), which highlights recurring issues such as incorrect functional group recognition, stereochemical mistakes, weak constraint tracking, and quantitative inaccuracies, illustrated with representative examples.
>
> We believe that our clarifications, the inclusion of additional experiments, and the extended analyses address the concerns raised, and we hope the reviewer will consider revising their overall evaluation in light of these improvements.
>
> [1] Runcie. Assessing the Chemical Intelligence of Large Language Models
>
> [2] Li. Beyond Chemical QA: Evaluating LLM’s Chemical  Reasoning with Modular Chemical Operations

---

### Official Review · Reviewer_31oX · 2025-11-01

**Soundness:** 3
**Presentation:** 3
**Contribution:** 2
**Rating:** 4
**Confidence:** 4

**Summary:**

This work introduces a new benchmark to evaluate LLMs on chemistry tasks. All tasks are grounded in the molecule’s graph structure and have answers that can be checked by a symbolic solver. The authors evaluates 34 LLMs and find that the largest mixture-of-experts models with high reasoning budgets achieve the best accuracy.

**Strengths:**

1. The paper presents a verifiable benchmark and all tasks have ground-truth solutions, allowing reliable automatic evaluation.

2. The benchmark varies the molecular complexity and tests different chemical reasoning skills.

**Weaknesses:**

1. The benchmark excludes tasks like quantitative property prediction or reaction prediction, so it does not evaluate LLMs’ ability on some real-world chemistry problems.

2. The tasks in the benchmark are fundamental checks that chemical softwares can do straightforwardly. They may be somewhat disconnected from how humans typically solve chemistry problems. For example, asking a model to ‘generate a molecule with two rings and five heterostoms’ is more like a puzzle or exercise than creative problem-solving.

3. The benchmark uses SMILES strings for molecules and the authors noticed that LLMs may use pattern recognition rather than structural reasoning. Similar arguments have been discussed in the literature, such as the inconsistency of LLMs in molecular representations which indicates LLMs fail to capture the underlying chemistry. The authors may consider adding similar consistency checks in the benchmark.

**Questions:**

1. Does a better performance on the benchmark always suggest a stronger reasoning? Is there a way to explicitly evaluate if the model is doing reasoning, or is it doing pattern recognition?

2. Some LLMs use external tool calls to solve chemistry tasks such as ChemCrow. Do the authors envision the benchmark being used not just standalone LLMs, but also LLM-based agents?

---

> ### Author Response · Authors · 2025-11-14
> **Request for Clarification**
>
> Dear reviewer, thank you for your constructive feedback. Before addressing your specific comments, we would like to clarify one point:
>
> (W3) We are not fully sure what you mean by “consistency checks” for assessing whether the LLM relies on pattern recognition rather than structural reasoning.
>
> To probe pattern-recognition and memorization effects, we have already compared model performance between canonical and randomized SMILES and observed a clear performance drop under randomized SMILES (Table C9). If this is the type of consistency check you had in mind, we will make these experiments and their implications more explicit in the manuscript. If not, could you please clarify what additional form of consistency check you are referring to?

---

> ### Author Response · Authors · 2025-11-28
> **Response to Reviewer 31oX**
>
> We thank you for your constructive and detailed feedback. Below, we address each of the weaknesses and questions you raised:
>
> - **(W1)** Your feedback regarding the absence of property or reaction prediction tasks was very insightful. Thank you. It made us realize that our initial introduction did not clearly articulate why we deliberately did not include such tasks. For a more detailed discussion, please refer to the first part of our general response. Based on your comment, we revised the introduction to make this rationale explicit.
> - **(W2)** We agree that our tasks are fundamental checks (with respect to the understanding of molecular graphs), and this is intentional. As discussed in the general response, they target prerequisite structural competencies that underpin realistic property and reaction prediction. While tools such as RDKit can solve our tasks trivially, our goal is not to propose new models for these tasks, but rather to diagnose which basic capabilities LLMs do or do not possess. These limitations can directly cap downstream performance. In practice, specialized QSAR and reaction models (explicitly trained on those tasks) still outperform chemistry LLMs on most end-task benchmarks [1, 2]. Thus, the added value of LLMs lies in functioning as general “all-in-one” assistants or agents, where they must internally handle such sub-tasks reliably. Evaluating LLMs on these seemingly puzzle-like tasks is therefore analogous to unit tests for core competencies, aligned with recent work showing that solving well-defined sub-tasks can improve or reveal a model’s broader reasoning abilities [3].
> - **(W3)** Thank you for this suggestion. To better separate pattern recognition from genuine structural reasoning, we carried out an additional analysis across several SMILES pre-processing schemes. Specifically, we now compare performance under canonical aromatic, canonical kekulized, randomized aromatic, and randomized kekulized SMILES (Fig. D2), and we additionally evaluate the impact of ring-closure re-enumeration (Fig. D3). Overall, the consistent performance drop under SMILES augmentations indicates that some correct predictions indeed arise from surface-level pattern recognition rather than robust graph-level reasoning. Your comment directly motivated this analysis, and we highlight these findings in both the Results section and Appendix.
> - **(Q1)** Better performance on our benchmark correlates with, but does not automatically guarantee, stronger reasoning. What MolecularIQ measures is the ability to produce correct answers in settings where shortcuts such as memorization, data leakage, and superficial pattern matching are intentionally minimized. Because all tasks are symbolically verifiable and constructed directly from molecular graphs rather than from literature-derived labels, performance cannot be attributed to recall of known facts or exposure to public datasets. Still, for some questions, models may rely on heuristics or shallow pattern recognition that happen to align with some tasks. Our benchmark provides several mechanisms to disentangle these effects:
>     - Counting vs. indexing: Counting can sometimes be answered by surface-level cues; indexing requires locating the correct substructure in the molecular graph. For a model of interest, a large drop from counting to indexing would indicate shortcut behavior (Fig. D12).
>     - Multitask load: As tasks become more compositional, superficial shortcuts should fail due to combinatorial growth. For a model of interest, steep declines with increasing multitask load would signal limitations in compositional reasoning (Fig. D11).
>     - SMILES augmentations (extended analysis inspired by your suggestion): Canonical and randomized SMILES encode the same graph but differ strongly in surface form. Larger performance drops under randomization indicate reliance on surface patterns. Importantly, our dataset contains a 50/50 split of canonicalized and randomized SMILES specifically to measure such effects.
> - **(Q2)** No. Our benchmark evaluates the structural understanding capabilities of standalone LLMs. An agent with access to RDKit, or similar cheminformatics toolkits, could solve essentially all tasks, meaning the evaluation would measure coding/tool-use abilities rather than internal chemical reasoning. For this reason, tool use is deliberately excluded from the benchmark.
>
> We hope that our responses address all raised questions and concerns, and will encourage the reviewer to consider raising their overall evaluation.
>
>
> [1] Yu. LlaSMol: Advancing Large Language Models for Chemistry with a Large-Scale, Comprehensive, High-Quality Instruction Tuning Dataset
>
> [2] Zhao. Developing ChemDFM as a large language foundation model for chemistry
>
> [3] Alampara. Task Alignment Outweighs Framework Choice in Scientific LLM Agents

---

### Official Review · Reviewer_yX2q · 2025-11-01

**Soundness:** 3
**Presentation:** 2
**Contribution:** 1
**Rating:** 4
**Confidence:** 4

**Summary:**

This paper introduces MoleculeIQ, a molecular structure reasoning benchmark composed of symbolically verifiable tasks. Specifically, the tasks consist of three types: (1) feature counting, (2) index-based attribution, and (3) constrained generation, and the features of interest include functional groups, chemical properties, synthesis, and so on. The MoleculeIQ dataset is composed of 849 molecules. In addition, the paper proposes a framework that dynamically computes the ground-truth labels for the designed tasks, which it calls MoleculeQID. The experimental results show that current LLMs exhibit a significant gap in the compositional and structural reasoning abilities required.

**Strengths:**

* This paper provides a wide-ranging evaluation across diverse model types and sizes.
* The introduced MoleculeQID can be further utilized for new and open molecules, guaranteeing its scalability.

**Weaknesses:**

* This study is grounded in the belief that there is a positive correlation between molecular structural understanding and molecular reasoning ability for complex property prediction. However, this belief is not explicitly demonstrated, so the necessity of building a structure-reasoning benchmark appears limited. I suggest presenting the relationship between structural understanding and predictive performance on molecular properties.
* Overthinking is a well-known pitfall in molecular structure understanding, yet the results here differ from conventional wisdom. Providing an in-depth analysis of this aspect would strengthen the contribution of the proposed benchmark and offer clearer grounds for the necessity of reinforcement learning (RL) training.
* The results clearly show where models fail, but the paper would be stronger with deeper qualitative error analysis. Presenting examples of incorrect molecules generated by top models and categorizing the types of structural mistakes would give model developers more actionable insights.

**Questions:**

* Could the authors provide the failure cases on the multi-task scenario?
* What kinds of substructures do LLMs fail to capture?

---

> ### Author Response · Authors · 2025-11-14
> **Request for Clarification**
>
> Dear reviewer, thank you for your constructive feedback. Before addressing your specific comments, we would like to clarify one point:
>
> (W1) We are unsure what you mean by *“Overthinking is a well-known pitfall in molecular structure understanding …”*.
>
> Do you refer to the phenomenon where LLMs get stuck in iterative reasoning loops during rollouts? Or over-interpretation/over-constraint of molecular structure when predicting properties?
>
> Clarifying this, or a pointer to a reference, especially in the latter case, would help us respond more accurately.

---

> ### Author Response · Authors · 2025-11-28
> **Response to Reviewer yX2q**
>
> We thank you for your constructive and detailed feedback. Below, we respond to each of the weaknesses you identified and the questions you raised:
>
> - **(W1)** We agree that clarifying the connection between structural reasoning and molecular property prediction is essential. As outlined in our general response (section: Structural reasoning is a prerequisite for molecular prediction), we revised the introduction to more clearly articulate why structural understanding is foundational for meaningful downstream tasks. Your feedback helped us recognize that this link needed to be made more explicit, and we hope the revised framing makes the relevance of evaluating structural reasoning for sophisticated chemical tasks, such as property prediction, much clearer.
> - **(W2)** We agree that LLMs getting stuck in long, unproductive reasoning chains is a well-known issue. As you correctly noted, our results show that increasing the reasoning budget tends to improve performance overall (Fig. D4). Motivated by your comment, we carried out a deeper analysis to investigate whether “overthinking” occurs in our benchmark.  Fig. C5  reveals clear evidence of overthinking: tasks in the lowest accuracy bins are associated with substantially longer responses (especially for GPT-OSS and GLM families), while high-accuracy predictions are produced with much shorter reasoning traces. Additionally, our in-depth trace analysis (Sec. D.2.6, Figures D16–C17) shows that reasoning flow is far from optimal in top-performing models, failing traces exhibit weak constraint tracking, poor quantitative precision, and limited error awareness.
> - **(W3)** Thank you for the suggestion. As described in point 3 of our general response (section: Need for deeper failure-mode analysis), we have expanded the manuscript with a more thorough investigation of failure patterns, which we believe meaningfully strengthens the work.
> - **(Q1)** Sec. D.2.7 includes examples showing how models fail under compositional load even when single-task capabilities are present. Additionally, Fig. D17 scores traces on "constraint tracking", clear degradation under higher multitask load indicates models struggle to maintain multiple simultaneous requirements, even when each sub-operation is individually within reach. Fig. D11 quantifies this: success→failure transitions consistently dominate over failure→success as load increases.
> - **(Q2)** To address your question, we performed a detailed analysis across functional-group families (Sec. D.2.5). Among all categories, alcohols, phenols, and enols appear to be the most challenging functional-group family for current models.
>
> We hope that our responses and the inclusion of the suggested failure mode analysis fully address all raised questions and will encourage the reviewer to consider raising their overall evaluation.

---

### Author Response · Authors · 2025-11-14

Dear reviewers,

We thank you for your thoughtful and constructive reviews. The quality and depth of your comments indicate the substantial effort you have invested in evaluating our submission, and your feedback will help us strengthen the manuscript.

First, we thank the reviewers for recognizing the strengths of our work, in particular the clear and focused problem statement, the fully symbolically verifiable benchmark with ground-truth solutions, the three-axis task design that probes different chemical reasoning skills, and the broad evaluation across diverse model families with a scalable setup for new molecules.

After carefully analyzing your feedback, we identified three main concerns that appear to result from insufficient clarity in our presentation rather than fundamental limitations of our work. We are confident we can address these during the rebuttal phase:

- **Relevance of the chosen tasks.**

    The reviewers raised concerns regarding (a) the absence of real-world tasks and (b) the unclear relevance of structural reasoning. Upon rereading our manuscript, we realized that our motivation for focusing solely on symbolically verifiable tasks is not articulated clearly enough. We thank the reviewers for highlighting this important shortcoming. We are revising Sections 1 (Introduction) and 3 (MolecularIQ) to clarify this motivation.

- **Limitations of the chosen data modality.**

    The reviewers correctly pointed out that our evaluation relies solely on 2D molecular (SMILES) representations. We acknowledge this as an important limitation. In our revision, we will ensure that both (a) the relevance of SMILES-based evaluations for the current research landscape and (b) their limitations are discussed adequately. We will also adopt your suggestions to outline a roadmap towards incorporating more sophisticated molecular representations.

- **Desire for deeper failure-mode analysis.**

    The reviewers requested a more detailed examination of model failure modes to better inform future model development. We agree that this would strengthen the paper, and we are preparing an extended failure-mode analysis that we will include as part of the rebuttal.

We will, of course, also address all individual comments. However, a few points remain unclear to us, and we kindly ask for clarification to ensure that our rebuttal is as accurate and helpful as possible (see below in individual answers).

---

> ### Author Response · Authors · 2025-11-28
> **General Response to Reviewers - Full Rebuttal (Part 1)**
>
> We have now uploaded the full rebuttal, including detailed responses to all questions and concerns raised by the reviewers. We thank the reviewers for their patience, as some of the additional analysis took some time.
>
> To facilitate verification of all modifications, we provide (i) a revised manuscript with all edits highlighted in the supplementary material and (ii) an updated clean version as the main submission.
>
> Below, we address the three main concerns identified previously:
>
> ## **Relevance of Tasks**
>
> - **Why no property/reaction benchmarks**
>
>     Strong benchmarks for property prediction and reaction modeling already exist (MoleculeNet, USPTO-derived sets). Our goal is to *complement*, not replicate, these with a structural-reasoning evaluation designed for verifiability and leakage resistance.
>
>     Property/reaction datasets are very likely present in LLM pretraining corpora, making it difficult to disentangle reasoning from memorization of molecule–label pairs. In contrast, while our molecules are drawn from PubChem, our task formulations are procedurally generated: the specific questions, index requests, and constraint combinations are constructed programmatically and, to our knowledge, do not appear verbatim in existing datasets or chemistry resources. This design substantially reduces the risk of task-level leakage, allowing MolecularIQ to probe structural reasoning rather than cached associations.
>
> - **Structural reasoning is a prerequisite for property prediction**
>
>     This connection is foundational in cheminformatics. The SAR principle and QSAR methods, from classical fingerprints to graph neural networks, rest on the premise that molecular structure determines properties. A model that cannot reliably identify functional groups, ring systems, or atomic connectivity cannot be expected to reason about how these features influence reactivity or bioactivity.
>
>     Testing structural competence is analogous to verifying arithmetic before trusting calculus: failure on the simpler task undermines claims of competence on harder ones. MolecularIQ targets exactly this prerequisite layer.
>
> - **Multitask compositions approximate real-world complexity**
>
>     Realistic chemistry tasks, reaction prediction, retrosynthesis, constrained molecule design, decompose into elementary structural operations: locating functional groups, counting heteroatoms, assigning stereocenters. Our multitask axis (1→2→3→5 simultaneous sub-tasks) tests whether models can coordinate multiple such operations, mirroring the compositional demands of real-world reasoning.
>
> - **Verifiable tasks enable safe RL**
>
>     Symbolic verifiability provides deterministic, exact reward signals, essential for reinforcement learning without reward hacking. Progress in math and code reasoning has been driven by such verifiable substrates. MolecularIQ offers chemistry an analogous platform: models can be optimized against exact structural objectives before deployment on noisier property/reaction tasks.
>
>
> In summary, MolecularIQ does not aim to replace property/reaction benchmarks. It provides a verifiable, leakage-resistant foundation for evaluating the structural competence that is *necessary* for success on those more complex tasks.
>
> We recognize that our original introduction did not convey this motivation clearly enough. We have therefore substantially revised the Introduction to articulate the rationale and relevance of our benchmark more explicitly, emphasizing why isolating structural reasoning is both essential and complementary to existing real-world evaluations.

---

> ### Author Response · Authors · 2025-11-28
> **General Response to Reviewers - Full Rebuttal (Part 2)**
>
> ## **SMILES as Input Modality**
>
> We acknowledge this limitation and have revised the manuscript accordingly:
>
> - Motivation (Sec. 3.3): Nearly all current chemistry LLMs operate on SMILES. Decades of QSAR research demonstrate that 2D connectivity often suffices for effective property models, making SMILES a practical and widely-used interface for evaluating structural reasoning.
> - Limitations and Roadmap (Sec. 6): We now explicitly acknowledge that our benchmark omits stereoelectronic and conformational effects. We outline concrete extensions: near-term (CIP verification from 3D coordinates, distance constraints), medium-term (conformer-aware tasks), and long-term (multi-molecule tasks like reaction stoichiometry and scaffold ranking)—all while preserving symbolic verifiability.
>
> ## **Deeper Failure-Mode Analysis**
>
> All reviewers requested more insight into how and where models fail. We have substantially expanded the analysis:
>
> - **Capability fingerprints and compositional stress (Fig. 2)**
>
>     Figure 2 now provides compact capability profiles for top models: success rates by feature group and functional-group family, gradients across Bertz complexity, and multitask load. Two compositional stress tests reveal: (i) success→failure transitions when moving from single- to multi-task prompts with matched sub-properties, and (ii) counting→indexing transitions probing whether correct counts are grounded in explicit substructure localization.
>
> - **Universal failures (Sec. D.2.1, Fig. D9)**
>
>     Over 1,000 questions that no model solves correctly, decomposed by task type, multitask load, complexity, and feature group—highlighting regimes (high multitask load, complex molecules, synthesis features) where structural reasoning remains systematically out of reach.
>
> - **Reasoning vs. formatting errors (Sec. D.2.4, Fig. D13)**
>
>     Type validity (format correctness) reaches 80–90% for top models while accuracy is substantially lower, confirming most errors are semantic reasoning failures, not extraction artifacts.
>
> - **Substructure-level analysis (Sec. D.2.5, Fig. D14–C15)**
>
>     Ten functional-group families with computed success ratios reveal that hydroxyl and sulfur/oxidized-nitrogen motifs are substantially harder than aromatic or alkyl patterns.
>
> - **Chain-of-thought taxonomy (Secs. D.2.6–D.2.7)**
>
>     Manual analysis of 300 failing traces scored on chemistry and reasoning dimensions (Figs. C16–C17), with representative examples showing breakdowns in functional-group recognition, stereochemistry, constraint tracking, and quantitative precision.
> These additions directly address pattern-matching vs. reasoning by: (i) separating counting from indexing performance, (ii) analyzing SMILES perturbation sensitivity (Tabs. D8–D10), and (iii) disentangling format validity from semantic correctness.

---

### Author Response · Authors · 2025-11-28
**TLDR for AC**

Dear new AC,

We anticipate you will have a lot to read and work through in the coming days, so we would like to provide a concise version of our rebuttal here.

Across the reviews, we identified two primary concerns:

- Relevance and scope of the benchmark

    We remain fully convinced of the benchmark’s relevance and utility for the community (including 2D representations). We interpret the reviewers’ feedback as an indication that our original introduction did not communicate our rationale clearly enough. In response, we have substantially rewritten the Introduction to better connect our work to the areas highlighted by the reviewers and to articulate more clearly why symbolically verifiable structural reasoning tasks are important.

- Desire for deeper failure-mode analysis

    With the exception of one suggestion (branch perturbations, which we concluded would not yield additional insight), we followed all reviewer recommendations and added the requested analyses. The reviewers provided strong ideas here, and we believe the manuscript is now significantly improved as a result.

Below, we include our original detailed responses (general and per-reviewer) in case you would like to see the complete arguments. A marked-up version of the revised manuscript is provided in the supplementary material.

Thank you for your time and consideration.

---

### Meta-Review · Area_Chair_Fenb · 2026-01-05

**Summary:**

This paper proposes MolecularIQ a benchmark for measuring the reasoning capabilitites of LLMs operating over molecules represented in the SMILES language. The tasks in the benchmark cover counting certain chemical features (e.g., presence of aromatic rings or classify sub-molecule structures), index-based feature attributions (i.e., pinpointing the sub-molecule of interest), and constrained generation of molecules. The benchmark proves to be challenging enough for several SoTA LLMs.

Reviewers provided initial scores (4/4/4) that highlight how the paper could be promising, but needs to be improved under some aspects. These include strengthening the motivation, as the benchmark deals with tasks that are not so relevant or challenging (i.e., could be solved by an algorithm easily), and providing an in-depth analysis of why current SoTA models fail and when. The authors provided a revised version of the paper that addresses the presentation-related concerns as well as a number of qualitative analysis that provide interesting failure modes.

I believe this paper is more interesting to the reasoning community than to the chemistry community, and should be treated as papers like [1,2,3] more than being used to solve any real-world chemistry-related task. I invite authors to take this perspective even more in their revised introduction and stress this comparison. As such, I find it interesting and insightful enough. Furthermore, I believe two reviewers would increase their scores (see below), so pointing towards acceptance.

As such the paper is accepted as poster.

[1] Li et al. 2025, Beyond Chemical QA: Evaluating LLM’s Chemical Reasoning with Modular Chemical Operations
[2] Valenti et al. 2024, ChemAlgebra: Algebraic Reasoning on Chemical Reactions
[3] Runcie et al. 2025, Assessing the Chemical Intelligence of Large Language Models

**Reviewer Concerns:**

The concern over the relevance and scope of the benchmark is still a bit open. Authors have improved their scope in the revised version and clarified how MolecularIQ posits itself in the broader panorama of chemistry-related benchmarks for LLMs and ML models. I agree that we don't need another benchmark to predict molecule properties (or similar discriminative tasks), neither another benchmark to generate molecules.

The concern regarding using SMILES is legit, but it is not a good reason to reject the paper per se. I think authors rebutted correctly to reviewer 31oX. The fact that LLMs will fail pray of shortcuts is a sign of the limitation of SoTA, not of the benchmark per se.

The request to have a more in-depth analysis of when and how SoTA models fail has been completely addressed by the authors who provided several new plots including insights on how "multi-task" complexity impacts performance, the effect of making syntax errors and type validity and some failures of sub-molecule structures.

**Reviewer Scores:**

yX2q - this review was on the more positve side and essentially just asked clarification questions and stated "the paper would be stronger with deeper qualitative error analysis". I believe authors fully answered the questions and delivered the requested analysis, therefore the reviewer would have increased the score to 6.

31oX - the reviewer would likely not have changed their score, as their concerns are related to intrinsic motivation of the paper. However, as I argued above, this should be found in the reasoning community and not in the chemistry community.

9zuz - very thourough review. I believe that the authors answered all comments well and improved the current paper presentation. No big concerns standing there.The reviewer might have likely bumped their score to 6.

---

> ### Public Comment · ~Johannes_Schimunek1 · 2026-03-01
> **Camera-Ready Revisions in Response to Meta Review**
>
> Dear Area Chair,
>
> Thank you for your constructive meta review and for positioning MolecularIQ within the broader reasoning literature.
>
> In preparing the camera-ready version, we focused on aligning the main manuscript more explicitly with the perspective you outlined — namely, presenting MolecularIQ as a structured reasoning benchmark operating over a formally defined symbolic domain, rather than primarily as a traditional chemistry application benchmark.
>
> The principal revisions to the main paper include:
>
> (1) Clearer reasoning-oriented framing in the introduction and related work.
> We refined the introduction to reduce emphasis on shortcomings of existing chemistry benchmarks and instead foreground MolecularIQ as a diagnostic stress test for structural and compositional reasoning. The benchmark dimensions (task type, multitask load, and molecular complexity) are framed more explicitly as controlled axes for probing reasoning behavior. We also reworked the related work section to situate MolecularIQ more directly within the literature on benchmarking structural and compositional reasoning, rather than primarily comparing against general chemistry capability benchmarks.
>
> (2) Representation-invariance and shortcut analysis.
> We expanded and sharpened the discussion of SMILES representation variants (canonical, randomized, Kekulized, and ring enumeration perturbations), explicitly interpreting systematic performance degradation under representation changes as evidence of syntactic shortcut reliance rather than representation-invariant structural reasoning.
>
> (3) Updated Figures 1 and 2.
> We redesigned Figures 1 and 2 to reflect these refinements, improving the visual communication of the benchmark’s structure, task taxonomy, and reasoning-focused positioning.
>
> We believe these revisions further strengthen the manuscript’s alignment with the reasoning-oriented perspective highlighted in your meta review.

---

### Decision · Program_Chairs · 2026-01-26

Accept (Poster)